# ShinkaEvolve: Towards Open-Ended and Sample-Efficient Program Evolution

**Robert Tjarko Lange**[*]
Sakana AI
robert@sakana.ai

**Yuki Imajuku**
Sakana AI
imajuku@sakana.ai

**Edoardo Cetin**
Sakana AI
edo@sakana.ai

## Abstract

We introduce ShinkaEvolve: a new framework leveraging large language models (LLMs) to advance scientific discovery with state-of-the-art performance and efficiency. The field of LLM-driven scientific discovery has seen significant progress, but has yet to overcome a critical limitation: sample inefficiency, requiring thousands of samples to identify effective solutions. ShinkaEvolve takes a concrete step towards addressing this critical limitation by introducing three key innovations: a parent sampling technique balancing exploration and exploitation, code novelty rejection-sampling for efficient search space exploration, and a bandit-based LLM ensemble selection strategy. When applied to the canonical circle-packing optimization task, ShinkaEvolve discovers a new state-of-the-art circle packing solution using only 150 samples, orders of magnitude fewer than prior frameworks. Furthermore, applied to a broader set of engineering problems, ShinkaEvolve designs robust agentic harnesses for AIME mathematical reasoning tasks, identifies improvements to ALE-Bench competitive programming solutions, and discovers novel mixture-of-expert load balancing loss functions to stabilize LLM training itself. We provide ShinkaEvolve's full code together with this submission, which will be open-sourced to accelerate open advancements to open-ended automated discovery across diverse computational problems.

## 1 Introduction

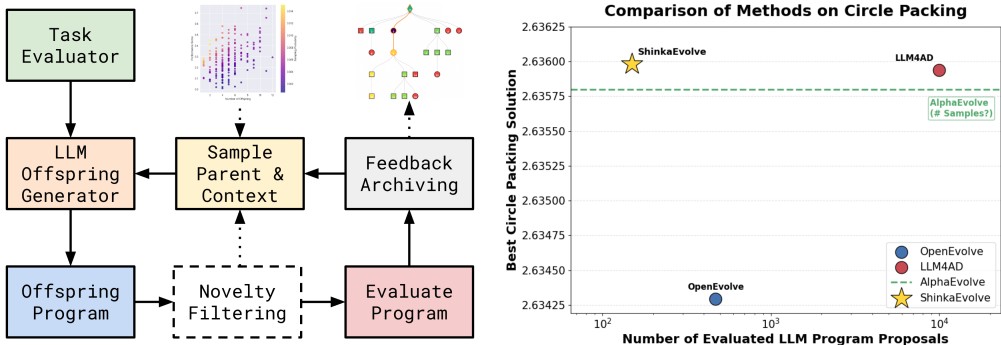

Figure 1: **High-level overview of ShinkaEvolve.** *Left:* ShinkaEvolve constructs an archive of evaluated programs, rejection-samples new ones, and evaluates their fitness. *Right:* ShinkaEvolve outperforms AlphaEvolve's circle packing solution in orders-of-magnitude fewer iterations.

The rapid advancement of large language models (LLMs) has transformed scientific discovery through agentic systems that autonomously conduct experiments and test hypotheses (Lu et al., 2024b; Yamada et al., 2025; Novikov et al., 2025; Zhang et al., 2025). These frameworks leverage LLMs as sophisticated mutation operators, iteratively refining candidate solutions with successful variants propagating through successive generations. This methodology has proven effective across domains such as competitive programming (Li et al., 2022), mathematical optimization (Romera-Paredes et al., 2024), and automated agentic design (Hu et al., 2024). However, current implementations face significant practical limitations. The primary challenge is substantial sample inefficiency

---

[*]Robert initiated and led the project; Robert and Edoardo were core code contributors. See Section 6 for detailed information. The code is available under https://github.com/SakanaAI/ShinkaEvolve.

as existing approaches typically require thousands of evaluations, making them computationally expensive and time-consuming. This inefficiency stems from naive exploration strategies that fail to effectively leverage accumulated knowledge from previous generations. SHINKAEVOLVE addresses these challenges through three key algorithmic innovations that work synergistically to enhance sample efficiency. Our adaptive parent and LLM sampling intelligently balances exploration of novel regions with exploitation of known high-quality areas. Next, our code proposal novelty rejection sampling ensures efficient program mutations. Finally, our bandit-based LLM selection strategy dynamically adapts to the evolving state of the sampled archive parents and inspiration programs. Experimental validation across diverse domains demonstrates substantial improvements in both efficiency and solution quality, with SHINKAEVOLVE achieving state-of-the-art results using orders of magnitude fewer evaluations than existing approaches. SHINKAEVOLVE's full code, provided with this submission, will be open-sourced to accelerate open advancements to automated discovery across a broader range of problems. In summary:

1. We introduce SHINKAEVOLVE, an evolutionary framework with three key algorithmic innovations: a novel parent program sampling strategy, code novelty rejection-sampling, and adaptive performance-based LLM ensemble selection.

2. We compare SHINKAEVOLVE with prior frameworks on the canonical circle-packing task, achieving state-of-the-art results with orders-of-magnitude fewer iterations.

3. We demonstrate SHINKAEVOLVE's ability to innovate beyond human and LLM-generated solutions across three additional engineering domains: agentic scaffolding (AIME), competitive programming (ALE-Bench), and LLM training design (mixture-of-expert loss).

## 2 RELATED WORK

**Evolutionary Code Optimization with LLMs**. One particular flavor of test-time compute is evolutionary code optimization: the usage, mutation, and recombination of previously generated code to produce new samples. This approach has been used to optimize reward and preference objectives (Lu et al., 2024a; Ma et al., 2023), mathematical science code (Romera-Paredes et al., 2024), and other applications (Lehman et al., 2022; Lange et al., 2024; Meyerson et al., 2023; Berman, 2025; Lange et al., 2025). Through prompting, LLMs are used as recombination engines (Lange et al., 2023; Meyerson et al., 2023) capable of simulating crossover between diverse code snippets and the rationales that produced them. These types of program archive-building systems resemble a population-based LLM-guided tree search (Jiang et al., 2025; Inoue et al., 2025). Most closely related to our work are *AlphaEvolve* (Novikov et al., 2025), *OpenEvolve* (Sharma, 2025), and *LLM4AD* (Liu et al., 2024a). We advance this line of work, demonstrating unprecedented sample efficiency with our combination of rejection-sampling, LLM prioritization, and online meta-scratchpad drafting.

**Open-Ended Agentic Discovery**. The integration of LLMs with open-ended evolutionary principles enables agentic systems capable of continuous innovation (Stanley et al., 2017; Zhang et al., 2025). Unlike traditional novelty search that relies on explicit diversity metrics (Lehman et al., 2008; Lehman & Stanley, 2011), LLM agents leverage learned representations to generate new solutions while maintaining semantic coherence (Faldor et al., 2024; Hu et al., 2024; Novikov et al., 2025). These agents construct evolutionary trees of programs where LLM-guided mutations connect related solutions across generations (Lehman et al., 2020). SHINKAEVOLVE systematically combines stepping stones, suboptimal intermediate solutions that serve as building blocks for breakthrough innovations, by employing LLM agents to both generate mutations and evaluate program relationships, enabling successful patterns to propagate across search branches rapidly.

## 3 METHOD

**Algorithm Overview.** SHINKAEVOLVE's control-flow entails three main phases:

1. *Parent and inspiration sampling* from an archive of island program subpopulations. Importantly, we emphasize the trade-off between exploration and exploitation in parent selection.

2. *Program mutation* via LLM-guided code edit proposals. We utilize novelty rejection-sampling based on code embedding similarity and an LLM-as-a-novelty-judge assessment.

3. *Program execution and world feedback* guiding the LLM ensemble selection probabilities and online meta-scratchpad drafting for documentation and knowledge diffusion.

### 3.1 PARENT AND INSPIRATION SAMPLING

**Archive Maintenance, Island Populations & Mutation Context Construction**. SHINKAEVOLVE maintains a fixed-size archive of previously evaluated programs with fitness scores and meta information, implementing an elite size constraint. The mutation context incorporates a primary parent program alongside inspiration programs drawn from top-performing solutions and random archive samples, providing the LLM with diverse exemplars for creative recombination. We follow Romera-Paredes et al. (2024); Novikov et al. (2025) and employ an island model approach with independent subpopulations seeded from the same initial program. The islands evolve in parallel to enhance diversity and prevent premature convergence. Island members can occasionally migrate between islands to diffuse knowledge across "discovery substreams". To protect the uniqueness of each island, we prevent the island-specific best-performing program from migrating (Tanese, 1989; Romera-Paredes et al., 2024). Sampling occurs hierarchically: with the island ID first sampled uniformly from the archive, later used as the origin for both parent and inspirations.

**Balancing Exploration & Exploitation: Parent Program Selection**. Given an island subpopulation, SHINKAEVOLVE implements multiple different parent sampling strategies that balance exploration and exploitation. First, we employ power law sampling where programs are ranked by fitness with ranks $r_i$ ($r_i = 1$ for the best program). The selection probability follows $p_i = \frac{r_i^{-\alpha}}{\sum_{j=1}^{n} r_j^{-\alpha}}$, where $\alpha$ controls exploitation intensity. Setting $\alpha = 0$ yields uniform sampling, while $\alpha \to \infty$ implements hill-climbing. Inspired by Zhang et al. (2025), we contrast this with weighted sampling, incorporating performance and novelty. Given programs, $P_i$, with offspring count $N(P_i)$, we first compute the median fitness $\alpha_0 = \text{median}(\{F(P_1), F(P_2), ..., F(P_n)\})$. The performance component uses sigmoid scaling: $s_i = \sigma(\lambda \cdot (F(P_i) - \alpha_0))$ where $\sigma(x) = \frac{1}{1+e^{-x}}$ and $\lambda$ controls selection pressure. The novelty component $h_i = \frac{1}{1+N(P_i)}$ favors programs with fewer offspring. The final probability combines these: $p_i = \frac{w_i}{\sum_{j=1}^{n} w_j}$ where $w_i = s_i \cdot h_i$ balances performance and novelty. By default, SHINKAEVOLVE uses the weighted sampling strategy. We provide a visual comparison of all these strategies in Figure 2 below:

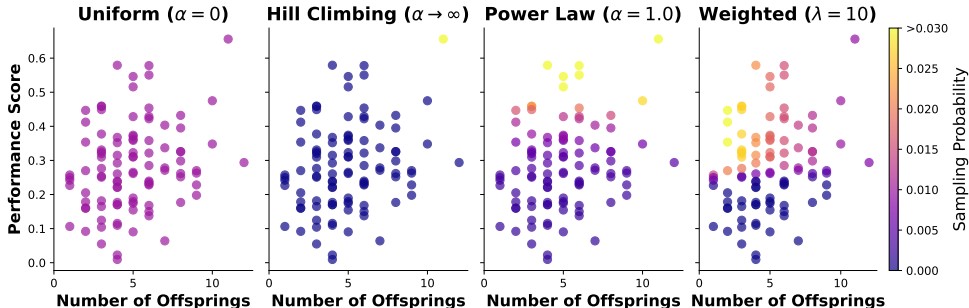

Figure 2: **SHINKAEVOLVE Parent Sampling.** The strategies range from pure exploration (uniform sampling) to pure exploitation (hill-climbing) to a combination of performance and novelty.

### 3.2 PROGRAM MUTATION AND NOVELTY ASSESSMENT

**LLM-Guided Program Mutations**. To generate new programs, SHINKAEVOLVE starts by sampling a specific LLM and a set of sampling parameters (e.g., temperature or reasoning budget) from a pre-specified pool. Our framework provides support for models from leading API providers, including GPT, Gemini, Claude, and DeepSeek (OpenAI, 2023; Team, 2025; Anthropic, 2024; Guo et al., 2025). After sampling a model, SHINKAEVOLVE employs three distinct mutation approaches to foster diversity and creativity in the LLM-generated program variants:

1. **Diff-Based Edits.** We implement diff edits using LLMs following the approach outlined in Novikov et al. (2025), utilizing SEARCH/REPLACE blocks for targeted modifications.

2. **Full Rewrites.** We enable full program rewrites to allow greater flexibility, programmatically ensuring that non-mutable blocks remain unchanged during the LLM rewrite process.

3. **Crossover Mutation.** We leverage crossover mutations (Lehman et al., 2022; Lange et al., 2025) where an LLM is prompted to combine the parent and an additional archive program.

Following Novikov et al. (2025), we use text markers (EVOLVE-BLOCK-START & EVOLVE-BLOCK-END) to ensure that immutable code is not changed during LLM rewrites. After a code change proposal, we enforce that the immutable code is not touched and resample a new proposal if a patch is invalid, providing parsing feedback using Reflexion (Shinn et al., 2024).

**Program Diversity via Novelty Rejection Sampling**. To enhance the creativity of executed code proposals, we leverage an LLM ensemble combined with temperature sampling. Additionally, we introduce *code novelty rejection sampling* using an embedding model to embed mutable parts of the program code. Afterwards, we compute cosine similarity scores across the island subpopulation programs. If the maximal score exceeds a threshold (e.g., $\eta = 0.95$), we query an LLM to further assess whether the program is meaningfully different. The approach is illustrated in Figure 3:

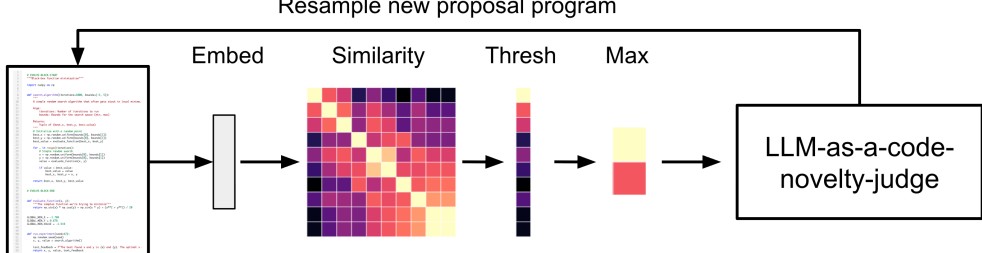

Figure 3: **SHINKAEVOLVE Program Novelty Rejection Sampling.** SHINKAEVOLVE embeds mutable code snippets, computes similarities across the archive; if the maximal score exceeds a threshold, another LLM is queried to assess whether the program is meaningfully novel.

### 3.3 EXECUTION AND WORLD FEEDBACK

**Multi-Objective Optimization & Textual Feedback**. After a program obtained with the above steps is executed, SHINKAEVOLVE performs multi-objective assessment yielding both its scalar fitness value $r_i$ together with a set of exposed "public metrics" and textual feedback. SHINKAEVOLVE then stores this full multi-objective assessment in the population archive to provide an informative context for future generations of language model mutations using a simple prompting format:

```
Example of Diff Edit Prompt with Textual Feedback

# Current program
Here is the current program we are trying to improve (you will need to propose a modification to it below):
```{language}
{code_content}
```
Here are the performance metrics of the program:
{performance_metrics}{text_feedback_section}

# Instructions
...
# Task
...
IMPORTANT: Do not rewrite the entire program - focus on targeted improvements.
```

**Adaptive LLM sampling evolution**. The performance of different LLMs to propose mutations can vary across problem domains and based on the current state of the sampled archive parents and inspiration programs. SHINKAEVOLVE dynamically adapts to this non-stationarity by evolving the LLM sampling probability throughout at the end of each generation. Our approach is based on the UCB1 algorithm (Auer et al., 2002), associating each LLM with a visitation counter and an estimate of the expected score updated with the performance of its sampled mutations. We introduce changes tailored to the domain of LLM-driven discovery. In particular, rather than the absolute fitness of each mutation $F(P_i)$, we update the LLM distribution using: $F(P_i)^u = \exp\big(\max(F(P_i) - F(P_i)^b, 0)\big) - 1$, where $F(P_i)^b$ is the baseline reward for program $i$ computed as the maximum between its parent program and the initial program in the database, ensuring each LLM is evaluated based on its relative improvement to account for the non-stationarity of the program archive. At the same time, the $\exp(\cdot)$ and $\max(\cdot, 0)$ operations help precisely promote LLMs able to come up with bold, high-risk, high-reward mutations, over "safer" minor improvements. We use the tracked statistics over the observed rewards to normalize $F(P_i)^u$ and ensure invariance to the fitness scale of each domain.

**Meta-Scratchpad & Online Refinement**. SHINKAEVOLVE implements a meta-scratchpad system that periodically analyzes successful solutions. Every $T$ generations, we summarize the recent program evaluations and identify common optimization strategies and design principles. The meta-agent synthesizes insights into actionable recommendations appended to the mutation prompt, providing high-level guidance from accumulated evolutionary experience.

## 4 RESULTS

In this Section, we demonstrate how SHINKAEVOLVE's innovations lead to concrete breakthroughs across four relevant scientific and engineering domains. Furthermore, we conclude by providing an in-depth ablation analysis quantifying the significance of each of SHINKAEVOLVE's main components. To complement the shared code, we also refer the interested readers to Appendix B for full implementation details and hyperparameter configurations together with Appendix D for program listings representing each of SHINKAEVOLVE's final solutions.

### 4.1 CIRCLE PACKING: REPRODUCING & IMPROVING ALPHAEVOLVE RESULTS

**Task Description.** The circle packing optimization problem requires placing 26 circles within a unit square such that the sum of their radii is maximized while ensuring no circles overlap and all circles remain fully contained within the square boundary. This constrained optimization challenge combines discrete placement decisions with continuous radius optimization, making it a complex benchmark for evolutionary algorithms. The problem exhibits multiple local optima and requires sophisticated strategies to discover high-quality solutions without suboptimal space allocation.

**SHINKAEVOLVE's Discovery Dynamics.** SHINKAEVOLVE was executed for only 150 evolutionary generations before finding a state-of-the-art solution, in contrast to existing approaches using at least thousands of evaluations (Figure 1). Figure 4 (left) shows the improvement trajectory exhibits three distinct phases: an initial rapid improvement phase where the algorithm quickly discovers fundamental radii optimization strategies, a sustained exploration phase with incremental gains as more sophisticated techniques emerge (constraint-based optimization), and a final convergence phase where the best solutions are refined through restarts. The tree structure in Figure 4 (right) reveals how successful innovations propagate through the population, with high-performing solutions (green and yellow) serving as parents for subsequent generations. Notably, the algorithm demonstrates sophisticated exploration patterns, with multiple evolutionary branches exploring different algorithmic approaches before converging toward the optimal solution path shown in black.

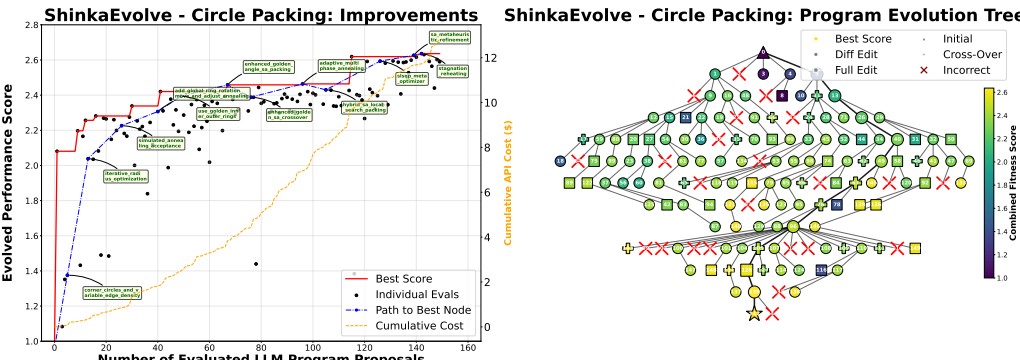

Figure 4: **SHINKAEVOLVE on Circle Packing Task.** *Left:* SHINKAEVOLVE outperforms AlphaEvolve's solution in less than 150 program evaluations. *Right:* SHINKAEVOLVE's program evolution tree demonstrates the iterative composition of stepping stones into high-performing solutions.

**SHINKAEVOLVE's Discovered Solution.** The final program (Section D.1) combines three key innovations: (1) a sophisticated initialization that places circles in a structured golden-angle spiral pattern with strategic corner and edge positioning, (2) a hybrid optimization approach integrating SLSQP gradient-based refinement with simulated annealing for global exploration, and (3) intelligent perturbation mechanisms that alternate between local circle movements and global ring rotations to escape local optima. The discovered solution employs adaptive temperature scheduling with reheating strategies to prevent premature convergence, while maintaining feasibility through constraint-aware radius computation. This multi-level approach, from structured initialization through meta-heuristic exploration to gradient-based polishing, exemplifies how SHINKAEVOLVE can discover effective algorithmic compositions that outperform hand-designed baselines.

## 4.2 AIME: EVOLVING AGENT SCAFFOLDS FOR MATH REASONING

**Task Description.** We evaluate SHINKAEVOLVE on AIME 2024 (AIM, 2024) mathematical reasoning problems, consisting of 30 challenging competition-level questions requiring sophisticated problem-solving strategies (Hu et al., 2024). The task involves evolving agent scaffold designs constrained to a maximum of 10 LLM queries per problem for computational efficiency. Using gpt-4.1-nano as the base model, we discover scaffold designs for 75 generations, with each candidate evaluated across three independent runs on the complete question set.

**SHINKAEVOLVE's Discovery Dynamics.** SHINKAEVOLVE discovers scaffold designs that significantly outperform hand-designed baselines, including simple single-query agents and sophisticated majority-voting approaches. The search reveals a Pareto frontier between efficiency and performance (Figure 5, left), with 7 LLM queries yielding maximum performance while an alternative scaffold achieves comparable results using the full 10-query budget. We evaluate generalization by testing on 2023 and 2025 AIME problems, displaying different transfer patterns (Figure 5, middle): smaller improvements on 2023 problems suggest potential saturation due to training data contamination, while larger gains on 2025 problems indicate successful generalization to recent, unseen challenges. Cross-LLM model transfer experiments validate robustness, with successful adaptation to gpt-4.1-mini, gpt-4.1, and o4-mini demonstrating that discovered architectures capture generalizable strategies rather than model-specific optimizations (Figure 5, right).

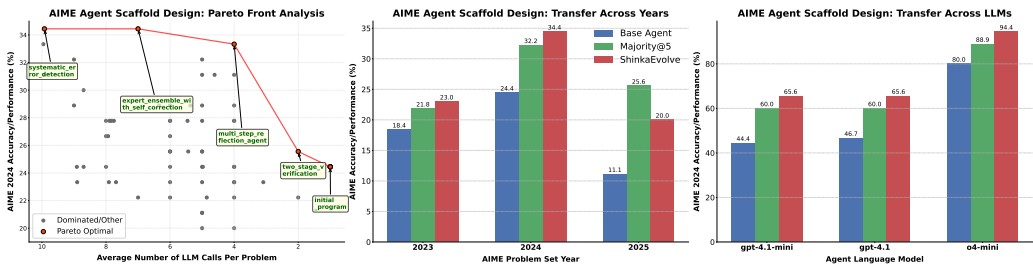

Figure 5: **SHINKAEVOLVE for Agent Scaffold Design.** *Left:* SHINKAEVOLVE discovers a Pareto frontier between performance and query budget. *Middle:* The discovered scaffold generalizes to unseen AIME problems. *Right:* The scaffold boosts performance regardless of the underlying LLM.

**SHINKAEVOLVE's Discovered Solution.** The evolved agent implements a three-stage architecture leveraging diverse expert personas, critical peer review, and synthesis mechanisms. Three specialized experts generate independent solutions using distinct approaches: a meticulous step-by-step reasoner, an intuitive pattern-recognition specialist, and an algorithmic computer science-oriented mathematician, each operating at 0.7 temperature. The second stage introduces critical peer review, where each solution undergoes rigorous scrutiny from a skeptical reviewer at low temperature (0.1). The reviewer validates pattern-based reasoning by testing patterns on multiple examples, identifies logical flaws, and provides corrections when necessary, significantly improving solution quality. The final synthesis stage employs an editor-in-chief persona operating at zero temperature to analyze all solutions and critiques, identify the most reliable approach, and construct a canonical solution. Robust fallback mechanisms resort to majority voting among reviewed solutions, then original solutions, ensuring reliable output when components fail. This architecture effectively utilizes 7 LLM calls (3 generation + 3 review + 1 synthesis) even less than the specified 10-call constraint.

## 4.3 ALE-BENCH: EVOLVING PROGRAMS FOR COMBINATORIAL OPTIMIZATION

**Task Description.** We apply SHINKAEVOLVE to ALE-Bench LITE (Imajuku et al., 2025), a collection of 10 competitive programming contests hosted by AtCoder to test the performance of LLMs on heuristic problems. We explore SHINKAEVOLVE's ability to improve high-performing solutions using the best programming solution from ALE-Agent (Imajuku et al., 2025) as an initial program. We run SHINKAEVOLVE for 50 generations, using the public set score as the fitness function. We then submit and report the score of our final solution to the private test set.

**SHINKAEVOLVE's Discovery Dynamics.** SHINKAEVOLVE is able to improve the solutions discovered by ALE-Agent by approximately 2.3% across the 10 tasks on average (Figure 6). Furthermore, on task ahc039, SHINKAEVOLVE's final solution even outperformed the second place submission on the AtCoder leaderboard. These notable gains came from fine-grained refinements that preserved the high-level algorithmic structure to ALE-Agent's solutions.

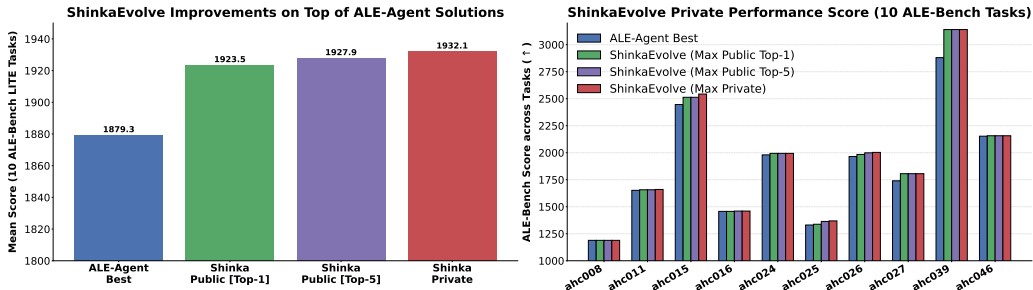

Figure 6: **SHINKAEVOLVE for Improving ALE-Bench solutions.** *Left:* SHINKAEVOLVE improves ALE-Agent's solution by ∼ 2.3%. *Right:* On one task, ahc039, the solution improved from 5th to 2nd place submission on the AtCoder leaderboard if it had participated in the contest.

**SHINKAEVOLVE's Discovered Solution.** We focus on two tasks to illustrate the discovered improvements of SHINKAEVOLVE, ahc039 and ahc025. The objective of ahc039 is to find an optimal, axis-aligned polygon to maximize the number of mackerels it contains minus the number of sardines, subject to given constraints. The base solution by ALE-Agent applies simulated annealing with kd-tree data structure (5th, 2880 performance). SHINKAEVOLVE further improved the solution (2nd, 3140 performance) by introducing modifications such as caching the validation process and enhancing neighborhood operators. For the caching, the kd-tree was augmented to cache subtree statistics, including bounding boxes and fish counts, at each node. For the neighborhood operators, a novel "targeted edge move" was introduced, which heuristically identifies a misclassified fish (e.g., a mackerel outside the polygon) and greedily moves the nearest edge to correct its state. These changes strengthened the directionality of the search. For ahc025, the task is to use a balance scale to compare the total weights of any two subsets of items, aiming, after a fixed number of weighings, to partition the items into groups with as equal total weights as possible. SHINKAEVOLVE improved the ALE-Agent's simulated annealing baseline by introducing faster caching, refining fallback weight estimation, and ultimately replacing simulated annealing with a more focused optimization combining greedy moves and targeted local search. Comparison with top human solutions suggests that for many tasks, there is ample room for improvement. Furthermore, often times SHINKAEVOLVE tended to explore modifications staying close to the ALE-Agent's solution. This indicates the potential of overfitting to the initialization solution.

## 4.4 LLM TRAINING: EVOLVING LOSSES FOR BALANCED AND EFFECTIVE EXPERTS

**Task Description.** The Mixture-of-Expert (MoE) architecture (Szymanski & Lemmon, 1993; Shazeer et al., 2017; Lepikhin et al., 2020; Fedus et al., 2022) has been a critical advancement, ubiquitous amongst modern open and closed-source flagship models (Google AI Blog, 2024; Guo et al., 2025; Meta-AI, 2025; Yang et al., 2025; Team, 2025). The basic idea is simple: replace traditional large feed-forward residual blocks with ensembles of efficient smaller modules (the "experts") that can each specialize in distinct problem domains (Fedus et al., 2022). For each MoE layer and token, only the outputs of the top-K experts selected by a router classifier are computed, effectively splitting the computation and making both training and inference cheaper and faster. However, due to the non-differentiability of the top-K expert selection operation, it is critical to provide the router with an auxiliary load balancing loss (LBL), which serves to avoid early collapse toward uneven expert distribution of the token load. We deploy SHINKAEVOLVE precisely to tackle this open architectural design challenge, which has been one core focus driving recent MoE advancements (Shazeer et al., 2017; Fedus et al., 2022; Du et al., 2022; Zoph et al., 2022; Xue et al., 2024; Dai et al., 2024; Qiu et al., 2025; Muennighoff et al., 2024): Devising an effective load balancing loss to incentivize efficiency and specialization, without hindering the model's expressivity.

**SHINKAEVOLVE's Discovery Dynamics.** We ground the problem of LBL design by pretraining a MoE model with 556M parameters, $N_E = 64$ total experts of which only $K = 8$ active for any given token. This results in only 82M parameters sparsely activated in each forward pass, excluding the token embeddings. We train this small model on over 2B tokens from fineweb (Penedo et al., 2024) by minimizing the MoE loss function, computed by adding the LBL, weighted by $\lambda = 0.01$, to the model's cross-entropy loss (CE). The fitness function of each program then measures a simple objective: minimize the sum of the final CE together with the model's overall "load imbalance" as measured by the L1 deviation from a uniform distribution of tokens between the MoE experts. Given

the cost of pretraining, we run SHINKAEVOLVE for only 30 iterations. We evaluate the generality of SHINKAEVOLVE's best-performing solutions by training a much larger MoE with 2.7B parameters on slightly under 30B fineweb tokens across three LBL coefficients $\lambda \in \{0.001, 0.01, 0.1\}$, yielding different levels of regularization. We compare this solution against the "global-batch LBL" used to train some of the most popular open LLMs (Yang et al., 2025), in terms of final perplexity (Figure 7, left) and end task performance (Figure 7, center) as evaluated across different downstream benchmarks (Talmor et al., 2018; Zellers et al., 2019; Mihaylov et al., 2018; Bisk et al., 2020; Sap et al., 2019; Sakaguchi et al., 2021; Clark et al., 2018). We provide our results below as a function of load imbalance, showing that SHINKAEVOLVE's new loss achieves a consistent edge across our training configurations, growing larger with the value of the $\lambda$ coefficient.

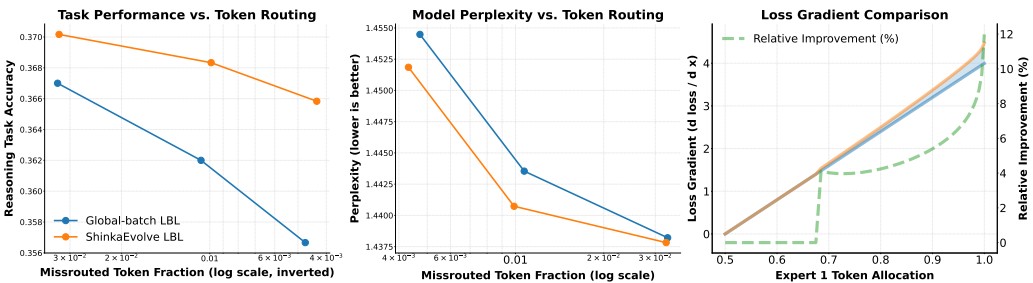

Figure 7: **SHINKAEVOLVE for discovering Mixture-of-Experts Load Balancing Loss Functions.** *Left:* Downstream task performance across seven benchmarks. *Middle:* Final perplexity across missroute fraction levels. *Right:* Load imbalance gradient as a function of token allocation.

**SHINKAEVOLVE's Discovered Solution.** The discovered LBL is a new twist on the established global-batch LBL, which was used for seeding the evolutionary search. SHINKAEVOLVE complements this popular LBL with a new term, specifically targeted toward regularizing the MoE layers with individual under-specialized experts. Concretely, let $f_{\ell,i}$ and $P_{\ell,i}$ correspond to the selection frequency and the average router probabilities for each expert $i$ located in layer $\ell$. SHINKAEVOLVE's LBL uses a normalized complement to the entropy in each layer $s(P_\ell) = 0.5 + \left(1 - \frac{H(P_\ell)}{\log N_E}\right)$ and a minimum usage threshold target $\tau = 0.064/N_E$ to compute:

$$L_{\text{LBL}} = \underbrace{N_E \cdot \frac{1}{L} \sum_{\ell=1}^{L} \sum_{i=1}^{N_E} f_{\ell,i}\, P_{\ell,i}}_{\text{Global-batch LBL}} + \underbrace{\frac{0.1}{L} \sum_{\ell=1}^{L} s(P_\ell) \sum_{i=1}^{N_E} \max\!\left(0, \tau - f_{\ell,i}\right)}_{\text{SHINKAEVOLVE new regularization}}. \tag{1}$$

The effects of SHINKAEVOLVE's new regularization term can be seen through its induced gradients acting on the router's token allocation in a simplified two-expert scenario (Figure 7, right). Intuitively, this term affects the MoE router of any layer where experts are allocated a fraction of tokens less than $\tau$. The multiplier $s(P_\ell)$ strengthens this push when the layer's routing entropy $H(P_\ell)$ is low and the router concentrates on a few dominating experts. This closes a blind spot of the global-batch LBL: the dot product $f \cdot P$ can look "balanced" even if few experts are barely touched. Thus, this term can be seen as a safety net that adaptively activates and vanishes once an expert crosses the floor, preventing dead experts and avoiding over-regularizing well-balanced layers. We refer to Appendix B, for further results and an extended discussion on how SHINKAEVOLVE differs from prior approaches.

## 5 ABLATIONS & ANALYSIS

**Impact of Parent Selection Strategies.** To understand the importance of parent selection, we compare different strategies for choosing which programs to evolve. The *Best-of-N* baseline ignores the evolutionary history, always using the initial program as parent without feedback. In contrast, *Hill Climbing* represents a greedy approach that only selects the highest-performing program as the parent for mutations. Our proposed *Weighted Sampling* strategy balances exploration and exploitation by probabilistically selecting parents based on their fitness and number of offspring.

*Takeaways.* Weighted sampling consistently outperforms random search and hill climbing across all tasks. Hill climbing shows strong initial performance but plateaus quickly, while weighted sam-

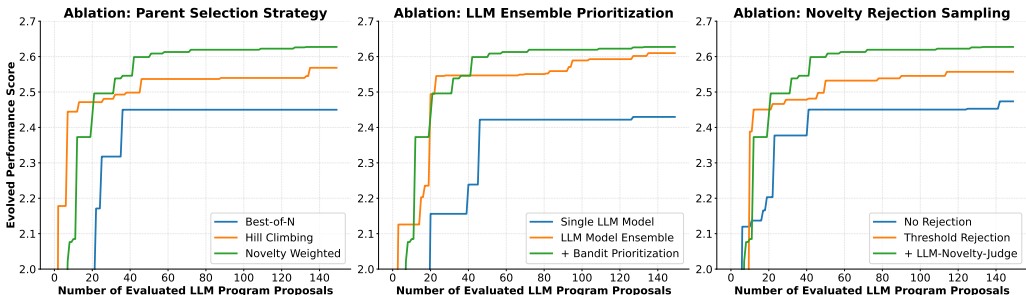

Figure 8: **SHINKAEVOLVE Method Ablation Studies on Circle Packing.** *Left:* Weighted parent sampling outperforms random search and hill climbing. *Middle:* Bandit-based LLM ensembling slightly improves the performance over a fixed uniform ensemble distribution. *Right:* Embedding-based rejection sampling with LLM as a novelty judge strongly outperforms no rejection sampling.

pling maintains steady improvement throughout evolution. Random search demonstrates the poorest convergence, highlighting the importance of leveraging fitness-based parent selection.

**Impact of LLM Ensembling and Prioritization.** Evolutionary agents can benefit from diverse coding capabilities by leveraging multiple LLMs. We investigate this hypothesis by comparing a *Single LLM* baseline (GPT-5-nano) against ensemble approaches. The *Fixed LLM Ensemble* provides diversity by sampling uniformly from a predetermined set of models, while our *Bandit-Based LLM Ensemble* adaptively learns which models contribute most effectively to fitness improvements, balancing exploration of underutilized models with exploitation of high-performing ones.

*Takeaways.* The bandit-based LLM ensemble significantly outperforms both single LLM and fixed ensemble approaches. While the fixed ensemble shows moderate improvements over single LLM usage, the adaptive bandit strategy achieves the highest performance by dynamically prioritizing more effective models based on their contribution to fitness improvements.

**Impact of Code Embedding-Based Rejection Sampling.** Similar code variants can waste computational resources without advancing the search frontier. To address this challenge, we examine different novelty filtering mechanisms. The *No Rejection Sampling* baseline accepts any LLM proposal, potentially allowing near-duplicate programs to proliferate. Our *Embedding-Based Rejection Sampling* approach leverages text embeddings to identify and reject proposals with similarity scores exceeding 0.95. We also explore an *Additional LLM-as-a-novelty-judge* variant that supplements embedding-based filtering with explicit LLM assessment of program novelty.

*Takeaways.* Code embedding-based rejection sampling provides substantial performance gains over no rejection sampling by preventing redundant mutations. The additional LLM-as-a-novelty-judge offers marginal improvements, suggesting that embedding similarity is already an effective proxy for novelty assessment without requiring additional computational overhead.

## 6 DISCUSSION

**Contributions.** This work introduces SHINKAEVOLVE, an evolutionary framework tackling the inefficiency of LLM-driven scientific discovery. SHINKAEVOLVE achieves state-of-the-art results across four domains: circle packing with 150 evaluations (orders of magnitude improvement over prior baselines), sophisticated AIME reasoning scaffolds, ALE-Bench algorithmic improvements, and novel mixture-of-expert load balancing. By sharing its full code, we hope to remove barriers and accelerate community-driven open advancements.

**Limitations.** While SHINKAEVOLVE makes significant strides toward improving sample efficiency and reducing costs, it still shares some of the other limitations of prior approaches (Novikov et al., 2025). In particular, our framework still requires manual task specification, relying on human expertise in the target domain for providing objective functions. Furthermore, SHINKAEVOLVE is still constrained to problems with well-defined, implemented numerical objectives, making its wider applicability to arbitrary human preferences and heuristics an open problem.

**Extensions.** Automated task specification through LLM task generation could enable greater autonomy and unlock applications in unexplored domains. Transitioning to true open-endedness, where systems generate their own objectives, represents a new compelling future frontier.

## ETHICS STATEMENT

SHINKAEVOLVE's aims to further advance the field of evolutionary optimization and make it accessible to researchers and practitioners previously lacking access to proprietary frameworks, following on the same path as Sharma (2025). Given its purpose and objective, we thus do not expect additional specific issues regarding fairness, privacy, or security, or any other harmful societal implications that are not already inherent to the field. However, we still want to highlight that our framework relies on closed-source models, and API costs from large-scale LLM usage could create economic barriers, potentially constraining democratization goals.

## REPRODUCIBILITY STATEMENT

We provide the full anonymized SHINKAEVOLVE code in the supplementary material uploaded with this submission. Moreover, we provide full implementation details and hyperparameter configurations in Appendix B, together with program listings representing each of SHINKAEVOLVE's final solutions in Appendix D. We will also open-source a fully-documented version of SHINKAEVOLVE's code to facilitate open reproducibility and accelerate advancements to open-ended automated discovery across diverse computational problems.

## LLM USAGE DISCLOSURE

The authors would like to acknowledge the use of LLMs to improve the grammar, clarity, and overall presentation of this manuscript. The authors reviewed, edited, and take full responsibility for the final content.

## ACKNOWLEDGMENTS

We thank David Ha, Takuya Akiba, Taishi Nakamura, Luca Grilloti, Yutaro Yamada, and the rest of the Sakana AI team for helpful discussions throughout the project.

## AUTHOR CONTRIBUTION LIST

**Robert Tjarko Lange**   Initiated and led the project, designed the core SHINKAEVOLVE codebase, and implemented as well as collected results for Circle Packing, AIME, and ALE-Bench. Wrote the manuscript.

**Yuki Imajuku**   Helped setting up the ALE-Bench infrastructure, advised on the ALE-Bench results and supported writing the manuscript section on ALE-Bench.

**Edoardo Cetin**   Was involved in design discussions for SHINKAEVOLVE and came up as well as implemented the adaptive LLM sampling method and the Hydra configuration. He implemented and collected results for LLM training and Mixture-of-Experts evolution. Co-wrote the manuscript.

## SHINKAEVOLVE NAMING

The Japanese term 'shinka' translates to 'evolution' or 'innovation'. SHINKAEVOLVE refers to the vision of an open-ended self-refining innovation engine. While SHINKAEVOLVE might be a bit funny for some Japanese readers, as it sounds like Evolve-Evolve or 進化ー進化, similar bilingual repetitive terms appear quite often in Japan.

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

APPENDIX

# A  SHINKA IMPLEMENTATION DETAILS

- SHINKAEVOLVE uses a queue based implementation where LLMs generate program proposals sequentially. Afterwards, they are added to a job evaluation queue. Each proposal is based on all jobs that have completed so far and are stored in the database.

- Throughout development, we experimented with a fully asynchronous implementation that leverages both a job and a proposal queue. This allows for higher throughput but introduces a degree of "off-archiveness" in the sense that new code proposals are generated in advance and not based on all the previously submitted jobs. Furthermore, jobs from faster to query models will be executed earlier since their proposal jobs will be processed earlier. Many open research questions remain regarding the optimal trade-off between throughput, sample efficiency, and off-archiveness.

- Below we provide an overview of the Python API. It roughly adopts the high-level interface of `OpenEvolve` (Sharma, 2025):

```python
from shinka.core import EvolutionRunner, EvolutionConfig
from shinka.database import DatabaseConfig
from shinka.launch import LocalJobConfig

# Minimal config - only specify what's required
job_config = LocalJobConfig(eval_program_path="evaluate.py")
db_config = DatabaseConfig()
evo_config = EvolutionConfig(init_program_path="initial.py",)

# Run evolution with defaults
runner = EvolutionRunner(
    evo_config=evo_config,
    job_config=job_config,
    db_config=db_config,
)
runner.run()
```

Listing 1: Minimal SHINKAEVOLVE configuration and usage example.

### `evaluate.py` - Evaluation Script

```python
from shinka.core import run_shinka_eval

def main(program_path: str,
         results_dir: str):
    metrics, correct, err = run_shinka_eval(
        program_path=program_path,
        results_dir=results_dir,
        experiment_fn_name="run_experiment",
        num_runs=3, # Multi-evals to aggreg.
        get_experiment_kwargs=get_kwargs,
        aggregate_metrics_fn=aggregate_fn,
        validate_fn=validate_fn,  # Optional
    )

def get_kwargs(run_idx: int) -> dict:
    return {"param1": "value", "param2": 42}

def aggregate_fn(results: list) -> dict:
    score = results[0]
    text = results[1]
    return {
        "combined_score": float(score),
        "public": {...},  # shinka-visible
        "private": {...},  # shinka-invisible
        "extra_data": {...},  # store as pkl
        "text_feedback": text,  # str fb
    }

if __name__ == "__main__":
    # argparse program path & dir
    main(program_path, results_dir)
```

### `initial.py` - Starting Solution

```python
# EVOLVE-BLOCK-START
def advanced_algo():
    # This will be evolved
    return solution
# EVOLVE-BLOCK-END

def run_experiment(**kwargs):
    """Main called by evaluator"""
    result = solve_problem(kwargs)
    return result

def solve_problem(params):
    solution = advanced_algo()
    return solution
```

## B   TASK IMPLEMENTATION DETAILS

### B.1   CIRCLE PACKING PROBLEM

**Detailed Task Description.** The circle packing task requires placing 26 circles within a unit square such that the sum of their radii is maximized while ensuring no circles overlap and all circles remain fully contained within the square boundary.

**Verification Methodology with Slack.** For the main SHINKAEVOLVE run presented in the paper, we employed the verification script provided by `OpenEvolve` (Sharma, 2025), which allows for $1 \times 10^{-6}$ numerical slack. To ensure the robustness of our results, we additionally validated our solutions using `AlphaEvolve`'s (Novikov et al., 2025) exact verification code. We found that our discovered solution can be made trivially exact by reducing each circle's radius by $1 \times 10^{-8}$, demonstrating the high precision of our approach. The adjustment from the relaxed to exact formulation reduces the sum of radii for our discovered solution by a negligible amount, from 2.635983099011548 to 2.6359828390115476, representing a relative change of less than $10^{-6}$.

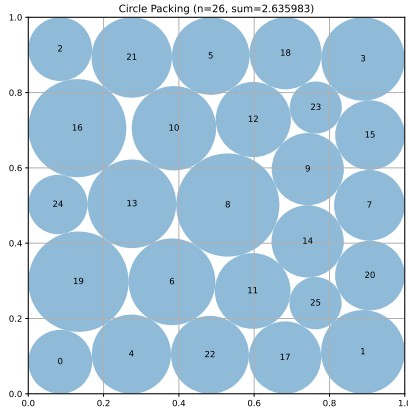

Figure 9: Discovered Circle Packing solution by SHINKAEVOLVE.

**Verification Methodology with Exact Constraint.** Additionally, we replicated the solution using the exact verification code from `AlphaEvolve` Figure 10 with a score of 2.63597770931127. The discovery of the solution requires more samples to be evaluated. This illustrates an important principle: surrogate relaxed tasks can be effectively used during evolution and subsequently post-processed to discover exact state-of-the-art solutions.

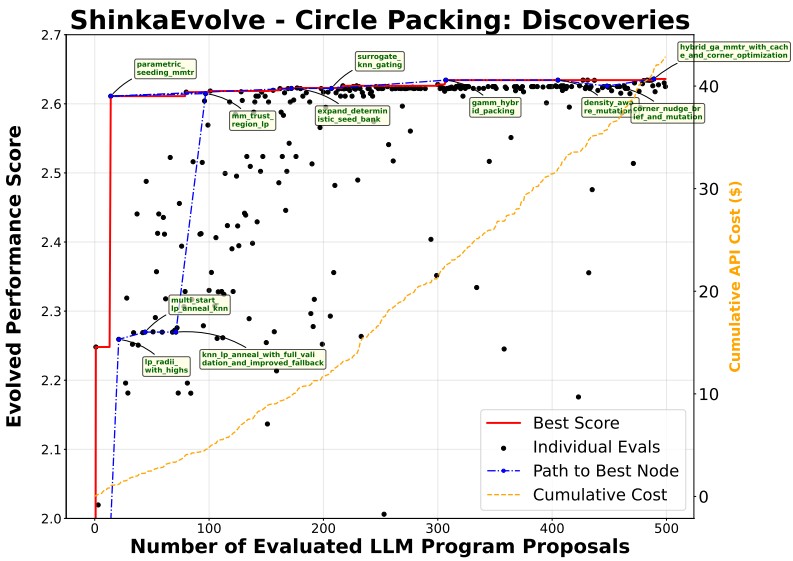

Figure 10: Circle packing asynchronous evolution results for exact circle packing verification showing convergence behavior and solution quality over time.

**Baseline Comparisons.** Our performance benchmarks are established against solutions from three primary sources. The AlphaEvolve sum of radii is taken from their paper (Novikov et al., 2025). The `OpenEvolve` baseline scores are derived from their official implementation and examples available in their repository. Additionally, we compare against `LLM4AD` results, specifically their circle packing implementations and Evolution of Heuristics (EoH) experimental configurations. These

baselines provide comprehensive coverage of existing automated algorithm design approaches, enabling fair and thorough performance evaluation of our method.

**SHINKAEVOLVE's Hyperparameter Configuration.**

| Parameter | Value | Parameter | Value |
|---|---|---|---|
| **Database configuration** | | | |
| Archive size | 40 | Elite selection ratio | 0.3 |
| Archive inspirations | 4 | Top-$k$ inspirations | 2 |
| Migration interval | 10 | Migration rate | 0.0 |
| Island elitism | true | Parent selection strategy | weighted |
| Parent selection $\lambda$ | 10.0 | Number of islands | 2 |
| **Evolution configuration** | | | |
| Patch types | [diff, full, cross] | Patch type probs | [0.45, 0.45, 0.1] |
| Generations | 150 | Max parallel jobs | 5 |
| Max patch resamples | 3 | Max patch attempts | 3 |
| Meta recommendation interval | 10 | Max meta recommendations | 5 |
| Embedding model | text-embedding-3-small | Max novelty attempts | None |
| Code embed sim threshold | 0.95 | Problem implementation | Python |
| LLM dynamic selection | ucb1 | Exploration coefficient | 1.0 |
| **LLM models** | | | |
| gemini-2.5-pro | × | gemini-2.5-flash | × |
| claude-sonnet-4 | ✓ | o4-mini | ✓ |
| gpt-5 | × | gpt-4.1-nano | ✓ |
| gpt-4.1 | ✓ | gpt-4.1-mini | ✓ |
| **LLM settings** | | | |
| Temperatures | [0.0, 0.5, 1.0] | Max tokens | 16,384 |
| Meta models | [gpt-5-nano] | Meta temperatures | [0.0] |
| Novelty models | [gpt-5-nano] | Novelty temperatures | [0.0] |

Table 1: SHINKAEVOLVE hyperparameter configuration for the Circle Packing task.

## B.2 AIME MATH REASONING AGENTIC HARNESS

**Detailed Task Description.** For the agent scaffold design task, we evaluate SHINKAE-VOLVE on AIME 2024 mathematical reasoning problems, consisting of 30 challenging competition-level questions requiring sophisticated problem-solving strategies (AIM, 2024). We limit the maximum number of LLM queries per problem to 10 for computational and cost efficiency. Using gpt-4.1-nano as the base model, we evolve scaffold designs over 75 generations. Additionally and to combat stochasticity in LLM queries, we evaluated each candidate evaluated across three independent runs on the complete question set. After evolution, we evaluate the discovered scaffold designs on 2023 and 2025 AIME problems (AIM, 2023; 2025) to assess generalization as well as robustness to different base agent language models.

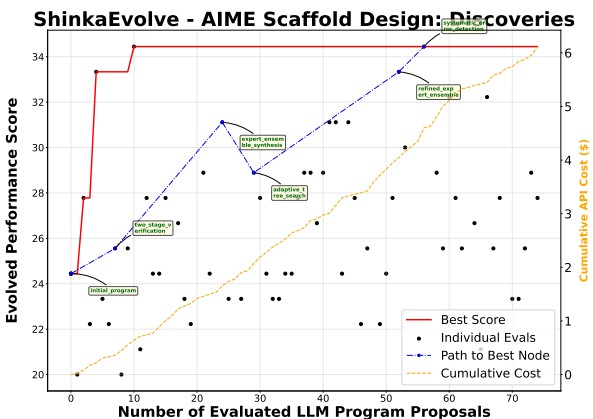

Figure 11: SHINKAEVOLVE's Discovery Trajectory for Math Agent Scaffold Design.

**SHINKAEVOLVE's Hyperparameter Configuration.**

| Parameter | Value | Parameter | Value |
|---|---|---|---|
| **Database configuration** | | | |
| Archive size | 40 | Elite selection ratio | 0.3 |
| Archive inspirations | 4 | Top-$k$ inspirations | 2 |
| Migration interval | 10 | Migration rate | 0.1 |
| Island elitism | true | Parent selection strategy | weighted |
| Parent selection $\lambda$ | 10.0 | Number of islands | 4 |
| **Evolution configuration** | | | |
| Patch types | [diff, full, cross] | Patch type probs | [0.6, 0.3, 0.1] |
| Generations | 75 | Max parallel jobs | 1 |
| Max patch resamples | 3 | Max patch attempts | 3 |
| Meta recommendation interval | 10 | Max meta recommendations | 5 |
| Embedding model | text-embedding-3-small | Max novelty attempts | 3 |
| Code embed sim threshold | 0.95 | Problem implementation | Python |
| LLM dynamic selection | null | Exploration coefficient | 0.0 |
| **LLM models** | | | |
| gemini-2.5-pro | ✓ | gemini-2.5-flash | ✗ |
| claude-sonnet-4 | ✓ | o4-mini | ✓ |
| gpt-5 | ✗ | gpt-5-nano | ✗ |
| gpt-4.1 | ✗ | gpt-4.1-mini | ✗ |
| **LLM settings** | | | |
| Temperatures | [0.0, 0.5, 1.0] | Max tokens | 16,384 |
| Meta models | [gpt-4.1] | Meta temperatures | [0.0] |
| Novelty models | [gpt-4.1] | Novelty temperatures | [0.0] |

Table 2: SHINKAEVOLVE Hyperparameter Configuration for the Math Reasoning Agentic Harness.

### B.3    ALE-BENCH PROBLEMS

**Detailed Task Description.**    The ALE-Bench benchmark (Imajuku et al., 2025) is a collection of heuristic programming problems previously used in competitive programming contests (AtCoder). We evaluate SHINKAEVOLVE on the LITE subset of problems, which consists of 10 problems. We follow the evaluation protocol of the benchmark and use the score calculated on the 50 public test cases as the fitness function following ALE-Agent (Imajuku et al., 2025). Afterwards, we submit the best solution to the private test set and report the score. Additionally, in Figure 6, we provide scores for evaluating the top-5 publicly scored solutions and taking their maximum score on the private test set. While this does not resemble the traditional competitive programming setting, it allows us to assess the generalization ability of the discovered solutions. The average solution score improves by a negligible amount from 1923.5 to 1927.0. Hence, we do not observe significant evidence for overfitting to the public test cases.

**SHINKAEVOLVE's Hyperparameter Configuration.**

| Parameter | Value | Parameter | Value |
|---|---|---|---|
| **Database configuration** | | | |
| Archive size | 50 | Elite selection ratio | 0.3 |
| Archive inspirations | 2 | Top-$k$ inspirations | 2 |
| Migration interval | 10 | Migration rate | 0.1 |
| Island elitism | true | Parent selection strategy | weighted |
| Parent selection $\lambda$ | 10.0 | Number of islands | 2 |
| **Evolution configuration** | | | |
| Patch types | [diff, full, cross] | Patch type probs | [0.6, 0.3, 0.1] |
| Generations | 50 | Max parallel jobs | 1 |
| Max patch resamples | 3 | Max patch attempts | 3 |
| Meta recommendation interval | 5 | Max meta recommendations | 5 |
| Embedding model | None | Max novelty attempts | None |
| Code embed sim threshold | None | Problem implementation | C++ |
| LLM dynamic selection | ucb1 | Exploration coefficient | 1.0 |
| **LLM models** | | | |
| gemini-2.5-pro | ✓ | gemini-2.5-flash | ✓ |
| claude-sonnet-4 | ✓ | o4-mini | ✓ |
| gpt-5 | ✓ | gpt-5-mini | ✓ |
| gpt-4.1 | ✗ | gpt-4.1-mini | ✗ |
| **LLM settings** | | | |
| Temperatures | [0.0, 0.5, 1.0] | Max tokens | 16,384 |
| Meta models | [gpt-5-mini] | Meta temperatures | [0.0] |
| Novelty models | None | Novelty temperatures | None |

Table 3: SHINKAEVOLVE Hyperparameter Configuration for the ALE-Bench Problems.

## B.4  MIXTURE-OF-EXPERTS LOAD BALANCING LOSS

| Hyperparameter | Small MoE (evolution) | Large MoE (evaluation) |
|---|---|---|
| **Model architecture** | | |
| Model parameters | 556M | 2.7B |
| Activated parameters | 82M | 404M |
| Number of experts ($N_E$) / active per token ($K$) | 64 / 8 | 64 / 8 |
| Hidden size | 512 | 1024 |
| Hidden size in each MoE expert | 384 | 768 |
| Number of hidden layers | 12 | 16 |
| Number of attention heads | 8 | 16 |
| Number of key–value heads | 8 | 8 |
| Head dimension | 128 | 128 |
| Attention bias | false | false |
| Attention dropout | 0.0 | 0.0 |
| Initializer range | 0.02 | 0.02 |
| RoPE $\theta$ | 1,000,000 | 1,000,000 |
| Tied word embeddings | true | true |
| Output router logits | true | true |
| Decoder sparse step | 1 | 1 |
| Router auxiliary loss coefficient ($\lambda$) | 0.01 | 0.001, 0.01, 0.1 |
| Computation dtype | bfloat16 | bfloat16 |
| **Training setup** | | |
| Optimizer | AdamW | AdamW |
| Learning rate | $1.0 \times 10^{-3}$ | $3.0 \times 10^{-4}$ |
| Weight decay | 0.1 | 0.1 |
| Adam parameters ($\beta_1, \beta_2, \epsilon$) | $(0.9, 0.95, 1 \times 10^{-8})$ | $(0.9, 0.95, 1 \times 10^{-8})$ |
| Learning rate scheduler | Cosine decay | Cosine decay |
| Warmup steps | 70 | 490 |
| Maximum sequence length | 1024 | 1024 |
| Global train batch size (sequences) | 1024 | 2048 |
| Tokens per training step | 1,048,576 | 2,097,152 |
| Maximum steps | 2000 | 14,000 |
| Total tokens | 2.10B | 29.36B |
| Dataset | fineweb | fineweb |

Table 4: MoE architectures and training setup.

**Detailed Task Description.**  The Mixture-of-Expert (MoE) architecture (Szymanski & Lemmon, 1993; Shazeer et al., 2017; Lepikhin et al., 2020; Fedus et al., 2022) has been a critical advancement, enabling scaling breakthroughs in large language model training. MoEs are currently ubiquitous amongst modern open and closed-source flagship models (Google AI Blog, 2024; Guo et al., 2025; Meta-AI, 2025; Yang et al., 2025; Team, 2025). The core principle behind the MoE design is to replace traditional large feed-forward residual blocks with ensembles of smaller modules (the "experts"), which can be efficiently sharded during training and only partially activated during inference (Fedus et al., 2022). Each expert is itself a small feed-forward network $E_{\ell,i}$ located within a larger ensemble of size $N_E$ at layer $\ell$. The router, a layer-specific linear classifier $h_\ell$, selects the top-$K$ most relevant experts for each token, computing only their outputs:

$$y_\ell(x) = \sum_{i=1}^{N_E} g_{\ell,i}(x) \, E_{\ell,i}(x), \quad g_{\ell,i}(x) = \begin{cases} \frac{e^{h_{\ell,i}(x)}}{\sum_{j \in \mathcal{T}_K(x)} e^{h_{\ell,j}(x)}}, & \text{if } i \in \mathcal{T}_K(x) \\ 0, & \text{otherwise} \end{cases} \quad (2)$$

where $\mathcal{T}_K(x)$ denotes the set of indices corresponding to the top-$K$ router logits $h_{\ell,i}(x)$. This sparsely activated design allows different experts to specialize in distinct problem domains, enabling greater efficiency, scalability, and adaptability in handling diverse prompts.

However, due to the non-differentiability of the top-$K$ expert selection operation, it is critical to provide the router with an auxiliary load balancing loss (LBL). The LBL prevents collapse toward uneven token distributions and under-specialized experts. Devising an effective load balancing loss

that simultaneously encourages efficiency and expert specialization, without hindering expressivity, remains an open design challenge that has driven much of the recent progress in MoEs (Shazeer et al., 2017; Fedus et al., 2022; Du et al., 2022; Zoph et al., 2022; Xue et al., 2024; Dai et al., 2024; Qiu et al., 2025; Muennighoff et al., 2024). Minor design variations have been shown to significantly affect both efficiency and specialization ability (Dai et al., 2024; Jiang et al., 2024; Team, 2024; Liu et al., 2024b; Qiu et al., 2025).

One of the most widely adopted designs is the "global-batch" LBL introduced by Shazeer et al. (2017), which underpins several state-of-the-art open models such as Qwen 3 (Yang et al., 2025). For a layer $\ell$ with $N_E$ experts, it is defined as:

$$L_{LB} = N_E \cdot \frac{1}{L} \sum_{\ell=1}^{L} \sum_{i=1}^{N_E} f_{\ell,i} \cdot P_{\ell,i}, \tag{3}$$

where

$$f_{\ell,i} = \frac{\text{Tokens routed to expert } i}{\text{Total tokens in layer } \ell}, \qquad P_{\ell,i} = \frac{\sum_x h_{\ell,i}(x)}{\sum_{x,j} h_{\ell,j}(x)}.$$

This formulation encourages token usage across experts to align with the router's average soft assignment probabilities.

We evaluate SHINKAEVOLVE by pretraining a MoE model with 556M parameters, $N_E = 64$ experts of which only $K = 8$ are active for each token, corresponding to 82M sparsely activated parameters per forward pass (excluding embeddings). Training is performed on 2B tokens from fineweb (Penedo et al., 2024). For each program, we define a fitness function consisting of the cross-entropy (CE) loss together with an LBL term weighted by $\lambda = 0.01$. To additionally measure load imbalance, we track the L1 deviation from a uniform distribution of token allocations:

$$L_{\text{imb}} = \frac{1}{2} \sum_{i=1}^{N_E} \left| f_{\ell,i} - \tfrac{1}{N_E} \right|, \tag{4}$$

with lower values indicating more even load distribution. This grounding provides SHINKAEVOLVE a two-fold search objective: minimize CE while improving load balance. To avoid local noise affecting the cross-entropy calculations, we average it over the last 10M tokens. The final fitness score used during evolution is then the negated sum of the two:

$$r = -(L_{\text{CE}} + L_{\text{imb}}). \tag{5}$$

Given the expense of pretraining, we run SHINKAEVOLVE for only 30 iterations, focusing on `gpt-4.1`, `gemini-2.5-pro`, and `claude-sonnet-4`. To evaluate generality, we scale to a larger 2.7B-parameter MoE of which 404M active (excluding embeddings), trained on slightly under 30B fineweb tokens, and compare across three LBL coefficients $\lambda \in \{0.001, 0.01, 0.1\}$. We used `AdamW` (Loshchilov & Hutter, 2017) as the optimizer with cosine decay, and linear warmup. As common practice in modern training regimes, we used rotary positional embeddings (Su et al., 2024), SwiGLU MLPs (Shazeer, 2020), and half-precision bfloat16 to efficiently keep our model's weights on device. For the small model used during SHINKAEVOLVE's evolution, we use a batch size of slightly over $1M$ tokens, for 2K steps. For the larger MoE used double the batch size and seven times the total number of steps. After training, we benchmark against the global-batch LBL baseline in terms of perplexity (Figure 7, left) and downstream performance across seven standard evaluations: CommonSense QA (Talmor et al., 2018), HellaSwag (Zellers et al., 2019), OpenBook QA (Mihaylov et al., 2018), PIQA (Bisk et al., 2020), SIQA (Sap et al., 2019), WinoGrande (Sakaguchi et al., 2021), and ARC (Clark et al., 2018), truncating the number of questions to 1000 for large benchmarks as done by (Penedo et al., 2024).

As described in Section 4 and detailed in Appendix D, SHINKAEVOLVE discovers a new twist on the global-batch LBL from Equation 3, which was used for seeding evolutionary search. SHINKAEVOLVE discovers an augmentation of this loss with an additional regularization term to target under-specialized experts. As defined in Equation 3, let $f_{\ell,i}$ and $P_{\ell,i}$ denote the selection frequency and average router probabilities for expert $i$ in layer $\ell$. Furthermore, define $s(P_\ell) = 0.5 + \left(1 - \frac{H(P_\ell)}{\log N_E}\right)$

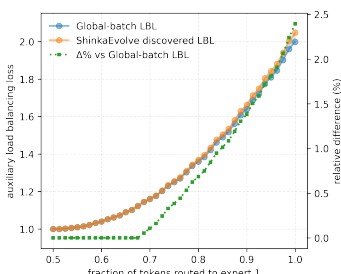

Figure 12: LBL loss comparison.

as a normalized complement of the routing entropy, and $\tau = 0.064/N_E$ as a minimum usage threshold. The final discovered LBL is:

$$L_{\text{LBL}} = \underbrace{N_E \cdot \frac{1}{L} \sum_{\ell=1}^{L} \sum_{i=1}^{N_E} f_{\ell,i} \, P_{\ell,i}}_{\text{Global-batch LBL}} + \underbrace{\frac{0.1}{L} \sum_{\ell=1}^{L} s(P_\ell) \sum_{i=1}^{N_E} \max(0, \tau - f_{\ell,i})}_{\textsc{ShinkaEvolve new regularization}} . \tag{6}$$

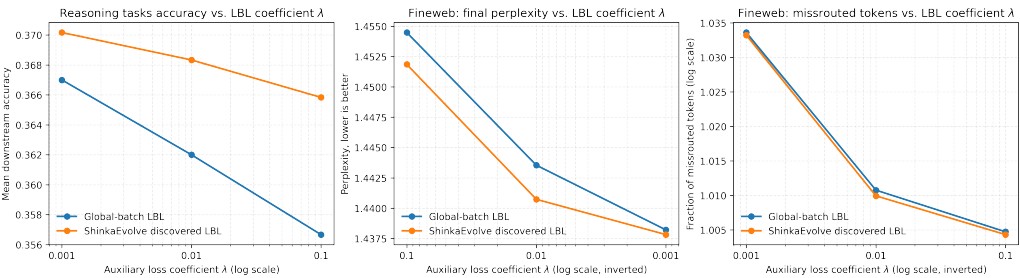

Figure 13: Mixture-of-Experts LBL design additional results.

| Task | Global LBL | ShinkaEvolve |
|---|---|---|
| HellaSwag | 0.391 | 0.379 |
| CQA | 0.192 | 0.192 |
| PIQA | 0.688 | 0.684 |
| Winogrande | 0.524 | 0.549 |
| ARC | 0.207 | 0.214 |
| OpenBookQA | 0.170 | 0.192 |
| Mean | 0.362 | **0.368** |

| Loss Type | $\lambda$ | CE Loss | Accuracy |
|---|---|---|---|
| Global LBL | 0.1 | 0.375 | 0.357 |
| ShinkaEvolve | | 0.373 | **0.366** |
| Global LBL | 0.01 | 0.367 | 0.362 |
| ShinkaEvolve | | 0.365 | **0.368** |
| Global LBL | 0.001 | 0.363 | 0.367 |
| ShinkaEvolve | | 0.363 | **0.37** |

Figure 14: Comparison of load balancing loss variants across downstream tasks with $\lambda = 0.01$ (left) and across LBL coefficients (right).

In addition to the results from Section 4, in Figure 13, we provide additional results comparing the global-batch LBL and SHINKAEVOLVE's discovered LBL. In particular, we report the average task performance, final perplexity, and the fraction of missrouted tokens, as a function of the LBL coefficient $\lambda$ used for training the MoEs. Consistent with our previous analysis, SHINKAEVOLVE's LBL appears to improve from the original LBL across both axes. Moreover, in the tables shown in Figure 14, we provide tables with details for the downstream task performance across our over considered benchmarks, as summarized in the center subplot of Figure 13. However, we also note that the architecture used for evolving and testing the employed LBL was quite similar, and the training budget was still limited. However, the consistent generalization results across training budgets and coefficients $\lambda$ provide an optimistic outlook for future extensions to much longer training regimes, where even small efficiency gains could scale to significant cost savings.

**SHINKAEVOLVE's Hyperparameter Configuration.**

| Parameter | Value | Parameter | Value |
|---|---|---|---|
| **Database configuration** | | | |
| Archive size | 20 | Elite selection ratio | 0.3 |
| Archive inspirations | 4 | Top-$k$ inspirations | 2 |
| Migration interval | 10 | Migration rate | 0.1 |
| Island elitism | true | Parent selection strategy | weighted |
| Parent selection $\lambda$ | 10.0 | Number of islands | 2 |
| **Evolution configuration** | | | |
| Patch types | [diff, full] | Patch type probs | [0.5, 0.5] |
| Generations | 20 | Max parallel jobs | 1 |
| Max patch resamples | 10 | Max patch attempts | 10 |
| Meta recommendation interval | 10 | Max meta recommendations | 5 |
| Embedding model | text-embedding-3-small | Max novelty attempts | 3 |
| Code embed sim threshold | 0.95 | Problem implementation | Python |
| LLM dynamic selection | ucb1 | Exploration coefficient | 1.0 |
| **LLM models** | | | |
| gemini-2.5-pro | ✓ | gemini-2.5-flash | ✗ |
| claude-sonnet-4 | ✓ | o4-mini | ✗ |
| gpt-5 | ✗ | gpt-5-nano | ✗ |
| gpt-4.1 | ✓ | gpt-4.1-mini | ✗ |
| **LLM settings** | | | |
| Temperatures | [0.0, 0.5, 1.0] | Max tokens | 16,384 |
| Meta models | [gpt-4.1] | Meta temperatures | [0.0] |
| Novelty models | [gpt-4.1] | Novelty temperatures | [0.0] |

Table 5: SHINKAEVOLVE Hyperparameter Configuration for the MoE LBL Discovery.

# C ADDITIONAL RESULTS

## C.1 CIRCLE PACKING: ROBUSTNESS ACROSS 3 INDEPENDENT RUNS

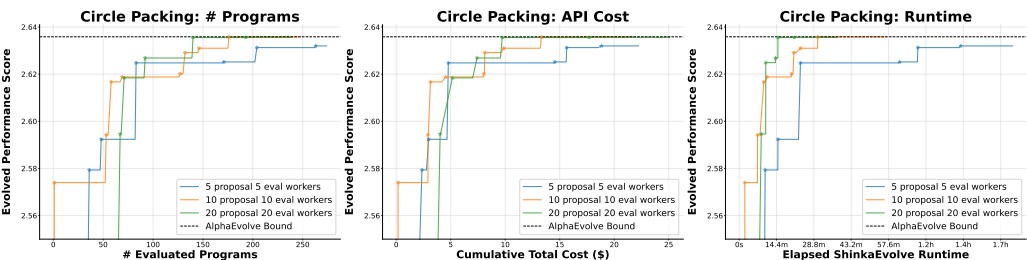

Figure 15: Circle Packing results across 3 independent runs. Two out of three runs discover solutions that outperform or perform on par with AlphaEvolve, demonstrating the reliability and effectiveness of our approach. We also compare different settings of asynchronous evaluation and program proposal workers for ShinkaEvolve.

## C.2 CIRCLE PACKING: API COST BREAKDOWN

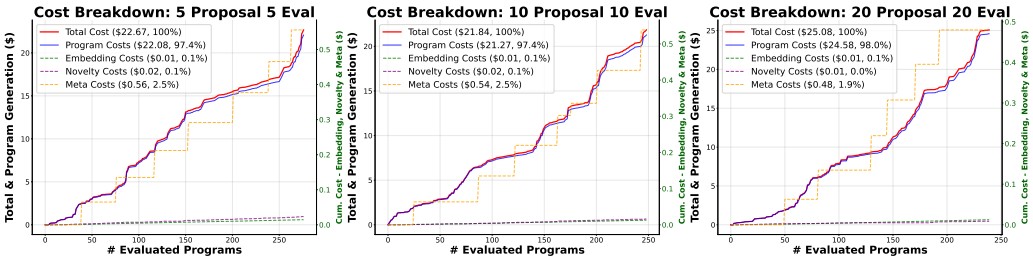

Figure 16: API cost breakdown for Circle Packing across different parallelization configurations. Approximately 97% of the budget is used on program generation, while embedding, novelty checking, and meta-recommendation generation take up the remaining 3%.

## C.3 CIRCLE PACKING: ASYNCHRONOUS THROUGHPUT SCALING

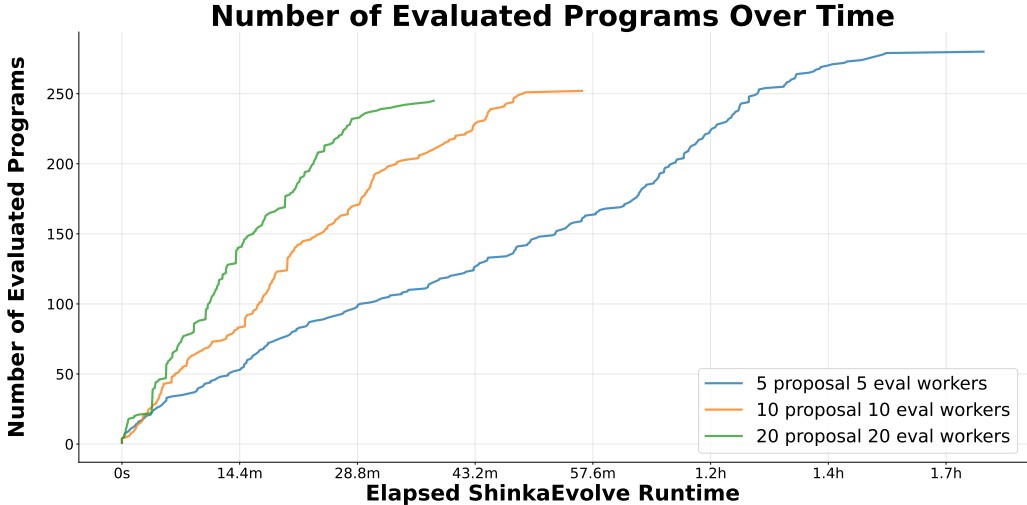

Figure 17: Throughput scaling for Circle Packing with different numbers of proposal and evaluation workers. The speedup is almost linear for fast-to-evaluate problems like Circle Packing, demonstrating efficient parallelization.

## C.4  CIRCLE PACKING: ROBUSTNESS ACROSS CODE EMBEDDING THRESHOLDS

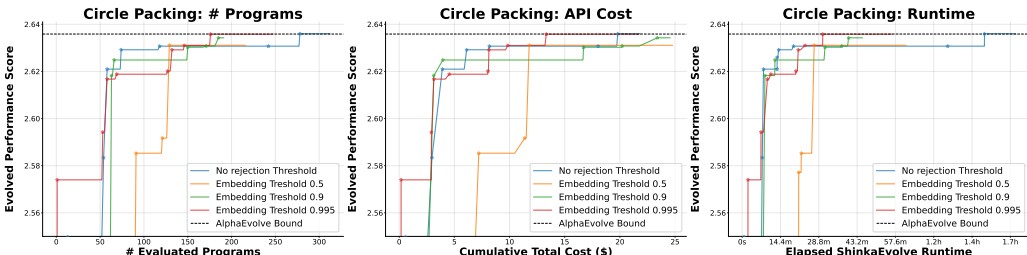

Figure 18: Performance comparison of different code embedding thresholds for Circle Packing. We compare thresholds of 1.0 (no rejection), 0.995, 0.9, and 0.5 (heavily rejecting similar programs). The larger values perform better, indicating that conservatively rejecting similar programs performs well for this domain.

## C.5  CIRCLE PACKING: ROBUSTNESS ACROSS LLM PRIORITIZATION APPROACHES

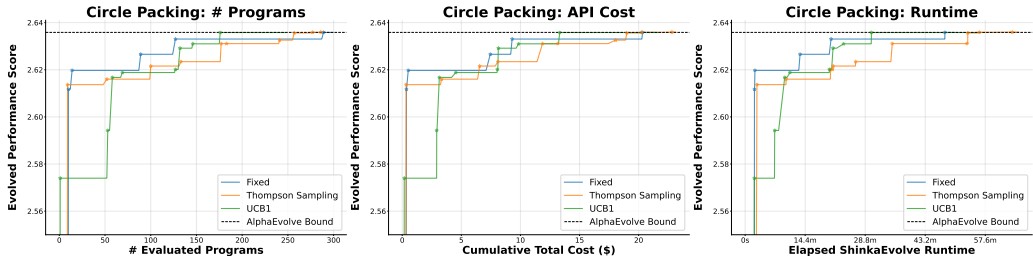

Figure 19: Performance comparison of different LLM prioritization approaches for Circle Packing. We compare UCB1 (our proposed approach), Thompson sampling, fixed (uniformly sampling models) strategies. While all approaches show similar asymptotic performance, UCB1 performs the most sample efficient.

## D  SHINKAEVOLVE DISCOVERED SOLUTIONS

### D.1  CIRCLE PACKING PROBLEM

```python
# EVOLVE-BLOCK-START
import numpy as np
from scipy.optimize import minimize, Bounds

def construct_packing():
    """
    Constructs an arrangement of 26 circles by combining a meta-heuristic
    search with a powerful SLSQP optimizer for refinement.
    """
    n = 26

    # --- Helper functions for the optimizer ---
    def objective_func(x):
        """The function to be minimized: the negative sum of radii."""
        return -np.sum(x[:n])

    def constraints_func(x):
        """
        Computes constraint violations. For SLSQP, each value must be >= 0.
        """
        radii = x[:n]
        centers = x[n:].reshape((n, 2))

        containment = np.concatenate(
            [
                centers[:, 0] - radii,
                centers[:, 1] - radii,
                1 - centers[:, 0] - radii,
                1 - centers[:, 1] - radii,
            ]
        )

        overlap = []
        for i in range(n):
            for j in range(i + 1, n):
                dist = np.linalg.norm(centers[i] - centers[j])
                overlap.append(dist - (radii[i] + radii[j]))

        return np.concatenate([containment, np.array(overlap)])

    def _compute_initial_radii(centers):
        """
        Computes a valid set of initial radii for a given set of centers
        to create a feasible starting point (x0) for the optimizer.
        """
        radii = np.min(
            [centers[:, 0], centers[:, 1], 1 - centers[:, 0], 1 - centers[:, 1]], axis=0
        )

        for _ in range(100):
            improved = False
            for i in range(n):
                for j in range(i + 1, n):
                    dist = np.linalg.norm(centers[i] - centers[j])
                    if radii[i] + radii[j] > dist:
                        excess = (radii[i] + radii[j] - dist) * 0.501
                        total_r = radii[i] + radii[j]
                        if total_r > 1e-9:
                            radii[i] -= excess * (radii[i] / total_r)
                            radii[j] -= excess * (radii[j] / total_r)
                            improved = True
            if not improved:
                break
        return np.maximum(radii, 1e-6)

    # --- 1. Generate a single high-quality initial guess ---
    centers_init = np.zeros((n, 2))
    inset = 0.06
    centers_init[0:4] = [
        [inset, inset],
        [1 - inset, inset],
        [inset, 1 - inset],
        [1 - inset, 1 - inset],
    ]
    centers_init[4:8] = [[0.5, inset], [0.5, 1 - inset], [inset, 0.5], [1 - inset, 0.5]]
    centers_init[8] = [0.5, 0.5]

    golden_angle = np.pi * (3 - np.sqrt(5))
    cx, cy = 0.5, 0.5
    inner_r, outer_r = 0.23, 0.48
    inner_idx, outer_idx = np.arange(9, 15), np.arange(15, 26)

    for i, idx in enumerate(inner_idx):
        angle = i * golden_angle
        centers_init[idx] = [cx + inner_r * np.cos(angle), cy + inner_r * np.sin(angle)]
    for i, idx in enumerate(outer_idx):
        angle = i * golden_angle * 1.003
```

```python
        centers_init[idx] = [cx + outer_r * np.cos(angle), cy + outer_r * np.sin(angle)]

centers_init += np.random.uniform(
    -0.01, 0.01, size=(n, 2)
)  # Increased initial jitter
centers_init = np.clip(centers_init, 0.01, 0.99)

# --- 2. Define bounds and constraints for the solver ---
bounds = Bounds([0.0] * n + [0.0] * (2 * n), [0.5] * n + [1.0] * (2 * n))
constraints = {"type": "ineq", "fun": constraints_func}

# --- 3. Initial baseline optimization ---
radii_init = _compute_initial_radii(centers_init)
x0 = np.concatenate([radii_init, centers_init.flatten()])

result = minimize(
    objective_func,
    x0,
    method="SLSQP",
    bounds=bounds,
    constraints=constraints,
    options={"maxiter": 600, "ftol": 1e-8, "disp": False},
)  # Increased initial maxiter

# Initialize current and best solutions for SA
best_x = result.x.copy()
current_x = result.x.copy()
best_score = -result.fun
current_score = -result.fun

# --- 4. Simulated Annealing loop: Perturb and refine with acceptance criterion ---
sa_iterations = 250  # Significantly increased iterations for SA
temperature = 0.05  # Initial temperature for SA
initial_temperature = temperature  # Preserve for potential reheating
cooling_rate = 0.995  # Slower cooling rate for broader search
perturb_step = 0.04  # Initial step size for perturbations
initial_perturb_step = perturb_step  # Preserve for potential reheating
step_decay = 0.999  # Decay rate for step size
last_improve = 0  # Iteration of last best improvement
stagnation_limit = sa_iterations // 4  # Iterations before triggering reheating

for iter_idx in range(sa_iterations):
    candidate_centers = (
        current_x[n:].reshape((n, 2)).copy()
    )  # Start from current state

    # Select a move type: 70% local, 30% global ring rotation
    if np.random.rand() < 0.7:
        # Local move: perturb a few circles
        num_to_move = np.random.randint(2, 6)
        indices = np.random.choice(n, num_to_move, replace=False)
        candidate_centers[indices] += np.random.normal(
            0, perturb_step, size=(num_to_move, 2)
        )
    else:
        # Global move: rotate one of the rings
        idx_to_rotate = inner_idx if np.random.rand() < 0.5 else outer_idx
        center_point = candidate_centers[8]  # Center of the overall pattern
        angle = np.random.normal(
            0, 0.15
        )  # Angular perturbation (can be fixed or scaled)
        rel_pos = candidate_centers[idx_to_rotate] - center_point
        cos_a, sin_a = np.cos(angle), np.sin(angle)
        rotated = np.column_stack(
            [
                cos_a * rel_pos[:, 0] - sin_a * rel_pos[:, 1],
                sin_a * rel_pos[:, 0] + cos_a * rel_pos[:, 1],
            ]
        )
        candidate_centers[idx_to_rotate] = center_point + rotated

    candidate_centers = np.clip(
        candidate_centers, 0.01, 0.99
    )  # Clip to stay within bounds

    # Create a new starting point and run a shorter refinement optimization
    x0_candidate = np.concatenate(
        [_compute_initial_radii(candidate_centers), candidate_centers.flatten()]
    )
    refine_result = minimize(
        objective_func,
        x0_candidate,
        method="SLSQP",
        bounds=bounds,
        constraints=constraints,
        options={"maxiter": 150, "ftol": 1e-6, "disp": False},
    )  # Reduced maxiter, looser ftol

    new_score = -refine_result.fun

    # Simulated Annealing Acceptance Criterion
    # Accept if better, or with probability if worse (based on temperature)
```

```python
        if new_score > current_score or (
            temperature > 1e-7
            and np.random.rand() < np.exp((new_score - current_score) / temperature)
        ):
            current_score = new_score
            current_x = refine_result.x.copy()  # Update current state
            if new_score > best_score:
                best_score = new_score
                best_x = refine_result.x.copy()  # Update global best
                last_improve = iter_idx  # Reset stagnation counter on improvement
        # If not accepted, current_x remains unchanged for the next iteration (implicit)

        # Cool down temperature and decay perturbation step size
        temperature *= cooling_rate
        perturb_step *= step_decay
        if temperature < 1e-7:
            temperature = 1e-7  # Prevent division by zero
        if perturb_step < 1e-5:
            perturb_step = 1e-5  # Prevent step from becoming too small
        # Reheat if stagnated beyond stagnation_limit
        if iter_idx - last_improve > stagnation_limit:
            temperature = initial_temperature
            perturb_step = initial_perturb_step
            last_improve = iter_idx

    # --- 5. Final Polishing Run on the best found solution ---
    final_result = minimize(
        objective_func,
        best_x,
        method="SLSQP",
        bounds=bounds,
        constraints=constraints,
        options={"maxiter": 1000, "ftol": 1e-9, "disp": False},
    )  # Increased maxiter for final polish

    # Check if the final polishing improved the best_x from SA
    if -final_result.fun > best_score:
        best_x = final_result.x.copy()  # Make sure to copy

    # --- 6. Unpack and return the best result ---
    final_radii = best_x[:n]
    final_centers = best_x[n:].reshape((n, 2))
    return final_centers, final_radii

def compute_max_radii(centers):
    """
    This function is retained for structural compatibility with the evaluation
    framework but is not used by the new 'construct_packing' logic.
    It computes maximum radii for a fixed set of centers.
    """
    n = centers.shape[0]
    radii = np.empty(n)
    for i in range(n):
        x, y = centers[i]
        radii[i] = min(x, y, 1 - x, 1 - y)

    for _ in range(60):
        improved = False
        for i in range(n):
            for j in range(i + 1, n):
                dist = np.linalg.norm(centers[i] - centers[j])
                if radii[i] + radii[j] > dist:
                    excess = (radii[i] + radii[j] - dist) * 0.5
                    total = radii[i] + radii[j]
                    if total > 0:
                        reduce_i = excess * (radii[i] / total)
                        reduce_j = excess * (radii[j] / total)
                        radii[i] = max(0.001, radii[i] - reduce_i)
                        radii[j] = max(0.001, radii[j] - reduce_j)
                        improved = True
        if not improved:
            break
    return radii

# EVOLVE-BLOCK-END

# This part remains fixed (not evolved)
def run_packing():
    """Run the circle packing constructor for n=26"""
    np.random.seed(7)
    centers, radii = construct_packing()
    # Calculate the sum of radii
    sum_radii = np.sum(radii)
    return centers, radii, sum_radii

centers, radii, sum_radii = run_packing()
```

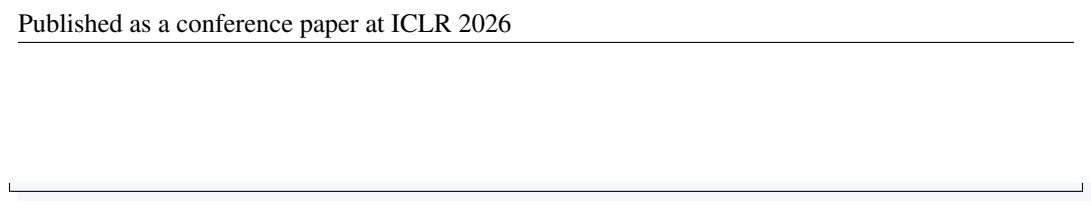

Listing 2: SHINKAEVOLVE Discovered Circle Packing Solution.

## D.2 AIME MATH REASONING AGENTIC HARNESS

```python
"""Agent design evaluation on math tasks."""

import re
from typing import Callable, List, Optional, Tuple, Dict
from collections import Counter, defaultdict
from math_eval import agent_evaluation

# EVOLVE-BLOCK-START
import re
from collections import Counter

class Agent:
    def __init__(
        self,
        query_llm: Callable,
        temperature=0.0,
    ):
        self.query_llm = query_llm
        self.output_format_instructions = "On the final line output only the digits of the answer (0-999). Provide your final answer enclosed in a LaTeX \\boxed{{...}} command."

        # Parameters
        self.generation_temperature = 0.7
        self.review_temperature = 0.1
        self.synthesis_temperature = 0.0

        # Use 3 experts to stay within a 10-call limit (3 gen + 3 review + 1 synth = 7 calls)
        self.num_experts = 3
        self.expert_personas = [
            "You are a meticulous and cautious mathematician. Your guiding principle is 'slow and steady wins the race'. You solve problems by breaking them down into the smallest possible steps based on fundamental principles. You avoid leaps of logic and verify each step before proceeding.",
            "You are a brilliant and intuitive mathematician, known for finding elegant, non-obvious solutions. You look for symmetries, invariants, or a change of perspective that radically simplifies the problem. You trust your insights but explain them clearly.",
            "You are a mathematician with a strong background in computer science. You approach problems by trying to frame them algorithmically. You think in terms of states, transitions, and recurrence relations, and you analyze the behavior of these systems to find the solution.",
        ]

    def _extract_answer(self, text: str) -> Optional[str]:
        """Extracts the final answer from a \\boxed{} environment."""
        if not text:
            return None
        matches = re.findall(r"\\boxed\{(\d{1,3})\}", text)
        if matches:
            return matches[-1]
        return None

    def forward(self, problem: str) -> tuple[str, float]:
        """
        Solves a problem using a multi-persona ensemble with peer review and synthesis.
        """
        total_cost = 0.0

        # === STAGE 1: Generate Diverse Solutions with Expert Personas ===
        solutions = []
        for i in range(self.num_experts):
            persona = self.expert_personas[i % len(self.expert_personas)]
            prompt = f"Solve the following AIME problem by thinking step-by-step. {self.output_format_instructions}\n\nPROBLEM:\n{problem}\n\nSOLUTION:"
            try:
                response, cost = self.query_llm(
                    prompt=prompt,
                    system=persona,
                    temperature=self.generation_temperature,
                )
                solutions.append(response)
                total_cost += cost
            except Exception:
                # If a query fails, we proceed with fewer solutions.
                solutions.append(f"Expert {i + 1} failed to generate a solution.")

        # === STAGE 2: Independent Peer Review & Self-Correction ===
        critiques = []
        reviewer_system_prompt = "You are a skeptical peer reviewer examining a proposed solution to an AIME problem. Your task is to be extremely critical. Do not accept any statement at face value. Re-read the original problem carefully. Check calculations. Scrutinize the logical flow. **Pattern Verification:** If the solution relies on a pattern, you MUST test it on several new examples. If you find an error, clearly explain the flaw and provide a corrected line of reasoning and a final corrected answer. If the solution is completely sound, state that and re-state the final answer."
        for sol in solutions:
            prompt = f"Original Problem:\n{problem}\n\nProposed Solution to Review:\n{sol}\n\nYour Critical Review and Corrected Solution:"
            try:
                review, cost = self.query_llm(
                    prompt=prompt,
                    system=reviewer_system_prompt,
```

```
                    temperature=self.review_temperature,
                )
                critiques.append(review)
                total_cost += cost
            except Exception:
                critiques.append("Reviewer failed to provide a critique.")

        # === STAGE 3: Synthesize Final Answer ===
        synthesis_prompt_parts = [
            f"You are the Editor-in-Chief of a prestigious mathematics journal, responsible for publishing the
    final, canonical solution to this AIME problem. You have received {self.num_experts} independent
    attempts and their corresponding critical reviews. Your task is to produce the definitive solution.\n\
    nProblem:\n{problem}"
        ]
        for i, (sol, crit) in enumerate(zip(solutions, critiques)):
            synthesis_prompt_parts.append(
                f"\n--- ATTEMPT {i + 1} ---\nSolution: {sol}\nCritique: {crit}\n---"
            )

        synthesis_prompt_parts.append(
            f"\nSYNTHESIS AND FINAL JUDGEMENT:\n1. First, briefly state the final numerical answer proposed by
     each of the reviewed attempts.\n2. Based on the critiques, determine which approach is the most
    reliable, or if all are flawed. Explain your reasoning.\n3. Construct the final, clear, step-by-step,
    correct solution. Leverage insights from the valid parts of the attempts and correct any identified
    errors. {self.output_format_instructions}"
        )

        synthesizer_prompt = "\n".join(synthesis_prompt_parts)
        synthesizer_system_prompt = "You are a master mathematician and editor, synthesizing multiple reviewed
     solutions into one canonical, correct answer."

        final_response = ""
        try:
            final_response, cost = self.query_llm(
                prompt=synthesizer_prompt,
                system=synthesizer_system_prompt,
                temperature=self.synthesis_temperature,
            )
            total_cost += cost
        except Exception:
            pass  # Fallback logic will handle this.

        # === Fallback Logic ===
        if self._extract_answer(final_response) is None:
            # First, trust the reviewed answers
            reviewed_answers = [self._extract_answer(c) for c in critiques]
            valid_reviewed_answers = [
                ans for ans in reviewed_answers if ans is not None
            ]

            if valid_reviewed_answers:
                most_common_answer = Counter(valid_reviewed_answers).most_common(1)[0][
                    0
                ]
                final_response += f"\n\n[Fallback to Majority Vote on Reviewed Solutions]\n\\boxed{{{
    most_common_answer}}}"
            else:
                # If reviews didn't produce answers, check original solutions
                original_answers = [self._extract_answer(s) for s in solutions]
                valid_original_answers = [
                    ans for ans in original_answers if ans is not None
                ]
                if valid_original_answers:
                    most_common_answer = Counter(valid_original_answers).most_common(1)[
                        0
                    ][0]
                    final_response += f"\n\n[Fallback to Majority Vote on Original Solutions]\n\\boxed{{{
    most_common_answer}}}"
                else:
                    # Ultimate fallback
                    final_response += "\n\n[Fallback] Could not determine a final answer from any stage.\n\\
    boxed{000}"

        return final_response, total_cost

# EVOLVE-BLOCK-END

def run_experiment(**kwargs):
    from utils import query_llm, create_call_limited_query_llm
    from functools import partial

    # Create base query_llm function
    base_query_llm = partial(query_llm, model_name=kwargs["model_name"])

    # Wrap it with call limiting (max 10 calls per forward pass)
    limited_query_llm = create_call_limited_query_llm(
        base_query_llm,
        max_calls=kwargs["max_calls"],
    )
```

```
accuracy, cost_total, processed, num_llm_calls, df = agent_evaluation(
    Agent, limited_query_llm, year=kwargs["year"]
)
return accuracy, cost_total, processed, num_llm_calls, df
```

Listing 3: SHINKAEVOLVE Discovered AIME Agent Scaffold Design.

## D.3 ALE-BENCH PROBLEMS

### D.3.1 ALE-BENCH LITE TASK: AHC039

```cpp
// EVOLVE-BLOCK-START
#include <iostream>
#include <vector>
#include <algorithm>
#include <chrono>
#include <random>
#include <set>
#include <unordered_set>
#include <cmath>
#include <iomanip>
#include <numeric> // For std::iota
#include <string>
#include <map>

// === MACROS AND CONSTANTS ===
const int MAX_COORD_VAL = 100000;
const int MAX_VERTICES = 1000;
const int MAX_PERIMETER = 400000;
const double TIME_LIMIT_SECONDS_SAFETY_MARGIN = 0.1; // Increased safety margin
double ACTUAL_TIME_LIMIT_SECONDS = 2.0;

// === RANDOM NUMBER GENERATION ===
struct XorShift {
    uint64_t x;
    XorShift() : x(std::chrono::steady_clock::now().time_since_epoch().count() ^ ((uint64_t)std::random_device
      ()() << 32) ^ std::random_device()()) {}
    uint64_t next() {
        x ^= x << 13;
        x ^= x >> 7;
        x ^= x << 17;
        return x;
    }
    int next_int(int n) { if (n <= 0) return 0; return next() % n; }
    int next_int(int a, int b) { if (a > b) return a; return a + next_int(b - a + 1); }
    double next_double() { return next() / (double)UINT64_MAX; }
};
XorShift rng;

// === TIMER ===
struct Timer {
    std::chrono::steady_clock::time_point start_time;
    Timer() { reset(); }
    void reset() { start_time = std::chrono::steady_clock::now(); }
    double elapsed() const {
        auto now = std::chrono::steady_clock::now();
        return std::chrono::duration_cast<std::chrono::duration<double>>(now - start_time).count();
    }
};
Timer global_timer;

// === GEOMETRIC STRUCTURES ===
struct Point {
    int x, y;
    bool operator<(const Point& other) const {
        if (x != other.x) return x < other.x;
        return y < other.y;
    }
    bool operator==(const Point& other) const {
        return x == other.x && y == other.y;
    }
    Point operator-(const Point& other) const {
        return {x - other.x, y - other.y};
    }
};

struct PointHash {
    std::size_t operator()(const Point& p) const {
        auto h1 = std::hash<int>{}(p.x);
        auto h2 = std::hash<int>{}(p.y);
        // Combining hashes: simple XOR might not be best, but often good enough.
        // For Point, a common way is boost::hash_combine.
        // h1 ^ (h2 << 1) is a common way that's okay.
        return h1 ^ (h2 << 1);
    }
};

long long cross_product(Point a, Point b) {
    return (long long)a.x * b.y - (long long)a.y * b.x;
}

struct Fish {
    Point p;
    int type; // 1 for mackerel, -1 for sardine
};
std::vector<Fish> all_fish_structs;

// === KD-TREE ===
```

```
struct KDNode {
    Point pt;
    int axis;
    KDNode *left = nullptr, *right = nullptr;
    int fish_struct_idx = -1;
    // Subtree bounding box
    int min_x, max_x, min_y, max_y;
    // Subtree counts
    int m_cnt = 0, s_cnt = 0;
};
KDNode* fish_kdtree_root = nullptr;

KDNode* build_kdtree(std::vector<int>& point_indices, int l, int r, int axis) {
    if (l > r) return nullptr;
    int mid = l + (r - l) / 2;

    std::nth_element(point_indices.begin() + l, point_indices.begin() + mid, point_indices.begin() + r + 1,
        [&](int a_idx, int b_idx) {
            const Point& pa = all_fish_structs[a_idx].p;
            const Point& pb = all_fish_structs[b_idx].p;
            if (axis == 0) return pa.x < pb.x;
            return pa.y < pb.y;
        });

    KDNode* node = new KDNode();
    node->fish_struct_idx = point_indices[mid];
    node->pt = all_fish_structs[node->fish_struct_idx].p;
    node->axis = axis;

    // Recurse
    node->left = build_kdtree(point_indices, l, mid - 1, 1 - axis);
    node->right = build_kdtree(point_indices, mid + 1, r, 1 - axis);

    // Initialize subtree bbox to this point
    node->min_x = node->max_x = node->pt.x;
    node->min_y = node->max_y = node->pt.y;
    // Initialize counts with this node's fish
    if (all_fish_structs[node->fish_struct_idx].type == 1) node->m_cnt = 1;
    else node->s_cnt = 1;

    // Merge children
    if (node->left) {
        node->min_x = std::min(node->min_x, node->left->min_x);
        node->max_x = std::max(node->max_x, node->left->max_x);
        node->min_y = std::min(node->min_y, node->left->min_y);
        node->max_y = std::max(node->max_y, node->left->max_y);
        node->m_cnt += node->left->m_cnt;
        node->s_cnt += node->left->s_cnt;
    }
    if (node->right) {
        node->min_x = std::min(node->min_x, node->right->min_x);
        node->max_x = std::max(node->max_x, node->right->max_x);
        node->min_y = std::min(node->min_y, node->right->min_y);
        node->max_y = std::max(node->max_y, node->right->max_y);
        node->m_cnt += node->right->m_cnt;
        node->s_cnt += node->right->s_cnt;
    }
    return node;
}

void delete_kdtree(KDNode* node) { // Recursively delete KD-tree nodes
    if (!node) return;
    delete_kdtree(node->left);
    delete_kdtree(node->right);
    delete node;
}

// === POLYGON UTILITIES ===
long long calculate_perimeter(const std::vector<Point>& poly) {
    if (poly.size() < 2) return 0;
    long long perimeter = 0;
    for (size_t i = 0; i < poly.size(); ++i) {
        const Point& p1 = poly[i];
        const Point& p2 = poly[(i + 1) % poly.size()];
        perimeter += std::abs(p1.x - p2.x) + std::abs(p1.y - p2.y);
    }
    return perimeter;
}

bool is_on_segment(Point p, Point seg_a, Point seg_b) {
    if (cross_product(seg_b - seg_a, p - seg_a) != 0) return false; // Not collinear
    return std::min(seg_a.x, seg_b.x) <= p.x && p.x <= std::max(seg_a.x, seg_b.x) &&
            std::min(seg_a.y, seg_b.y) <= p.y && p.y <= std::max(seg_a.y, seg_b.y);
}

bool is_inside_polygon_wn(Point p, const std::vector<Point>& polygon) {
    int n = polygon.size();
    if (n < 3) return false;

    // Check if on boundary first
```

```cpp
    for (int i = 0; i < n; ++i) {
        if (is_on_segment(p, polygon[i], polygon[(i + 1) % n])) return true;
    }

    int wn = 0; // Winding number
    for (int i = 0; i < n; ++i) {
        Point p1 = polygon[i];
        Point p2 = polygon[(i + 1) % n];
        if (p1.y <= p.y) { // Start y <= P.y
            if (p2.y > p.y && cross_product(p2 - p1, p - p1) > 0) { // An upward crossing, P is left of edge
                wn++;
            }
        } else { // Start y > P.y
            if (p2.y <= p.y && cross_product(p2 - p1, p - p1) < 0) { // A downward crossing, P is right of
  edge
                wn--;
            }
        }
    }
    return wn != 0; // wn != 0 means inside; wn == 0 means outside.
}

// Calculate score from scratch by checking all fish
long long point_segment_dist_sq_ortho(Point p, Point a, Point b) {
    long long dx, dy;
    if (a.x == b.x) { // Vertical segment
        dx = p.x - a.x;
        if (p.y < std::min(a.y, b.y)) {
            dy = p.y - std::min(a.y, b.y);
        } else if (p.y > std::max(a.y, b.y)) {
            dy = p.y - std::max(a.y, b.y);
        } else {
            dy = 0;
        }
    } else { // Horizontal segment
        dy = p.y - a.y;
        if (p.x < std::min(a.x, b.x)) {
            dx = p.x - std::min(a.x, b.x);
        } else if (p.x > std::max(a.x, b.x)) {
            dx = p.x - std::max(a.x, b.x);
        } else {
            dx = 0;
        }
    }
    return dx * dx + dy * dy;
}

void calculate_score_from_scratch(const std::vector<Point>& poly, int& m_count, int& s_count) {
    m_count = 0; s_count = 0;
    if (poly.size() < 3) return; // Not a valid polygon for containment
    for (const auto& fish_s : all_fish_structs) {
        if (is_inside_polygon_wn(fish_s.p, poly)) {
            if (fish_s.type == 1) m_count++;
            else s_count++;
        }
    }
}

// Calculate fish counts in a given rectangle using KD-tree
void calculate_score_delta_for_rectangle(KDNode* node, int r_min_x, int r_max_x, int r_min_y, int r_max_y,
                                         int& delta_m, int& delta_s) {
    delta_m = 0; delta_s = 0;

    if (!node || r_min_x > r_max_x || r_min_y > r_max_y) { // Invalid rectangle
        return;
    }

    // Iterative KD-tree traversal with subtree bbox pruning and whole-subtree aggregation.
    std::vector<KDNode*> stack;
    stack.reserve(64); // Reasonable reserve size for typical KD-tree depth
    stack.push_back(node);

    while (!stack.empty()) {
        KDNode* current_node = stack.back();
        stack.pop_back();
        if (!current_node) continue;

        // Disjoint?
        if (current_node->max_x < r_min_x || current_node->min_x > r_max_x || current_node->max_y < r_min_y ||
  current_node->min_y > r_max_y) {
            continue;
        }
        // Fully inside?
        if (r_min_x <= current_node->min_x && current_node->max_x <= r_max_x && r_min_y <= current_node->min_y
  && current_node->max_y <= r_max_y) {
            delta_m += current_node->m_cnt;
            delta_s += current_node->s_cnt;
            continue;
        }
        // Partial overlap: account this node's point, then traverse children
        const Point& pt = current_node->pt;
        if (pt.x >= r_min_x && pt.x <= r_max_x && pt.y >= r_min_y && pt.y <= r_max_y) {
```

```
                if (all_fish_structs[current_node->fish_struct_idx].type == 1) ++delta_m;
                else ++delta_s;
            }
            if (current_node->left) stack.push_back(current_node->left);
            if (current_node->right) stack.push_back(current_node->right);
        }
}

// Check intersection between two orthogonal segments p1s-p1e and p2s-p2e
bool segments_intersect(Point p1s, Point p1e, Point p2s, Point p2e) {
    // Normalize segments (sort endpoints to simplify overlap checks)
    if (p1s.x == p1e.x) { if (p1s.y > p1e.y) std::swap(p1s.y, p1e.y); } // Vertical, sort by y
    else { if (p1s.x > p1e.x) std::swap(p1s.x, p1e.x); } // Horizontal, sort by x
    if (p2s.x == p2e.x) { if (p2s.y > p2e.y) std::swap(p2s.y, p2e.y); }
    else { if (p2s.x > p2e.x) std::swap(p2s.x, p2e.x); }

    bool seg1_is_H = (p1s.y == p1e.y);
    bool seg2_is_H = (p2s.y == p2e.y);

    if (seg1_is_H == seg2_is_H) { // Both horizontal or both vertical
        if (seg1_is_H) { // Both horizontal
            // Check for y-alignment and x-overlap
            return p1s.y == p2s.y && std::max(p1s.x, p2s.x) <= std::min(p1e.x, p2e.x);
        } else { // Both vertical
            // Check for x-alignment and y-overlap
            return p1s.x == p2s.x && std::max(p1s.y, p2s.y) <= std::min(p1e.y, p2e.y);
        }
    } else { // One horizontal, one vertical (potential T-junction or cross)
        Point h_s = seg1_is_H ? p1s : p2s; Point h_e = seg1_is_H ? p1e : p2e;
        Point v_s = seg1_is_H ? p2s : p1s; Point v_e = seg1_is_H ? p2e : p1e;
        // Check if intersection point (v_s.x, h_s.y) lies on both segments
        return v_s.x >= h_s.x && v_s.x <= h_e.x && // x_intersect within horizontal segment's x-range
               h_s.y >= v_s.y && h_s.y <= v_e.y;  // y_intersect within vertical segment's y-range
    }
}

bool check_self_intersection_full(const std::vector<Point>& poly) {
    int M = poly.size();
    if (M < 4) return false;
    for (int i = 0; i < M; ++i) {
        Point p1s = poly[i];
        Point p1e = poly[(i + 1) % M];
        for (int j = i + 2; j < M; ++j) {
            // Skip checking adjacent edges.
            // Edge i is (poly[i], poly[(i+1)%M]). Edge j is (poly[j], poly[(j+1)%M]).
            // If i=0 and j=M-1, then edge i is (poly[0], poly[1]) and edge j is (poly[M-1], poly[0]). These
    are adjacent.
            if (i == 0 && j == M - 1) continue;

            Point p2s = poly[j];
            Point p2e = poly[(j + 1) % M];
            if (segments_intersect(p1s, p1e, p2s, p2e)) return true;
        }
    }
    return false;
}

// Local self-intersection check: checks edges starting at critical_edge_start_indices_const against all
    others
bool has_self_intersection_locally(const std::vector<Point>& poly, const std::vector<int>&
    critical_edge_start_indices_const) {
    int M = poly.size();
    if (M < 4) return false;

    std::vector<int> critical_indices = critical_edge_start_indices_const; // Make a copy to modify
    if (critical_indices.empty()) {
      return false;
    }

    std::sort(critical_indices.begin(), critical_indices.end());
    critical_indices.erase(std::unique(critical_indices.begin(), critical_indices.end()), critical_indices.end
      ());

    for (int edge1_s_idx_val_orig : critical_indices) {
        int edge1_s_idx_val = (edge1_s_idx_val_orig % M + M) % M; // Ensure positive modulo
        // No need to check edge1_s_idx_val bounds, it will be in [0, M-1]

        Point p1s = poly[edge1_s_idx_val];
        Point p1e = poly[(edge1_s_idx_val + 1) % M];

        for (int edge2_s_idx = 0; edge2_s_idx < M; ++edge2_s_idx) {
            bool is_adj_or_same_to_p1s_p1e = (edge2_s_idx == edge1_s_idx_val ||                      //
      Same edge
                                 edge2_s_idx == (edge1_s_idx_val + 1) % M ||          // edge2 starts
      where edge1 ends
                                 (edge2_s_idx + 1) % M == edge1_s_idx_val); // edge2 ends where edge1 starts
            if (is_adj_or_same_to_p1s_p1e) continue;

            Point p2s = poly[edge2_s_idx];
            Point p2e = poly[(edge2_s_idx + 1) % M];
            if (segments_intersect(p1s, p1e, p2s, p2e)) {
                return true;
```

```
            }
        }
    }
    return false;
}

bool has_distinct_vertices_unordered(const std::vector<Point>& poly) {
    if (poly.empty()) return true;
    std::unordered_set<Point, PointHash> distinct_pts;
    distinct_pts.reserve(poly.size()); // Pre-allocate for efficiency
    for(const auto& p : poly) {
        if (!distinct_pts.insert(p).second) return false; // Insertion failed, duplicate found
    }
    return true;
}

// Check basic structural validity of the polygon, uses cached perimeter
bool is_polygon_structurally_sound(const std::vector<Point>& poly, long long cached_perimeter) {
    int m = poly.size();
    if (m != 0 && (m < 4 || m > MAX_VERTICES)) return false;
    if (m == 0) return true;

    if (cached_perimeter > MAX_PERIMETER) return false;

    for (size_t i = 0; i < m; ++i) {
        const Point& p1 = poly[i];
        const Point& p2 = poly[(i + 1) % m];
        // Check coordinate bounds for p1
        if (p1.x < 0 || p1.x > MAX_COORD_VAL || p1.y < 0 || p1.y > MAX_COORD_VAL) return false;
        // The endpoint poly[(i+1)%m] will be checked as p1 in its own iteration,
        // but an explicit check here is also fine for robustness, though slightly redundant.
        if (poly[(i+1)%m].x < 0 || poly[(i+1)%m].x > MAX_COORD_VAL || poly[(i+1)%m].y < 0 || poly[(i+1)%m].y >
    MAX_COORD_VAL) return false;

        // Check axis-parallel and non-zero length edges
        if (p1.x != p2.x && p1.y != p2.y) return false; // Not axis-parallel
        if (p1.x == p2.x && p1.y == p2.y) return false; // Zero-length edge (duplicate consecutive vertices)
    }
    return true;
}

// Initial polygon generation using Kadane's algorithm on a coarse grid
std::vector<Point> create_initial_polygon_kadane() {
    const int GRID_SIZE_KADANE = 350; // Tunable parameter
    const int NUM_VALUES_KADANE = MAX_COORD_VAL + 1;
    // Ensure ACTUAL_CELL_DIM_KADANE is at least 1
    const int ACTUAL_CELL_DIM_KADANE = std::max(1, (NUM_VALUES_KADANE + GRID_SIZE_KADANE - 1) /
      GRID_SIZE_KADANE);

    std::vector<std::vector<long long>> grid_scores(GRID_SIZE_KADANE, std::vector<long long>(GRID_SIZE_KADANE,
      0));
    for (const auto& fish_s : all_fish_structs) {
        int r = fish_s.p.y / ACTUAL_CELL_DIM_KADANE;
        int c = fish_s.p.x / ACTUAL_CELL_DIM_KADANE;
        r = std::min(r, GRID_SIZE_KADANE - 1); r = std::max(r,0);
        c = std::min(c, GRID_SIZE_KADANE - 1); c = std::max(c,0);
        grid_scores[r][c] += fish_s.type; // Mackerel +1, Sardine -1
    }

    long long max_so_far = -3e18; // Sufficiently small number
    int best_r1 = 0, best_c1 = 0, best_r2 = -1, best_c2 = -1;

    // 2D Kadane's algorithm
    for (int c1_idx = 0; c1_idx < GRID_SIZE_KADANE; ++c1_idx) {
        std::vector<long long> col_strip_sum(GRID_SIZE_KADANE, 0);
        for (int c2_idx = c1_idx; c2_idx < GRID_SIZE_KADANE; ++c2_idx) {
            for (int r_idx = 0; r_idx < GRID_SIZE_KADANE; ++r_idx) {
                col_strip_sum[r_idx] += grid_scores[r_idx][c2_idx];
            }

            // 1D Kadane's on col_strip_sum
            long long current_strip_val = 0;
            int current_r1_1d = 0;
            for (int r2_idx_1d = 0; r2_idx_1d < GRID_SIZE_KADANE; ++r2_idx_1d) {
                long long val_here = col_strip_sum[r2_idx_1d];
                if (current_strip_val > 0 && current_strip_val + val_here > 0) { // Extend if sum remains
    positive
                    current_strip_val += val_here;
                } else { // Start new subarray
                    current_strip_val = val_here;
                    current_r1_1d = r2_idx_1d;
                }

                if (current_strip_val > max_so_far) {
                    max_so_far = current_strip_val;
                    best_r1 = current_r1_1d;
                    best_r2 = r2_idx_1d;
                    best_c1 = c1_idx;
                    best_c2 = c2_idx;
                }
            }
        }
```

```
        }
    }

    std::vector<Point> default_poly = {{0,0}, {1,0}, {1,1}, {0,1}}; // Minimal valid polygon

    // If no positive sum found, or issue, find best single cell
    if (best_r2 == -1 || max_so_far <=0 ) {
        max_so_far = -3e18; // Reset search for single best cell
        bool found_cell = false;
        for(int r=0; r<GRID_SIZE_KADANE; ++r) for(int c=0; c<GRID_SIZE_KADANE; ++c) {
            if(grid_scores[r][c] > max_so_far) {
                max_so_far = grid_scores[r][c];
                best_r1 = r; best_r2 = r; // Single cell
                best_c1 = c; best_c2 = c;
                found_cell = true;
            }
        }
        if (!found_cell || max_so_far <=0) return default_poly; // Still no good cell, return default
    }

    // Convert grid cell indices to actual coordinates
    int x_start = best_c1 * ACTUAL_CELL_DIM_KADANE;
    int y_start = best_r1 * ACTUAL_CELL_DIM_KADANE;
    int x_end = (best_c2 + 1) * ACTUAL_CELL_DIM_KADANE -1;
    int y_end = (best_r2 + 1) * ACTUAL_CELL_DIM_KADANE -1;

    // Clamp coordinates to valid range
    x_start = std::max(0, std::min(MAX_COORD_VAL, x_start));
    y_start = std::max(0, std::min(MAX_COORD_VAL, y_start));
    x_end = std::max(x_start, std::min(MAX_COORD_VAL, x_end)); // Ensure x_end >= x_start
    y_end = std::max(y_start, std::min(MAX_COORD_VAL, y_end)); // Ensure y_end >= y_start

    // Ensure non-zero dimensions for the polygon, minimum 1x1 actual area
    if (x_start == x_end) {
        if (x_start < MAX_COORD_VAL) x_end = x_start + 1;
        else if (x_start > 0) x_start = x_start -1; // Can't expand right, try expand left
        else return default_poly; // Single point at MAX_COORD_VAL, cannot form 1x1
    }
    if (y_start == y_end) {
        if (y_start < MAX_COORD_VAL) y_end = y_start + 1;
        else if (y_start > 0) y_start = y_start - 1;
        else return default_poly;
    }
    // After adjustment, if still degenerate, use default. This is rare.
    if (x_start == x_end || y_start == y_end) return default_poly;

    std::vector<Point> initial_poly = {
        {x_start, y_start}, {x_end, y_start}, {x_end, y_end}, {x_start, y_end}
    };
    return initial_poly;
}

// === SIMULATED ANNEALING ===
struct SAState {
    std::vector<Point> poly;
    int m_count;
    int s_count;
    long long perimeter_cache; // Added cache for perimeter

    SAState() : m_count(0), s_count(0), perimeter_cache(0) {} // Initialize perimeter_cache

    long long get_objective_score() const {
        return std::max(0LL, (long long)m_count - s_count + 1);
    }
    double get_raw_objective_score() const { // Used for SA acceptance probability
        return (double)m_count - s_count;
    }
};

// Calculates signed area * 2 of a polygon (shoelace formula)
long long polygon_signed_area_times_2(const std::vector<Point>& poly) {
    if (poly.size() < 3) return 0;
    long long area_sum = 0;
    for (size_t i = 0; i < poly.size(); ++i) {
        const Point& p1 = poly[i];
        const Point& p2 = poly[(i + 1) % poly.size()];
        area_sum += (long long)(p1.x - p2.x) * (p1.y + p2.y); // (x1-x2)(y1+y2) variant
    }
    return area_sum; // Positive for CCW, negative for CW
}

std::vector<int> sa_critical_edge_indices_cache; // Cache for local intersection check

// Guide coordinates for SA moves
std::vector<int> static_x_guides;
std::vector<int> static_y_guides;
std::vector<int> best_poly_x_guides;
std::vector<int> best_poly_y_guides;

void update_best_poly_guides(const SAState& new_best_state) {
    best_poly_x_guides.clear();
```

```
        best_poly_y_guides.clear();
        if (new_best_state.poly.empty()) return;

        std::set<int> temp_x_set, temp_y_set;
        for (const auto& p : new_best_state.poly) {
            temp_x_set.insert(p.x);
            temp_y_set.insert(p.y);
        }
        best_poly_x_guides.assign(temp_x_set.begin(), temp_x_set.end());
        best_poly_y_guides.assign(temp_y_set.begin(), temp_y_set.end());
}

void simulated_annealing_main() {
    SAState current_state;
    current_state.poly = create_initial_polygon_kadane();
    calculate_score_from_scratch(current_state.poly, current_state.m_count, current_state.s_count);
    current_state.perimeter_cache = calculate_perimeter(current_state.poly); // Calculate initial perimeter

    std::vector<Point> default_tiny_poly = {{0,0}, {1,0}, {1,1}, {0,1}};

    // Ensure initial polygon is valid, otherwise use default
    bool current_poly_initial_valid = is_polygon_structurally_sound(current_state.poly, current_state.
      perimeter_cache) &&
                                    current_state.poly.size() >= 4 &&
                                    has_distinct_vertices_unordered(current_state.poly) &&
                                    !check_self_intersection_full(current_state.poly);

    if (!current_poly_initial_valid) {
        current_state.poly = default_tiny_poly;
        calculate_score_from_scratch(current_state.poly, current_state.m_count, current_state.s_count);
        current_state.perimeter_cache = calculate_perimeter(current_state.poly); // Update perimeter for
      default
    }

    SAState best_state = current_state;
    update_best_poly_guides(best_state);

    // Prepare static guide coordinates from fish locations
    std::set<int> sx_set, sy_set;
    for(const auto& f_s : all_fish_structs) {
        sx_set.insert(f_s.p.x); sx_set.insert(std::max(0,f_s.p.x-1)); sx_set.insert(std::min(MAX_COORD_VAL,
      f_s.p.x+1));
        sy_set.insert(f_s.p.y); sy_set.insert(std::max(0,f_s.p.y-1)); sy_set.insert(std::min(MAX_COORD_VAL,
      f_s.p.y+1));
    }
    sx_set.insert(0); sx_set.insert(MAX_COORD_VAL); // Boundary guides
    sy_set.insert(0); sy_set.insert(MAX_COORD_VAL);

    static_x_guides.assign(sx_set.begin(), sx_set.end());
    static_y_guides.assign(sy_set.begin(), sy_set.end());

    double start_temp = 150.0;
    double end_temp = 0.01;

    long long current_signed_area = polygon_signed_area_times_2(current_state.poly);
    if (current_signed_area == 0 && current_state.poly.size() >=3) {
        current_signed_area = 1; // Avoid issues with zero area for sign logic
    }

    sa_critical_edge_indices_cache.reserve(10); // Max expected critical edges for current moves

    while (global_timer.elapsed() < ACTUAL_TIME_LIMIT_SECONDS) {
        double time_ratio = global_timer.elapsed() / ACTUAL_TIME_LIMIT_SECONDS;
        double temperature = start_temp * std::pow(end_temp / start_temp, time_ratio);
        // Fine-tune temperature near end or if it drops too fast
        if (temperature < end_temp && time_ratio < 0.95) temperature = end_temp;
        if (time_ratio > 0.95 && temperature > end_temp * 0.1) temperature = end_temp * 0.1; // Lower temp
      aggressively at the very end

        if (current_state.poly.size() < 4) { // Should not happen if logic is correct, but as a safeguard
            current_state.poly = default_tiny_poly;
            calculate_score_from_scratch(current_state.poly, current_state.m_count, current_state.s_count);
            current_state.perimeter_cache = calculate_perimeter(current_state.poly); // Update perimeter
            current_signed_area = polygon_signed_area_times_2(current_state.poly);
            if (current_signed_area == 0 && current_state.poly.size() >=3) current_signed_area = 1;
        }

        SAState candidate_state = current_state; // Copy current state
        sa_critical_edge_indices_cache.clear();

        int move_type_roll = rng.next_int(100);

        // Base probabilities for moves
        int targeted_move_prob = 35;
        int move_edge_prob = 35;
        int add_bulge_prob = 10;
        // simplify gets 20%

        bool near_vertex_limit = candidate_state.poly.size() + 2 > MAX_VERTICES;
        bool near_perimeter_limit = false;
```

```
      // Check perimeter using candidate_state's cached value
      if (candidate_state.poly.size() > 200 && candidate_state.perimeter_cache > MAX_PERIMETER * 0.9) {
          near_perimeter_limit = true;
      }

      // Adjust move probabilities based on polygon size/perimeter
      if (near_vertex_limit || near_perimeter_limit) {
          add_bulge_prob = 0;
          targeted_move_prob = 40;
          move_edge_prob = 40; // simplify is 20
      } else if (candidate_state.poly.size() > 400) {
          add_bulge_prob = 5;
          targeted_move_prob = 35;
          move_edge_prob = 35; // simplify is 25
      }

      int p_targeted = targeted_move_prob;
      int p_move_edge = p_targeted + move_edge_prob;
      int p_add_bulge = p_move_edge + add_bulge_prob;

      bool move_made = false;

      // Probabilities for snapping to guide coordinates
      double prob_dynamic_guide_snap = 0.20 + 0.20 * time_ratio;
      double prob_static_guide_snap_if_not_dynamic = 0.75;

      if (move_type_roll < p_targeted && candidate_state.poly.size() >= 4) { // Targeted Edge Move
          bool target_mackerel = rng.next_double() < 0.7;
          int n_fish_half = all_fish_structs.size() / 2;
          int fish_idx = target_mackerel ? rng.next_int(n_fish_half) : n_fish_half + rng.next_int(
n_fish_half);
          const auto& target_fish = all_fish_structs[fish_idx];
          bool is_inside = is_inside_polygon_wn(target_fish.p, candidate_state.poly);

          if ((target_fish.type == 1) == is_inside) {
              move_made = false; goto end_move_attempt_label;
          }

          long long min_dist_sq = -1;
          int best_edge_idx = -1;
          for (size_t i = 0; i < candidate_state.poly.size(); ++i) {
              long long d_sq = point_segment_dist_sq_ortho(target_fish.p, candidate_state.poly[i],
candidate_state.poly[(i+1)%candidate_state.poly.size()]);
              if (best_edge_idx == -1 || d_sq < min_dist_sq) {
                  min_dist_sq = d_sq;
                  best_edge_idx = i;
              }
          }
          if (best_edge_idx == -1) { move_made = false; goto end_move_attempt_label; }

          int edge_idx = best_edge_idx;
          Point p1_orig = candidate_state.poly[edge_idx];
          Point p2_orig = candidate_state.poly[(edge_idx + 1) % candidate_state.poly.size()];

          int new_coord_val;
          if (p1_orig.x == p2_orig.x) { new_coord_val = target_fish.p.x; }
          else { new_coord_val = target_fish.p.y; }

          new_coord_val = std::max(0, std::min(MAX_COORD_VAL, new_coord_val));

          int cur_delta_m=0, cur_delta_s=0;
          if (p1_orig.x == p2_orig.x) { // Vertical edge
              if (new_coord_val == p1_orig.x) {move_made = false; goto end_move_attempt_label;}

              int query_min_x, query_max_x;
              if (new_coord_val > p1_orig.x) { query_min_x = p1_orig.x + 1; query_max_x = new_coord_val; }
              else { query_min_x = new_coord_val; query_max_x = p1_orig.x - 1; }

              calculate_score_delta_for_rectangle(
                  fish_kdtree_root, query_min_x, query_max_x,
                  std::min(p1_orig.y, p2_orig.y), std::max(p1_orig.y, p2_orig.y),
                  cur_delta_m, cur_delta_s);

              int sign = (new_coord_val > p1_orig.x) ? 1 : -1;
              if (p1_orig.y > p2_orig.y) sign *= -1;
              if (current_signed_area < 0) sign *= -1;

              candidate_state.poly[edge_idx].x = new_coord_val;
              candidate_state.poly[(edge_idx + 1) % candidate_state.poly.size()].x = new_coord_val;
              candidate_state.m_count += sign * cur_delta_m;
              candidate_state.s_count += sign * cur_delta_s;
          } else { // Horizontal edge
              if (new_coord_val == p1_orig.y) {move_made = false; goto end_move_attempt_label;}

              int query_min_y, query_max_y;
              if (new_coord_val > p1_orig.y) { query_min_y = p1_orig.y + 1; query_max_y = new_coord_val; }
              else { query_min_y = new_coord_val; query_max_y = p1_orig.y - 1; }

              calculate_score_delta_for_rectangle(
                  fish_kdtree_root, std::min(p1_orig.x, p2_orig.x), std::max(p1_orig.x, p2_orig.x),
                  query_min_y, query_max_y,
                  cur_delta_m, cur_delta_s);
```

```
            int sign = (new_coord_val < p1_orig.y) ? 1 : -1;
            if (p1_orig.x > p2_orig.x) sign *= -1;
            if (current_signed_area < 0) sign *= -1;

            candidate_state.poly[edge_idx].y = new_coord_val;
            candidate_state.poly[(edge_idx + 1) % candidate_state.poly.size()].y = new_coord_val;
            candidate_state.m_count += sign * cur_delta_m;
            candidate_state.s_count += sign * cur_delta_s;
        }
        int M_cand = candidate_state.poly.size();
        sa_critical_edge_indices_cache.push_back((edge_idx - 1 + M_cand) % M_cand);
        sa_critical_edge_indices_cache.push_back(edge_idx);
        sa_critical_edge_indices_cache.push_back((edge_idx + 1) % M_cand);
        move_made = true;

    } else if (move_type_roll < p_move_edge && candidate_state.poly.size() >= 4 ) { // Move Edge
        int edge_idx = rng.next_int(candidate_state.poly.size());
        Point p1_orig = candidate_state.poly[edge_idx];
        Point p2_orig = candidate_state.poly[(edge_idx + 1) % candidate_state.poly.size()];

        int new_coord_val = -1;
        int cur_delta_m=0, cur_delta_s=0;
        bool coord_selected_successfully = false;

        // Determine which guides are relevant (X or Y)
        const std::vector<int>* relevant_dyn_guides = (p1_orig.x == p2_orig.x) ? &best_poly_x_guides : &
best_poly_y_guides;
        const std::vector<int>* relevant_static_guides = (p1_orig.x == p2_orig.x) ? &static_x_guides : &
static_y_guides;

        // Try snapping to dynamic (best poly) guides
        if (!relevant_dyn_guides->empty() && rng.next_double() < prob_dynamic_guide_snap) {
            new_coord_val = (*relevant_dyn_guides)[rng.next_int(relevant_dyn_guides->size())];
            coord_selected_successfully = true;
        }
        // If not, try snapping to static (fish) guides
        if (!coord_selected_successfully) {
            if (!relevant_static_guides->empty() && rng.next_double() <
prob_static_guide_snap_if_not_dynamic) {
                new_coord_val = (*relevant_static_guides)[rng.next_int(relevant_static_guides->size())];
                coord_selected_successfully = true;
            }
        }
        // If still not selected, use random displacement
        if (!coord_selected_successfully) {
            double step_factor = std::max(0.1, 1.0 - time_ratio * 0.95); // Step size decreases over time
            int base_step_max = std::max(1, (int)( (MAX_COORD_VAL/150.0) * step_factor + 1 ) );
            int random_displacement = rng.next_int(-base_step_max, base_step_max);
            if (time_ratio > 0.75 && rng.next_double() < 0.7) { // Very small steps near end
                random_displacement = rng.next_int(-2,2);
            }
            if (random_displacement == 0) random_displacement = (rng.next_double() < 0.5) ? -1:1;

            if (p1_orig.x == p2_orig.x) new_coord_val = p1_orig.x + random_displacement; // Vertical edge,
 move X
            else new_coord_val = p1_orig.y + random_displacement; // Horizontal edge, move Y
        }

        new_coord_val = std::max(0, std::min(MAX_COORD_VAL, new_coord_val)); // Clamp to bounds

        if (p1_orig.x == p2_orig.x) { // Vertical edge: (X_orig, Y_s) to (X_orig, Y_e)
            if (new_coord_val == p1_orig.x) {move_made = false; goto end_move_attempt_label;} // No change

            int query_min_x, query_max_x;
            if (new_coord_val > p1_orig.x) { // Moved right
                query_min_x = p1_orig.x + 1;
                query_max_x = new_coord_val;
            } else { // Moved left (new_coord_val < p1_orig.x)
                query_min_x = new_coord_val;
                query_max_x = p1_orig.x - 1;
            }

            calculate_score_delta_for_rectangle(
                fish_kdtree_root, query_min_x, query_max_x,
                std::min(p1_orig.y, p2_orig.y), std::max(p1_orig.y, p2_orig.y),
                cur_delta_m, cur_delta_s);

            int sign = (new_coord_val > p1_orig.x) ? 1 : -1; // Moving right is positive X change
            if (p1_orig.y > p2_orig.y) sign *= -1; // Correct for edge Y-direction (p1_orig.y to p2_orig.y
)
            if (current_signed_area < 0) sign *= -1; // Correct for CW polygon (area < 0)

            candidate_state.poly[edge_idx].x = new_coord_val;
            candidate_state.poly[(edge_idx + 1) % candidate_state.poly.size()].x = new_coord_val;
            candidate_state.m_count += sign * cur_delta_m;
            candidate_state.s_count += sign * cur_delta_s;
        } else { // Horizontal edge: (X_s, Y_orig) to (X_e, Y_orig)
            if (new_coord_val == p1_orig.y) {move_made = false; goto end_move_attempt_label;} // No change

            int query_min_y, query_max_y;
            if (new_coord_val > p1_orig.y) { // Moved up (Y increases)
```

```
                query_min_y = p1_orig.y + 1;
                query_max_y = new_coord_val;
            } else { // Moved down (Y decreases, new_coord_val < p1_orig.y)
                query_min_y = new_coord_val;
                query_max_y = p1_orig.y - 1;
            }

            calculate_score_delta_for_rectangle(
                fish_kdtree_root, std::min(p1_orig.x, p2_orig.x), std::max(p1_orig.x, p2_orig.x),
                query_min_y, query_max_y,
                cur_delta_m, cur_delta_s);

            int sign = (new_coord_val < p1_orig.y) ? 1 : -1; // Moving "down" (Y decreases) means positive
 sign if it expands area
            if (p1_orig.x > p2_orig.x) sign *= -1; // Correct for edge X-direction (p1_orig.x to p2_orig.x
)
            if (current_signed_area < 0) sign *= -1; // Correct for CW polygon

            candidate_state.poly[edge_idx].y = new_coord_val;
            candidate_state.poly[(edge_idx + 1) % candidate_state.poly.size()].y = new_coord_val;
            candidate_state.m_count += sign * cur_delta_m;
            candidate_state.s_count += sign * cur_delta_s;
        }
        int M_cand = candidate_state.poly.size();
        sa_critical_edge_indices_cache.push_back((edge_idx - 1 + M_cand) % M_cand);
        sa_critical_edge_indices_cache.push_back(edge_idx);
        sa_critical_edge_indices_cache.push_back((edge_idx + 1) % M_cand);
        move_made = true;

    } else if (move_type_roll < p_add_bulge && candidate_state.poly.size() + 2 <= MAX_VERTICES &&
candidate_state.poly.size() >=4) { // Add Bulge
        int edge_idx = rng.next_int(candidate_state.poly.size());
        Point p_s = candidate_state.poly[edge_idx]; // Start point of edge
        Point p_e = candidate_state.poly[(edge_idx + 1) % candidate_state.poly.size()]; // End point of
edge

        int new_coord_val = -1;
        bool coord_selected_successfully = false;

        const std::vector<int>* relevant_dyn_guides = (p_s.x == p_e.x) ? &best_poly_x_guides : &
best_poly_y_guides;
        const std::vector<int>* relevant_static_guides = (p_s.x == p_e.x) ? &static_x_guides : &
static_y_guides;

        // Try snapping bulge coord
        if (!relevant_dyn_guides->empty() && rng.next_double() < prob_dynamic_guide_snap) {
            new_coord_val = (*relevant_dyn_guides)[rng.next_int(relevant_dyn_guides->size())];
            coord_selected_successfully = true;
        }
        if (!coord_selected_successfully) {
            if (!relevant_static_guides->empty() && rng.next_double() <
prob_static_guide_snap_if_not_dynamic) {
                new_coord_val = (*relevant_static_guides)[rng.next_int(relevant_static_guides->size())];
                coord_selected_successfully = true;
            }
        }
        // If not snapped, random depth for bulge
        if (!coord_selected_successfully) {
            double depth_factor = std::max(0.1, 1.0 - time_ratio * 0.9);
            int base_depth_max = std::max(1, (int)( (MAX_COORD_VAL/300.0) * depth_factor + 1 ) );
            int random_abs_depth = rng.next_int(1, base_depth_max);
            if (time_ratio > 0.75 && rng.next_double() < 0.7) {
                random_abs_depth = rng.next_int(1,2);
            }
            int bulge_dir_sign = (rng.next_double() < 0.5) ? 1 : -1; // Randomly outwards or inwards
relative to edge line
            if (p_s.x == p_e.x) new_coord_val = p_s.x + bulge_dir_sign * random_abs_depth; // Vertical
edge, bulge in X
            else new_coord_val = p_s.y + bulge_dir_sign * random_abs_depth; // Horizontal edge, bulge in Y
        }

        new_coord_val = std::max(0, std::min(MAX_COORD_VAL, new_coord_val));

        Point v1_mod, v2_mod; // New vertices for the bulge
        int cur_delta_m=0, cur_delta_s=0;

        if (p_s.x == p_e.x) { // Original edge is vertical
            if (new_coord_val == p_s.x) {move_made = false; goto end_move_attempt_label;} // Bulge is flat
            v1_mod = {new_coord_val, p_s.y}; v2_mod = {new_coord_val, p_e.y};
            // Rectangle for delta score is between X=p_s.x and X=new_coord_val, over Y-span of original
edge
            calculate_score_delta_for_rectangle(
                fish_kdtree_root, std::min(p_s.x, new_coord_val), std::max(p_s.x, new_coord_val),
                std::min(p_s.y,p_e.y), std::max(p_s.y,p_e.y),
                cur_delta_m, cur_delta_s);
            int sign = (new_coord_val > p_s.x) ? 1 : -1; // Bulge to the right of edge is positive X
change
            if (p_s.y > p_e.y) sign *= -1; // Correct for edge Y-direction
            if (current_signed_area < 0) sign *= -1; // Correct for CW polygon
            candidate_state.m_count += sign * cur_delta_m;
            candidate_state.s_count += sign * cur_delta_s;
        } else { // Original edge is horizontal
```

```
            if (new_coord_val == p_s.y) {move_made = false; goto end_move_attempt_label;} // Bulge is flat
            v1_mod = {p_s.x, new_coord_val}; v2_mod = {p_e.x, new_coord_val};
            // Rectangle for delta score is between Y=p_s.y and Y=new_coord_val, over X-span of original
edge
            calculate_score_delta_for_rectangle(
                fish_kdtree_root, std::min(p_s.x,p_e.x), std::max(p_s.x,p_e.x),
                std::min(p_s.y, new_coord_val), std::max(p_s.y, new_coord_val),
                cur_delta_m, cur_delta_s);
            int sign = (new_coord_val < p_s.y) ? 1 : -1; // Bulge "downwards" (Y decreases) means positive
  sign if it expands area
            if (p_s.x > p_e.x) sign *= -1; // Correct for edge X-direction
            if (current_signed_area < 0) sign *= -1; // Correct for CW polygon
            candidate_state.m_count += sign * cur_delta_m;
            candidate_state.s_count += sign * cur_delta_s;
        }

        // Insert new vertices into polygon
        auto insert_pos_iter = candidate_state.poly.begin() + (edge_idx + 1);
        insert_pos_iter = candidate_state.poly.insert(insert_pos_iter, v1_mod);
        candidate_state.poly.insert(insert_pos_iter + 1, v2_mod);

        // Mark affected edges/vertices as critical for local intersection check
        sa_critical_edge_indices_cache.push_back(edge_idx);
        sa_critical_edge_indices_cache.push_back(edge_idx + 1);
        sa_critical_edge_indices_cache.push_back(edge_idx + 2);
        move_made = true;

    } else if (candidate_state.poly.size() > 4) { // Simplify Polygon (remove collinear vertex)
        int R_start_idx = rng.next_int(candidate_state.poly.size()); // Random start for search
        bool simplified_this_turn = false;
        for(int k_offset=0; k_offset < candidate_state.poly.size() ; ++k_offset) {
            int current_poly_size_before_erase = candidate_state.poly.size();
            if (current_poly_size_before_erase <= 4) break; // Cannot simplify further

            int p1_idx = (R_start_idx + k_offset) % current_poly_size_before_erase;
            int p0_idx_old = (p1_idx - 1 + current_poly_size_before_erase) %
current_poly_size_before_erase;
            int p2_idx_old = (p1_idx + 1) % current_poly_size_before_erase;

            const Point& p0 = candidate_state.poly[p0_idx_old];
            const Point& p1 = candidate_state.poly[p1_idx];
            const Point& p2 = candidate_state.poly[p2_idx_old];

            bool collinear_x = (p0.x == p1.x && p1.x == p2.x);
            bool collinear_y = (p0.y == p1.y && p1.y == p2.y);

            if (collinear_x || collinear_y) {
                candidate_state.poly.erase(candidate_state.poly.begin() + p1_idx);
                simplified_this_turn = true;

                int M_cand = candidate_state.poly.size();
                int critical_vertex_idx_in_new_poly;
                // Vertex p0 (at p0_idx_old) forms the new corner. Its index in new poly:
                if (p1_idx == 0) { // If p1 was poly[0], p0 was poly[last]. p0 is now poly[new_last]
                    critical_vertex_idx_in_new_poly = M_cand -1;
                } else { // Otherwise, p0's index p1_idx-1 is preserved.
                    critical_vertex_idx_in_new_poly = p1_idx - 1;
                }

                if (!candidate_state.poly.empty()) {
                    sa_critical_edge_indices_cache.push_back((critical_vertex_idx_in_new_poly - 1 + M_cand
) % M_cand);
                    sa_critical_edge_indices_cache.push_back(critical_vertex_idx_in_new_poly);
                    sa_critical_edge_indices_cache.push_back((critical_vertex_idx_in_new_poly + 1) %
M_cand);
                }
                break; // Simplified one vertex, enough for this turn
            }
        }
        if (!simplified_this_turn) {move_made = false; goto end_move_attempt_label;} // No simplification
found/possible
        move_made = true;
    }

    // After any move, recalculate perimeter for the candidate_state. This occurs only once per candidate.
    candidate_state.perimeter_cache = calculate_perimeter(candidate_state.poly);

    end_move_attempt_label:; // Label for goto if a move is aborted (e.g. no change)
    if (!move_made) continue; // No valid move attempted or made

    // Validate candidate polygon using the cached perimeter
    if (!is_polygon_structurally_sound(candidate_state.poly, candidate_state.perimeter_cache) ||
        candidate_state.poly.size() < 4 ||
        !has_distinct_vertices_unordered(candidate_state.poly)) {
        continue; // Invalid basic structure or duplicate vertices
    }

    if (has_self_intersection_locally(candidate_state.poly, sa_critical_edge_indices_cache)) {
         continue; // Self-intersection found
    }

    // Accept or reject candidate based on SA criteria
```

```
            double candidate_raw_obj_score = candidate_state.get_raw_objective_score();
            double current_raw_obj_score = current_state.get_raw_objective_score();
            double score_diff = candidate_raw_obj_score - current_raw_obj_score;

            if (score_diff >= 0 || (temperature > 1e-9 && rng.next_double() < std::exp(score_diff / temperature)))
            {
                current_state = std::move(candidate_state); // Accept move (perimeter_cache is moved as well)
                current_signed_area = polygon_signed_area_times_2(current_state.poly); // Update signed area
                 if (current_signed_area == 0 && !current_state.poly.empty() && current_state.poly.size() >=3)
         current_signed_area = 1; // Handle degenerate

                if (current_state.get_objective_score() > best_state.get_objective_score()) {
                    best_state = current_state; // New best solution found (perimeter_cache is copied here)
                    update_best_poly_guides(best_state); // Update dynamic guides
                }
            }
        }
    } // End SA loop

    // Final validation of the best found state: Recalculate perimeter explicitly for safety
    bool needs_reset_to_default = false;
    if (!is_polygon_structurally_sound(best_state.poly, calculate_perimeter(best_state.poly)) ||
        best_state.poly.size() < 4 ||
        !has_distinct_vertices_unordered(best_state.poly) ||
        check_self_intersection_full(best_state.poly) ) { // Full intersection check on best
        needs_reset_to_default = true;
    }

    if (needs_reset_to_default) { // If best state is invalid, revert to default
        best_state.poly = default_tiny_poly;
        calculate_score_from_scratch(best_state.poly, best_state.m_count, best_state.s_count);
        best_state.perimeter_cache = calculate_perimeter(best_state.poly); // Update for default
    }

    // If best score is 0, check if default polygon gives >0. (max(0, val+1))
    if (best_state.get_objective_score() == 0) {
        SAState temp_default_state; // Create a temporary default state to calculate its score
        temp_default_state.poly = default_tiny_poly;
        calculate_score_from_scratch(temp_default_state.poly, temp_default_state.m_count, temp_default_state.
     s_count);
        temp_default_state.perimeter_cache = calculate_perimeter(temp_default_state.poly); // Update for
     default

        if (best_state.get_objective_score() < temp_default_state.get_objective_score()) {
            best_state = temp_default_state;
        }
    }

    // Output the best polygon
    std::cout << best_state.poly.size() << "\n";
    for (const auto& p : best_state.poly) {
        std::cout << p.x << " " << p.y << "\n";
    }
}

int main(int argc, char *argv[]) {
    std::ios_base::sync_with_stdio(false);
    std::cin.tie(NULL);

    // Allow overriding time limit via command line arg, for local testing
    if (argc > 1) {
        try {
            ACTUAL_TIME_LIMIT_SECONDS = std::stod(argv[1]);
        } catch (const std::exception& e) { /* keep default if parse fails */ }
    }
    ACTUAL_TIME_LIMIT_SECONDS -= TIME_LIMIT_SECONDS_SAFETY_MARGIN;
    if (ACTUAL_TIME_LIMIT_SECONDS < 0.2) ACTUAL_TIME_LIMIT_SECONDS = 0.2; // Minimum sensible time limit

    // query_rect_indices_cache_kdtree.reserve(2 * 5000 + 500); // Removed: unused
    sa_critical_edge_indices_cache.reserve(10); // Small, for a few critical edges

    int N_half; // Number of mackerels (and sardines)
    std::cin >> N_half;

    all_fish_structs.resize(2 * N_half);
    std::vector<int> fish_indices_for_kdtree(2 * N_half);
    if (2 * N_half > 0) {
        std::iota(fish_indices_for_kdtree.begin(), fish_indices_for_kdtree.end(), 0);
    }

    // Read mackerels
    for (int i = 0; i < N_half; ++i) {
        std::cin >> all_fish_structs[i].p.x >> all_fish_structs[i].p.y;
        all_fish_structs[i].type = 1;
    }
    // Read sardines
    for (int i = 0; i < N_half; ++i) {
        std::cin >> all_fish_structs[N_half + i].p.x >> all_fish_structs[N_half + i].p.y;
        all_fish_structs[N_half + i].type = -1;
    }
```

```
    // Build KD-tree if there are fish
    if (!all_fish_structs.empty()) {
      fish_kdtree_root = build_kdtree(fish_indices_for_kdtree, 0, (int)all_fish_structs.size() - 1, 0);
    }

    simulated_annealing_main();

    // Clean up KD-tree memory
    if (fish_kdtree_root) delete_kdtree(fish_kdtree_root);

    return 0;
}
// EVOLVE-BLOCK-END
```

Listing 4: SHINKAEVOLVE Discovered `ahc039` Solution.

### D.3.2 ALE-BENCH LITE TASK: AHC025

```cpp
// EVOLVE-BLOCK-START
#include <iostream>
#include <vector>
#include <string>
#include <numeric>
#include <algorithm>
#include <iomanip>
#include <cmath>
#include <set>
#include <map>
#include <chrono>
#include <random>
#include <unordered_map>

// Timer
std::chrono::steady_clock::time_point program_start_time;
std::chrono::milliseconds time_limit_ms(1850);

// Global problem parameters
int N_items_global, D_groups_global, Q_total_global;
int queries_made = 0;

std::mt19937 rng_engine;

// Query Manager with optimized caching
class QueryManager {
private:
    int N, Q;
    int& queries_made_ref;
    std::vector<char> cmp1_flat; // flat N*N storage for 1v1 comparisons
    std::unordered_map<uint32_t, char> cmp1v2; // for 1v2 comparisons
    std::mt19937& rng;

    inline uint32_t key1v2(int a, int b, int c) const {
        int mn = std::min(b, c), mx = std::max(b, c);
        return (static_cast<uint32_t>(a) << 16) | (static_cast<uint32_t>(mn) << 8) | static_cast<uint32_t>(mx)
     ;
    }

    char perform_query_actual(const std::vector<int>& L_items, const std::vector<int>& R_items) {
        queries_made_ref++;
        std::cout << L_items.size() << " " << R_items.size();
        for (int item_idx : L_items) {
            std::cout << " " << item_idx;
        }
        for (int item_idx : R_items) {
            std::cout << " " << item_idx;
        }
        std::cout << std::endl;

        char result_char;
        std::cin >> result_char;
        return result_char;
    }

public:
    QueryManager(int N_, int Q_, int& qm, std::mt19937& r) : N(N_), Q(Q_), queries_made_ref(qm), rng(r) {
        cmp1_flat.assign(N * N, 0);
        cmp1v2.reserve(N * N / 4 + 10);
    }

    char compare1(int a, int b) {
        if (a == b) return '=';
        int mn = std::min(a, b), mx = std::max(a, b);
        char cached = cmp1_flat[mn * N + mx];
        if (cached != 0) {
            if (a == mn) return cached;
            return (cached == '<' ? '>' : (cached == '>' ? '<' : '='));
        }
        if (queries_made_ref >= Q) return '=';

        char res = perform_query_actual({a}, {b});
        if (a == mn) {
            cmp1_flat[mn * N + mx] = res;
        } else {
            if (res == '<') cmp1_flat[mn * N + mx] = '>';
            else if (res == '>') cmp1_flat[mn * N + mx] = '<';
            else cmp1_flat[mn * N + mx] = '=';
        }
        return res;
    }

    char compare1v2(int item_curr, int item_prev, int item_s_aux) {
        if (item_curr == item_prev || item_curr == item_s_aux || item_prev == item_s_aux) {
            if (item_prev == item_s_aux) return compare1(item_curr, item_prev);
            if (item_curr == item_prev) return compare1(item_curr, item_s_aux);
            return compare1(item_curr, item_prev);
        }
        uint32_t key = key1v2(item_curr, item_prev, item_s_aux);
        auto it = cmp1v2.find(key);
        if (it != cmp1v2.end()) return it->second;
```

```
        if (queries_made_ref >= Q) return '=';
        char res = perform_query_actual({item_curr}, {item_prev, item_s_aux});
        cmp1v2.emplace(key, res);
        return res;
    }

    void exhaust_queries() {
        if (N >= 2) {
            int a = 0, b = 1;
            while (queries_made_ref < Q) {
                perform_query_actual({a}, {b});
                ++b;
                if (b == a) ++b;
                if (b >= N) {
                    b = 0;
                    a = (a + 1) % N;
                    if (b == a) b = (b + 1) % N;
                }
            }
        }
    }
};

// Weight estimation module
class WeightEstimator {
private:
    static constexpr long long BASE_WEIGHT = 100000;
    static constexpr int FACTOR_GT = 200;
    static constexpr int FACTOR_LT = 50;
    static constexpr int FACTOR_XJ_FALLBACK = 100;

    QueryManager& qm;
    int N, D, Q;

    double estimate_log2(double val) {
        return (val <= 1.0) ? 0.0 : std::log2(val);
    }

    int calculate_query_cost(int N_val, int k_pivots) {
        if (k_pivots <= 0) return 0;
        if (k_pivots == 1) return std::max(0, N_val - 1);
        double cost = 0;
        cost += k_pivots * estimate_log2(k_pivots);
        for (int j = 2; j < k_pivots; ++j) {
            if (j - 1 > 0) cost += estimate_log2(j - 1);
        }
        cost += (N_val - k_pivots) * estimate_log2(k_pivots);
        return static_cast<int>(std::ceil(cost));
    }

    void merge_sort_pivots(std::vector<int>& pivots, int left, int right) {
        if (left >= right) return;
        int mid = (left + right) / 2;
        merge_sort_pivots(pivots, left, mid);
        merge_sort_pivots(pivots, mid + 1, right);

        int n1 = mid - left + 1, n2 = right - mid;
        std::vector<int> L(n1), R(n2);
        for (int i = 0; i < n1; ++i) L[i] = pivots[left + i];
        for (int j = 0; j < n2; ++j) R[j] = pivots[mid + 1 + j];

        int i = 0, j = 0, k = left;
        while (i < n1 && j < n2) {
            char cmp = qm.compare1(L[i], R[j]);
            if (cmp == '<' || cmp == '=') pivots[k++] = L[i++];
            else pivots[k++] = R[j++];
        }
        while (i < n1) pivots[k++] = L[i++];
        while (j < n2) pivots[k++] = R[j++];
    }

public:
    WeightEstimator(QueryManager& qm_, int N_, int D_, int Q_) : qm(qm_), N(N_), D(D_), Q(Q_) {}

    std::vector<long long> estimate_weights() {
        std::vector<long long> weights(N, BASE_WEIGHT);

        // Determine pivot count
        int k_pivots = (N > 0) ? 1 : 0;
        if (N > 1) {
            for (int k = N; k >= 1; --k) {
                if (calculate_query_cost(N, k) <= Q) {
                    k_pivots = k;
                    break;
                }
            }
        }
        k_pivots = std::min(k_pivots, N);

        if (k_pivots == 0) return weights;

        // Select and sort pivots
```

```
        std::vector<int> pivots(k_pivots);
        std::vector<int> indices(N);
        std::iota(indices.begin(), indices.end(), 0);
        std::shuffle(indices.begin(), indices.end(), rng_engine);
        for (int i = 0; i < k_pivots; ++i) pivots[i] = indices[i];

        if (k_pivots >= 2) {
            merge_sort_pivots(pivots, 0, k_pivots - 1);
        }

        // Estimate pivot weights
        if (k_pivots == 1) {
            weights[pivots[0]] = BASE_WEIGHT;
            for (int i = 0; i < N; ++i) {
                if (i == pivots[0]) continue;
                char res = qm.compare1(i, pivots[0]);
                if (res == '=') weights[i] = BASE_WEIGHT;
                else if (res == '<') weights[i] = std::max(1LL, BASE_WEIGHT * FACTOR_LT / 100);
                else weights[i] = std::max(1LL, BASE_WEIGHT * FACTOR_GT / 100);
            }
        } else {
            // Multi-pivot estimation
            weights[pivots[0]] = BASE_WEIGHT;

            // Handle p1
            char res_p1 = qm.compare1(pivots[1], pivots[0]);
            if (res_p1 == '=') weights[pivots[1]] = weights[pivots[0]];
            else if (res_p1 == '<') weights[pivots[1]] = std::max(1LL, weights[pivots[0]] * FACTOR_LT / 100);
            else weights[pivots[1]] = std::max(1LL, weights[pivots[0]] * FACTOR_GT / 100);

            if (res_p1 == '>' && weights[pivots[1]] == weights[pivots[0]]) {
                weights[pivots[1]] = weights[pivots[0]] + 1;
            }

            // Handle remaining pivots with binary search bracketing
            long long max_bound = BASE_WEIGHT * (N / std::max(1, D) + 10);
            for (int j = 2; j < k_pivots; ++j) {
                int cur = pivots[j], prev = pivots[j-1];
                char res = qm.compare1(cur, prev);

                if (res == '=') {
                    weights[cur] = weights[prev];
                } else if (res == '<') {
                    weights[cur] = std::max(1LL, weights[prev] * FACTOR_LT / 100);
                } else {
                    // Binary search to bracket X_j
                    long long X_low = 1, X_high = max_bound;
                    bool low_set = false, high_set = false;

                    int low_idx = 0, high_idx = j - 2;
                    int tries = std::max(1, static_cast<int>(std::ceil(estimate_log2(std::max(1, high_idx -
low_idx + 1)))));

                    for (int t = 0; t < tries && low_idx <= high_idx && queries_made < Q; ++t) {
                        int mid_idx = (low_idx + high_idx) / 2;
                        int s = pivots[mid_idx];
                        char res_1v2 = qm.compare1v2(cur, prev, s);

                        if (res_1v2 == '=') {
                            X_low = X_high = weights[s];
                            low_set = high_set = true;
                            break;
                        } else if (res_1v2 == '<') {
                            X_high = weights[s];
                            high_set = true;
                            high_idx = mid_idx - 1;
                        } else {
                            X_low = weights[s];
                            low_set = true;
                            low_idx = mid_idx + 1;
                        }
                    }

                    long long est_X;
                    if (low_set && !high_set) est_X = X_low * FACTOR_GT / 100;
                    else if (!low_set && high_set) est_X = X_high * FACTOR_LT / 100;
                    else if (low_set && high_set) est_X = (X_low + X_high) / 2;
                    else est_X = weights[prev] * FACTOR_XJ_FALLBACK / 100;

                    est_X = std::max(1LL, est_X);
                    weights[cur] = weights[prev] + est_X;
                }

                // Ensure monotonicity
                if (weights[cur] < weights[prev]) weights[cur] = weights[prev];
                if (res == '>' && weights[cur] == weights[prev]) weights[cur] = weights[prev] + 1;
            }

            // Estimate non-pivot weights
            std::vector<bool> is_pivot(N, false);
            for (int p : pivots) is_pivot[p] = true;
```

```
            for (int i = 0; i < N; ++i) {
                if (is_pivot[i]) continue;

                int low = 0, high = k_pivots - 1, found = -1;
                while (low <= high && queries_made < Q) {
                    int mid = (low + high) / 2;
                    char res = qm.compare1(i, pivots[mid]);
                    if (res == '=') { found = mid; break; }
                    else if (res == '<') high = mid - 1;
                    else low = mid + 1;
                }

                if (found != -1) {
                    weights[i] = weights[pivots[found]];
                    continue;
                }

                int pos = low;
                if (pos == 0) {
                    long long w0 = weights[pivots[0]];
                    if (k_pivots >= 2) {
                        long long w1 = weights[pivots[1]];
                        if (w1 > w0 && w0 > 0) weights[i] = std::max(1LL, w0 * w0 / w1);
                        else weights[i] = std::max(1LL, w0 / 2);
                    } else {
                        weights[i] = std::max(1LL, w0 / 2);
                    }
                } else if (pos == k_pivots) {
                    long long wk1 = weights[pivots[k_pivots - 1]];
                    if (k_pivots >= 2) {
                        long long wk2 = weights[pivots[k_pivots - 2]];
                        if (wk1 > wk2 && wk2 > 0) weights[i] = std::max(1LL, wk1 * wk1 / wk2);
                        else weights[i] = std::max(1LL, wk1 * 2);
                    } else {
                        weights[i] = std::max(1LL, wk1 * 2);
                    }
                } else {
                    long long wl = weights[pivots[pos - 1]];
                    long long wr = weights[pivots[pos]];
                    if (wl > 0 && wr > 0) {
                        weights[i] = static_cast<long long>(std::sqrt(static_cast<double>(wl) * wr));
                    } else {
                        weights[i] = (wl + wr) / 2;
                    }
                    weights[i] = std::max(weights[i], wl);
                    weights[i] = std::min(weights[i], wr);
                }
                weights[i] = std::max(1LL, weights[i]);
            }
        }

        // Final validation
        for (int i = 0; i < N; ++i) {
            if (weights[i] <= 0) weights[i] = BASE_WEIGHT;
        }

        return weights;
    }
};

// Assignment optimizer
class AssignmentOptimizer {
private:
    int N, D;
    std::vector<long long>& weights;
    std::mt19937& rng;

    double calc_variance(const std::vector<long long>& sums, long long total) {
        if (D <= 0) return 1e18;
        double mean = static_cast<double>(total) / D;
        double sum_sq = 0;
        for (long long s : sums) sum_sq += static_cast<double>(s) * s;
        double var = sum_sq / D - mean * mean;
        return std::max(0.0, var);
    }

public:
    AssignmentOptimizer(int N_, int D_, std::vector<long long>& w, std::mt19937& r)
        : N(N_), D(D_), weights(w), rng(r) {}

    std::vector<int> optimize() {
        std::vector<int> assignment(N, 0);
        std::vector<long long> group_sums(D, 0);
        std::vector<std::vector<int>> group_items(D);
        std::vector<int> item_pos(N);

        // Greedy initialization
        std::vector<std::pair<long long, int>> sorted_items;
        for (int i = 0; i < N; ++i) {
            sorted_items.emplace_back(-weights[i], i);
        }
        std::sort(sorted_items.begin(), sorted_items.end());
```

```
    long long total_sum = 0;
    for (auto [neg_w, item] : sorted_items) {
        int best_group = 0;
        for (int g = 1; g < D; ++g) {
            if (group_sums[g] < group_sums[best_group]) best_group = g;
        }
        assignment[item] = best_group;
        item_pos[item] = group_items[best_group].size();
        group_items[best_group].push_back(item);
        group_sums[best_group] += weights[item];
        total_sum += weights[item];
    }

    double current_var = calc_variance(group_sums, total_sum);

    // Enhanced local search with best-of-K
    if (D > 1) {
        const int MAX_ITERS = 400;
        const int K_ITEMS = 8;

        for (int iter = 0; iter < MAX_ITERS; ++iter) {
            if ((iter & 31) == 0) {
                auto now = std::chrono::steady_clock::now();
                if (std::chrono::duration_cast<std::chrono::milliseconds>(now - program_start_time) >=
time_limit_ms) break;
            }

            int max_g = 0, min_g = 0;
            for (int g = 1; g < D; ++g) {
                if (group_sums[g] > group_sums[max_g]) max_g = g;
                if (group_sums[g] < group_sums[min_g]) min_g = g;
            }
            if (max_g == min_g || group_items[max_g].empty()) break;

            // Find best relocate from max_g to min_g among top-K heaviest
            std::vector<std::pair<long long, int>> candidates;
            for (int item : group_items[max_g]) {
                candidates.emplace_back(weights[item], item);
            }
            if (candidates.empty()) break;
            std::sort(candidates.begin(), candidates.end(), [](const auto& a, const auto& b) { return a.
first > b.first; });
            if ((int)candidates.size() > K_ITEMS) candidates.resize(K_ITEMS);

            double best_var = current_var;
            int best_item = -1;
            for (auto [w, item] : candidates) {
                long long new_max = group_sums[max_g] - w;
                long long new_min = group_sums[min_g] + w;
                double new_var = calc_variance({new_max, new_min}, group_sums[max_g] + group_sums[min_g]);
                if (new_var + 1e-12 < best_var) {
                    best_var = new_var;
                    best_item = item;
                }
            }

            if (best_item == -1) break;

            // Apply move
            long long w = weights[best_item];
            group_sums[max_g] -= w;
            group_sums[min_g] += w;
            current_var = calc_variance(group_sums, total_sum);

            // Update tracking
            int pos = item_pos[best_item];
            int last = group_items[max_g].back();
            if (best_item != last) {
                group_items[max_g][pos] = last;
                item_pos[last] = pos;
            }
            group_items[max_g].pop_back();
            item_pos[best_item] = group_items[min_g].size();
            group_items[min_g].push_back(best_item);
            assignment[best_item] = min_g;
        }
    }

    // Targeted Simulated Annealing
    if (D > 1) {
        double T = std::max(1.0, current_var * 0.25);
        double cool_rate = 0.99985;
        std::uniform_real_distribution<double> unif(0.0, 1.0);
        int iterations = 0, no_imp = 0;

        while (true) {
            ++iterations;
            if ((iterations & 255) == 0) {
                auto now = std::chrono::steady_clock::now();
                if (std::chrono::duration_cast<std::chrono::milliseconds>(now - program_start_time) >=
time_limit_ms) break;
```

```
                            T *= cool_rate;
                            if (T < 1e-12) break;
                    }

                    // Targeted moves: 75% heavy-to-light relocate, 25% swap
                    if ((rng() % 4) != 0) {
                        // Targeted relocate
                        int max_g = 0, min_g = 0;
                        for (int g = 1; g < D; ++g) {
                            if (group_sums[g] > group_sums[max_g]) max_g = g;
                            if (group_sums[g] < group_sums[min_g]) min_g = g;
                        }

                        if (group_items[max_g].empty()) { ++no_imp; continue; }

                        // Pick heavy item from max group (best of 3 samples)
                        int item = group_items[max_g][rng() % group_items[max_g].size()];
                        for (int s = 0; s < 2; ++s) {
                            int cand = group_items[max_g][rng() % group_items[max_g].size()];
                            if (weights[cand] > weights[item]) item = cand;
                        }

                        long long w = weights[item];
                        long long new_max = group_sums[max_g] - w;
                        long long new_min = group_sums[min_g] + w;

                        double new_var = current_var;
                        new_var -= (static_cast<double>(group_sums[max_g]) * group_sums[max_g]) / D;
                        new_var -= (static_cast<double>(group_sums[min_g]) * group_sums[min_g]) / D;
                        new_var += (static_cast<double>(new_max) * new_max) / D;
                        new_var += (static_cast<double>(new_min) * new_min) / D;

                        double delta = new_var - current_var;
                        if (delta < 0 || unif(rng) < std::exp(-delta / T)) {
                            // Accept move
                            current_var = new_var;
                            group_sums[max_g] = new_max;
                            group_sums[min_g] = new_min;

                            int pos = item_pos[item];
                            int last = group_items[max_g].back();
                            if (item != last) {
                                group_items[max_g][pos] = last;
                                item_pos[last] = pos;
                            }
                            group_items[max_g].pop_back();
                            item_pos[item] = group_items[min_g].size();
                            group_items[min_g].push_back(item);
                            assignment[item] = min_g;

                            if (delta < -1e-12) no_imp = 0; else ++no_imp;
                        } else ++no_imp;
                    } else {
                        // Random swap
                        int g1 = rng() % D, g2 = rng() % D;
                        while (g2 == g1) g2 = rng() % D;
                        if (group_items[g1].empty() || group_items[g2].empty()) { ++no_imp; continue; }

                        int a = group_items[g1][rng() % group_items[g1].size()];
                        int b = group_items[g2][rng() % group_items[g2].size()];
                        long long wa = weights[a], wb = weights[b];

                        long long new_g1 = group_sums[g1] - wa + wb;
                        long long new_g2 = group_sums[g2] - wb + wa;

                        double new_var = current_var;
                        new_var -= (static_cast<double>(group_sums[g1]) * group_sums[g1]) / D;
                        new_var -= (static_cast<double>(group_sums[g2]) * group_sums[g2]) / D;
                        new_var += (static_cast<double>(new_g1) * new_g1) / D;
                        new_var += (static_cast<double>(new_g2) * new_g2) / D;

                        double delta = new_var - current_var;
                        if (delta < 0 || unif(rng) < std::exp(-delta / T)) {
                            current_var = new_var;
                            group_sums[g1] = new_g1;
                            group_sums[g2] = new_g2;

                            // Swap items
                            int pos_a = item_pos[a], pos_b = item_pos[b];
                            int back_a = group_items[g1].back(), back_b = group_items[g2].back();
                            if (a != back_a) { group_items[g1][pos_a] = back_a; item_pos[back_a] = pos_a; }
                            group_items[g1].pop_back();
                            if (b != back_b) { group_items[g2][pos_b] = back_b; item_pos[back_b] = pos_b; }
                            group_items[g2].pop_back();

                            item_pos[b] = group_items[g1].size(); group_items[g1].push_back(b); assignment[b] = g1
;
                            item_pos[a] = group_items[g2].size(); group_items[g2].push_back(a); assignment[a] = g2
;

                            if (delta < -1e-12) no_imp = 0; else ++no_imp;
                        } else ++no_imp;
```

```
                }

                if (no_imp > N * 12) break;
            }
        }

        return assignment;
    }
};

int main() {
    std::ios_base::sync_with_stdio(false);
    std::cin.tie(NULL);

    program_start_time = std::chrono::steady_clock::now();
    uint64_t seed = std::chrono::duration_cast<std::chrono::nanoseconds>(
        std::chrono::steady_clock::now().time_since_epoch()).count();
    rng_engine.seed(seed);

    std::cin >> N_items_global >> D_groups_global >> Q_total_global;

    QueryManager qm(N_items_global, Q_total_global, queries_made, rng_engine);
    WeightEstimator estimator(qm, N_items_global, D_groups_global, Q_total_global);

    std::vector<long long> weights = estimator.estimate_weights();

    qm.exhaust_queries();

    AssignmentOptimizer optimizer(N_items_global, D_groups_global, weights, rng_engine);
    std::vector<int> assignment = optimizer.optimize();

    for (int i = 0; i < N_items_global; ++i) {
        std::cout << assignment[i] << (i + 1 == N_items_global ? '\n' : ' ');
    }

    return 0;
}
// EVOLVE-BLOCK-END
```

Listing 5: SHINKAEVOLVE Discovered `ahc025` Solution.

## D.4 MIXTURE-OF-EXPERTS LOAD BALANCING LOSS

```python
def load_balancing_loss(
    gate_logits: tuple[torch.Tensor],
    num_experts: int,
    top_k: int = 2,
    attention_mask: Optional[torch.Tensor] = None,
) -> torch.Tensor:
    """
    Load balancing loss for Mixture-of-Experts models.

    parameters
    ----------
    layer_logits:
        list with shape (B, T, total_experts) per layer.
    total_experts:
        number of experts inside the moe feed-forward sub-block.
    top_k_experts:
        number of experts chosen per token (k in top-k gating).
    attention_mask:
        optional mask (B, T) where 0 marks padded tokens.

    returns
    -------
    torch.Tensor:
        scalar loss to be added to the training objective.
    """
    # determine device & flat token count
    device = gate_logits[0].device
    num_layers = len(gate_logits)
    bsz, seqlen = attention_mask.shape
    n_tokens = bsz * seqlen

    # merge layers into (tokens, layers, experts)
    stacked = torch.stack(gate_logits, dim=-2).to(device)
    logits = stacked.view(n_tokens, num_layers, num_experts)

    # obtain routing information
    _, routing_probs, sel_idx = route_logits_to_scores(logits, top_k)
    sel_mask = F.one_hot(sel_idx, num_experts)

    if attention_mask is None:
        # average over all tokens
        avg_sel = sel_mask.float().mean(dim=0)
        avg_prob = routing_probs.mean(dim=0)
    else:
        # expand & apply mask
        m_exp = (
            attention_mask.unsqueeze(-1)
            .unsqueeze(-1)
            .unsqueeze(-1)
            .expand(bsz, seqlen, num_layers, top_k, num_experts)
            .reshape(-1, num_layers, top_k, num_experts)
        )
        avg_sel = sel_mask.float().mul(m_exp).sum(dim=0) / m_exp.sum(dim=0)

        p_mask = (
            attention_mask.unsqueeze(-1)
            .unsqueeze(-1)
            .expand(bsz, seqlen, num_layers, num_experts)
            .reshape(-1, num_layers, num_experts)
        )
        avg_prob = routing_probs.mul(p_mask).sum(dim=0) / p_mask.sum(dim=0)

    # mismatch penalty
    per_layer = avg_sel * avg_prob.unsqueeze(-2)
    main_loss = per_layer.mean(0).sum() * num_experts

    # --- Minimum usage regularizer: softly penalize underused experts ---
    # avg_sel: (layers, top_k, experts)
    # For each expert, sum over top_k to get total selection per expert per layer
    avg_sel_sum = avg_sel.sum(dim=-2)   # (layers, experts)
    # Normalize so that sum over experts = 1 per layer
    avg_sel_norm = avg_sel_sum / (avg_sel_sum.sum(dim=-1, keepdim=True) + 1e-8)

    # Compute entropy of avg_prob per layer (routing distribution)
    entropy = -(avg_prob * torch.log(avg_prob + 1e-8)).sum(dim=-1)  # (layers,)
    max_entropy = torch.log(torch.tensor(num_experts, dtype=avg_prob.dtype, device=avg_prob.device))
    entropy_scale = 1.5 - entropy / (max_entropy + 1e-8)  # ranges from 0.5 (uniform) to 1.5 (concentrated)

    # Penalty: encourage each expert to be used at least min_threshold
    min_threshold = 0.01 * (64.0 / num_experts)

    min_usage_penalty = torch.relu(min_threshold - avg_sel_norm).sum(dim=-1)  # (layers,)
    penalty_coeff = 0.1

    # Final loss: main + entropy-scaled min usage penalty
    return main_loss + penalty_coeff * (min_usage_penalty * entropy_scale).mean()
```

Listing 6: SHINKAEVOLVE Discovered Mixture of Experts Load Balancing Loss.

