# OpenReview forum: "ShinkaEvolve: Towards Open-Ended and Sample-Efficient Program Evolution"
_ICLR.cc/2026/Conference — ICLR 2026 Poster_

### Official Review · Reviewer_nTzs · 2025-10-29

**Soundness:** 3
**Presentation:** 3
**Contribution:** 3
**Rating:** 6
**Confidence:** 2

**Summary:**

SHINKAEVOLVE is a framework that uses LLMs to evolve programs efficiently for scientific and engineering discovery. It addresses a major limitation of prior systems—sample inefficiency—by introducing three innovations: novelty-aware parent selection, code novelty rejection sampling, and a bandit-based strategy to choose the most effective LLMs during evolution. These components work together to dramatically reduce the number of evaluations needed to find high-quality solutions. On the classic circle packing optimization task, SHINKAEVOLVE achieves state-of-the-art results in only 150 program evaluations, orders of magnitude fewer than previous methods. It also generalizes across domains, improving math reasoning agents (AIME), competitive programming solutions (ALE-Bench), and even discovering a new load-balancing loss for training mixture-of-experts models. The system demonstrates that LLM-driven evolutionary search can be both powerful and sample-efficient, advancing progress toward open-ended automated discovery. However, it still relies on human-defined fitness functions and structured problem objectives, leaving full autonomy as future work.

**Strengths:**

SHINKAEVOLVE’s key strength is its high sample efficiency, achieving state-of-the-art results in as few as 150 iterations, which is orders of magnitude faster than prior evolutionary LLM systems. It also demonstrates generalization across domains, improving solutions in optimization, mathematical reasoning, competitive programming, and even neural network training. Additionally, it introduces reasonable mechanisms—like novelty-aware sampling and adaptive LLM selection—that make the search process both effective and scalable, setting a foundation for more open-ended AI discovery systems.

**Weaknesses:**

1. #1 claim (sample efficiency) is shown on one canonical task with incomplete fairness controls. They highlight “SOTA in <150 evaluations” on circle packing and show an evolution tree and cost curve (Fig. 4), but I don’t see: multiple seeds, wall-clock comparisons, or matched compute/budget against the strongest recent baselines. The paper states 150 generations and that it “outperforms AlphaEvolve in less than 150 evaluations,” but it seems not to report variance or rigorous ablation under equalized budgets. That makes the “orders-of-magnitude fewer iterations” headline hard to audit.

2. Every domain here is driven by an explicit numeric fitness (packing objective; CE+load-balance proxy for MoE; AIME accuracy under a fixed query budget). That’s fine for optimization, but the paper’s “towards open-ended discovery” framing isn’t backed by tasks where the objective is underspecified or emergent.

3. For the MoE study, SHINKAEVOLVE runs only 30 iterations during pretraining and optimizes a fitness that is the sum of CE and a load-imbalance term (L1 distance to uniform), with downstream generalization only implied. They later “scale up” to 2.7B parameters, but again emphasize CE/LBL trade-offs rather than end-task metrics; this makes it unclear whether the new regularizer robustly improves model quality beyond balancing tokens.

**Questions:**

N/A

---

> ### Author Response · Authors · 2025-11-25
> **Rebuttal Response Reviewer nTzs**
>
> We thank the reviewer for their thoughtful comments and appreciation. We would like to take the opportunity to discuss their suggestions and questions.
>
>
> > “#1 claim (sample efficiency) is shown on one canonical task with incomplete fairness controls. They highlight “SOTA in <150 evaluations” on circle packing and show an evolution tree and cost curve (Fig. 4), but I don’t see: multiple seeds, wall-clock comparisons, or matched compute/budget against the strongest recent baselines. The paper states 150 generations and that it “outperforms AlphaEvolve in less than 150 evaluations,” but it seems not to report variance or rigorous ablation under equalized budgets. That makes the “orders-of-magnitude fewer iterations” headline hard to audit.”
>
> Following the reviewer’s feedback, we have added to the latest revision 3 additional independent runs of ShinkaEvolve on the circle packing problem. 2 out of 3 runs discover solutions that outperform the previous best-known bound found by AlphaEvolve (see appendix, Figure 15) within less than 200 program evaluations. This highlights the reproducibility of our results. Figures 15 and 17 also demonstrate the overall scalability of ShinkaEvolve with the number of proposal and evaluation workers. Overall wall-clock time can be almost linearly reduced by a linear increase in resources. We note that the computational overhead added by ShinkaEvolve itself is negligible compared to LLM API calls and evaluation runtimes.
>
> Finally, to demonstrate the robustness of ShinkaEvolve, we further added to the latest revision new results on 8 additional mathematical optimization tasks. We hope these addotopmal inclusions will help reinforce the evidence supporting ShinkaEvolve’s robust discovery dynamics.
>
> > “Every domain here is driven by an explicit numeric fitness (packing objective; CE+load-balance proxy for MoE; AIME accuracy under a fixed query budget). That’s fine for optimization, but the paper’s “towards open-ended discovery” framing isn’t backed by tasks where the objective is underspecified or emergent.”
>
> We understand the reviewer’s concern and agree, here the term open-endedness is used loosely in the context of individual tasks. Our results demonstrate the continuous accumulation of stepping stones (see search tree, Figure 4, right). Furthermore, while ShinkaEvolve's parent sampling does require a fitness score, it also supports the provision of non-numerical feedback during optimization. Fully open-ended program evolution would arguably require innovation and reframing of the tasks themselves, and continuous recombination, as well as an interleaved literature search. While these are exciting future directions, we do believe that practical open-endedness does still require sample efficiency, and our work aims to make tangible steps in that direction.
>
> Following the reviewer’s feedback, we will edit Section 6 to make this discussion more explicit when mentioning the current limitations and future potential of our framework.

---

> > ### Author Response · Authors · 2025-11-25
> > **Rebuttal Response Reviewer nTzs Continued**
> >
> > > “For the MoE study, SHINKAEVOLVE runs only 30 iterations during pretraining and optimizes a fitness that is the sum of CE and a load-imbalance term (L1 distance to uniform), with downstream generalization only implied. They later “scale up” to 2.7B parameters, but again emphasize CE/LBL trade-offs rather than end-task metrics; this makes it unclear whether the new regularizer robustly improves model quality beyond balancing tokens.”
> >
> > In the center subplot of Figure 7 in our paper, we show that the load balancing loss discovered by ShinkaEvolve not only yields better CE/LBL trade-offs, but also higher end task metrics across a wide range of language modeling tasks [1, 2, 3, 4, 5, 6]:
> >
> > | Loss Type    |   LBL Coefficient |   CE Loss |   Missrouted Fraction |   Downstream task Accuracy |
> > |:-------------|------------------:|----------:|----------------------:|---------------------------:|
> > | Global LBL   |             0.1   |     0.375 |               0.00472 |                      0.357 |
> > | ShinkaEvolve |             0.1   |     0.373 |               0.00431 |                      **0.366** |
> > | Global LBL   |             0.01  |     0.367 |               0.0107  |                      0.362 |
> > | ShinkaEvolve |             0.01  |     0.365 |               0.00989 |                      **0.368** |
> > | Global LBL   |             0.001 |     0.363 |               0.0331  |                      0.367 |
> > | ShinkaEvolve |             0.001 |     0.363 |               0.0327  |                     **0.37**  |
> >
> > Following the reviewer's feedback, we realized our discussion and result exposition in our original revision were too centered around CE/LBL. To address this, in the latest revision, we made edits to Section 4.3 and  Appendix B to extend the discussion on how end-task metrics improvements vary with different regularization strengths and added new tables for a more thorough exposition.
> >
> > [1] Talmor, Alon, et al. "Commonsenseqa: A question answering challenge targeting commonsense knowledge." Proceedings of the 2019 Conference of the North American Chapter of the Association for Computational Linguistics: Human Language Technologies, Volume 1 (Long and Short Papers). 2019.
> >
> > [2] Zellers, Rowan, et al. "Hellaswag: Can a machine really finish your sentence?." arXiv preprint arXiv:1905.07830 (2019).
> >
> > [3] Mihaylov, Todor, et al. "Can a suit of armor conduct electricity? a new dataset for open book question answering." arXiv preprint arXiv:1809.02789 (2018).
> >
> > [4] Bisk, Yonatan, et al. "Piqa: Reasoning about physical commonsense in natural language." Proceedings of the AAAI conference on artificial intelligence. Vol. 34. No. 05. 2020.
> >
> > [5] Sakaguchi, Keisuke, et al. "Winogrande: An adversarial winograd schema challenge at scale." Communications of the ACM 64.9 (2021): 99-106.
> >
> > [6] Clark, Peter, et al. "Think you have solved question answering? try arc, the ai2 reasoning challenge." arXiv preprint arXiv:1803.05457 (2018).

---

### Official Review · Reviewer_Smk8 · 2025-10-31

**Soundness:** 1
**Presentation:** 1
**Contribution:** 1
**Rating:** 0
**Confidence:** 4

**Summary:**

This paper claims to introduce a new method that leverages LLMs to perform evolutionary algorithm search for the benefit of scientific discovery. Empirical results across a handful of domains are presented.

**Strengths:**

The setting is promising and timely. Generally, the text was typo-free.

**Weaknesses:**

The paper is not coherently written, and I suspect it is entirely generated by an LLM. A (very non-exhaustive) list of suspicious features include the following.
- Introduction is mostly the abstract repeated verbatim.
- Paragraph starting line 110 is suspicious. Many terms are imprecisely used or not defined, e.g. “elite size,” “mutation context,” “primary parent,” “archive samples.” I can’t imagine a human writing “diverse exemplars for creative recombination.” The rest of the paragraph is similarly flowery language while not being quite precise enough for an ML audience.
-The overall problem is not defined anywhere
- Line 129: F, P_i are not defined before use. p_i is defined twice, differently.
- r_i is both used for the rank (line 124) and for fitness value (line 210)
- The name ShinkaEvolve is never explained.

Even if the paper was not LLM generated, it does a poor job of explaining the concepts with enough details that I would be able to reconstruct the algorithm or convince me that the authors understand the novelty and components.

**Questions:**

Unfortunately, even if the authors can prove there are no ethical issues, I cannot understand the novelty of the contributions from the writing.

**Details Of Ethics Concerns:**

I suspect this is generated by an LLM.

---

> ### Author Response · Authors · 2025-11-28
> **Rebuttal Response Reviewer Smk8**
>
> Dear Reviewer Smk8,
>
> We thank the reviewer for their time and their assessment that the “setting is promising and timely.” While unfortunately, the review does not contain any technical questions regarding our paper, we want to refute the claim that this paper is “fully LLM-generated”.
>
> All authors of this paper are non-native English speakers. We have declared in the “LLM usage disclosure” section that we “acknowledge the use of LLMs to improve the grammar, clarity, and overall presentation of this manuscript". That being said, all of the paper was, at its core, drafted by humans. LLMs were solely used to adjust tone and reformulate text to fit into the page limits.
>  All empirical investigations, methodological innovations, and initial writing were conceived or conducted by the authors. All discovered code artifacts were attached as executable programs, and we have included the core codebase as part of our submission. We thereby follow the [policies and code of ethics](https://blog.iclr.cc/2025/08/26/policies-on-large-language-model-usage-at-iclr-2026/) outlined by ICLR program chairs.
>
> > “Introduction is mostly the abstract repeated verbatim.”
>
> The introduction of the paper provides a more detailed motivation for the work, includes a summary of current approaches, and outlines the innovations and results presented throughout the paper. There are natural parallels between the abstract and introduction (as in every paper), but there is no verbatim repetition of the abstract. We have generated all multi-phrase dual occurrences of both sections below:
>
> > The rapid advancement of **[[ large language models (LLMs) ]]** has transformed **[[ scientific discovery ]]** through agentic systems that autonomously conduct experiments and test hypotheses (Lu et al., 2024b; Yamada et al., 2025; Novikov et al., 2025; Zhang et al., 2025). These frameworks leverage LLMs as sophisticated mutation operators, iteratively refining candidate solutions with successful variants propagating through successive generations. This methodology has proven effective across domains such as competitive programming (Li et al., 2022), mathematical optimization (RomeraParedes et al., 2024), and automated agentic design (Hu et al., 2024). However, current implementations face significant practical limitations. The primary challenge is substantial sample inefficiency as existing approaches typically require **[[ thousands of ]]** evaluations, making them computationally expensive and time-consuming. This inefficiency stems from naive exploration strategies that fail to effectively leverage accumulated knowledge from previous generations. ShinkaEvolve addresses these challenges through **[[ three key ]]** algorithmic innovations that work synergistically to enhance sample efficiency. Our adaptive parent and LLM sampling intelligently balances exploration of novel regions with exploitation of known high-quality areas. Next, our code proposal novelty rejection sampling ensures efficient program mutations. Finally, our bandit-based LLM selection strategy dynamically adapts to the evolving state of the sampled archive parents and inspiration programs. Experimental validation across diverse domains demonstrates substantial improvements in both efficiency and solution quality, with ShinkaEvolve achieving state-of-the art results using orders of magnitude fewer evaluations than existing approaches. ShinkaEvolve full code, provided with this submission, will be open-sourced to accelerate open advancements to automated discovery across a broader range of problems. In summary: 1. We introduce ShinkaEvolve, an evolutionary framework with three key algorithmic **[[ innovations: a ]]** novel parent program sampling strategy, **[[ code novelty ]]** rejection-sampling, and adaptive performance-based **[[ LLM ensemble ]]** selection. 2. We compare ShinkaEvolve with prior frameworks on **[[ the canonical circle-packing ]]** task, achieving state-of-the-art results with orders-of-magnitude fewer iterations. 3. We demonstrate ShinkaEvolve ability to innovate beyond human and LLM-generated solutions across three additional engineering domains: agentic scaffolding (AIME), **[[ competitive programming ]]** (ALE-Bench), and **[[ LLM training ]]** design (mixture-of-expert loss).
>
> > ‘Paragraph starting line 110 is suspicious. Many terms are imprecisely used or not defined, e.g. “elite size,” “mutation context,” “primary parent,” “archive samples.”’
>
> These are common terms known to researchers with a background in evolutionary optimization. While we do acknowledge that these may appear foreign to researchers without this background, they made sense to all other reviewers who scored the presentation of our work, ranging from excellent to fair (4, 3, 3, 2). We can clarify the definition of these terms, but are irritated that they “count as evidence” for a fully LLM-generated paper.

---

> > ### Author Response · Authors · 2025-11-28
> > **Rebuttal Response Reviewer Smk8 Continued**
> >
> > > “The overall problem is not defined anywhere”
> >
> > The abstract explicitly states the problem as follows: “The field of LLM-driven scientific discovery has seen significant progress, but has yet to overcome a critical limitation: sample inefficiency, requiring thousands of samples to identify effective solutions. ShinkaEvolve takes a concrete step towards addressing this critical limitation by introducing three key innovations: [...]”.
> >
> > **Analysis by Pangram**
> >
> > As additional evidence that this paper was not “entirely generated by an LLM", we forward the analysis conducted by Pangram: https://www.pangram.com/history/2cf38e1f-6337-4a1f-b366-33c9a22330e6. It states that at least 61.7 percent of the paper was entirely generated by human authors. Furthermore, there is only a single passage deemed as a “high confidence” AI detection. This is in line with what we have disclosed and the ICLR guidelines.
> >
> > **Questions about our work’s results or contribution**
> >
> > Beyond the full LLM-generation claim and a couple of notes on notation, the review unfortunately does not provide a technical assessment of our work’s contributions and findings. We have improved the notation for the bandit fitness normalization as well as provided more ablations with regard to robustness and hyperparameter sensitivity. We are happy to answer any technical questions by the reviewer within the remainder of this rebuttal period.

---

### Official Review · Reviewer_6V9n · 2025-11-01

**Soundness:** 2
**Presentation:** 4
**Contribution:** 3
**Rating:** 6
**Confidence:** 3

**Summary:**

The paper introduces ShinkaEvolve, a method that uses LLMs in a sample-efficient way to generate code and algorithms for scientific discovery. The system is built to improve sample inefficiency by guiding the LLMs to only pursue promising ideas through three key steps: adaptive parent sampling, code novelty rejection-sampling and a bandit-based LLM selection strategy. The authors demonstrate their results on four scientific and engineering tasks.

**Strengths:**

- The paper aims to establish the building blocks for efficient agentic scientific discovery, which is a potentially high impact problem across multiple fields.
- The paper is well-structured, demonstrating clear explanations and principled ablations supporting its key contributions.
- The experiments cover four diverse and relevant domains, including an MoE architectural design challenge which is more open-ended. The results seem convincing and the discovered LBL regularization term found through this method is interesting (at the very least).
- The authors have shared their code and scripts for all experiments.

**Weaknesses:**

- The system is relatively complex, raising concerns that it may require quite different configurations across tasks or domains, which could limit its generalizability.
- A potential concern with using embedding similarity for code novelty rejection-sampling is that it may favor semantic or structural similarity, overlooking small but critical code modifications. Also, the LLM-as-a-judge component seems redundant with only marginal improvements at a high cost.
- The paper claims "order of magnitudes" fewer iterations compared to baselines for the circle packing task. However, it seems like the observed efficiency is at least partially an artifact of optimizing a surrogate objective of a relaxed task (Section B.1, Figure 10).

**Questions:**

1. Could you provide an estimate of the system's sensitivity across domains? Does the performance observed in one domain generalize to a new one without requiring extensive retuning of the system’s parameters?
2. Could you please elaborate on the limitations of using an embedding model for code novelty rejection-sampling? Also, in Table 3 (ALE-Bench config) it seems like it’s not being used.
3. Could you further clarify the core technical novelty of ShinkaEvolve and how it distinguishes itself form existing state-of-the-art evolutionary-based methods for LLM scientific discovery (e.g. AlphaEvolve [1])?

[1] Novikov et al. "AlphaEvolve: A coding agent for scientific and algorithmic discovery." (2025).

---

> ### Author Response · Authors · 2025-11-25
> **Rebuttal Response Reviewer 6V9n**
>
> We thank the reviewer for their time, feedback, and questions.
>
> > “The system is relatively complex, raising concerns that it may require quite different configurations across tasks or domains, which could limit its generalizability.”
>
> While ShinkaEvolve introduces a handful of additional hyperparameters, we found that default parameters are usually capable of making consistent discoveries, improving performance metrics. Additionally, following the reviewer’s feedback, we have added new results to the latest revision on 8 additional mathematical optimization tasks used in AlphaEvolve. All of them use the same hyperparameter settings and an API budget limit of $20 per task. 4 out of 9 tasks ShinkaEvolve either outperforms or performs approximately on par with AlphaEvolve (see table below):
>
> | Task                                   | Pre-AlphaEvolve | AlphaEvolve | ShinkaEvolve ($20) |
> |----------------------------------------|------------------|-------------|---------------------|
> | Analysis: first_autocorr (↓)           | 1.5098           | 1.5053      | 1.520388829         |
> | Analysis: sec_autocorr (↑)             | 0.88922          | 0.8962      | **0.90461387**      |
> | Analysis: uncert_ineq (↓)              | 0.35229          | 0.3521      | 0.35209909          |
> | Combinatorics: min_overlap (↓)         | 0.380927         | 0.380924    | 0.380989888         |
> | Combinatorics: sumdiff_sets (↑)        | 1.14465          | 1.1584      | 1.11843304          |
> | Geometry: maxmin_dist_ratio (↓)        | 12.890           | 12.88927    | **12.8892299**      |
> | Geometry: heilbronn_triangles (↑)      | 0.036            | 0.0365      | **0.03652584**      |
> | Geometry: kissing_11d (↑)              | 592              | 593         | 348.0               |
> | Geometry: pack_circ26 (↑)              | 2.634            | 2.63586     | **2.63598325**      |
>
> This shows the robustness and applicability of ShinkaEvolve across tasks. Additionally, the results included in the paper submission demonstrate the applicability across domains, including math, agent design, competitive coding, and loss function design.
>
> > “A potential concern with using embedding similarity for code novelty rejection-sampling is that it may favor semantic or structural similarity, overlooking small but critical code modifications. Also, the LLM-as-a-judge component seems redundant with only marginal improvements at a high cost.”
>
> Indeed, we agree that code embedding similarity alone is not sufficient to judge the novelty of a code edit. E.g., a simple change of the learning rate within a neural network training script would lead to a minimal change in the code embedding and therefore would be filtered out, while potentially being a meaningful mutation. That's why we introduced the “LLM-as-a-novelty-judge” second stage, which can provide additional safeguarding beyond superficial semantic similarity. Interestingly, the costs of this two-step filtering procedure are very low. In Figure 16, we demonstrate that on average, they only account for 0.2% of the total API costs.
>
> Following the reviewer’s feedback, we have updated the manuscript to note the key role of the LLM-as-a-judge component for coding tasks and better highlight its trivial cost.
>
>
> > “The paper claims "order of magnitudes" fewer iterations compared to baselines for the circle packing task. However, it seems like the observed efficiency is at least partially an artifact of optimizing a surrogate objective of a relaxed task (Section B.1, Figure 10).”
>
> Following the reviewer’s feedback, we have added to the latest revision more rebuttal experiments on the non-surrogate task (see appendix, Figure 15). We ran 3 independent experiments on the exact circle packing task. 2 out of 3 runs rediscovered the state-of-the-art packing solution using between 145 and 190 program mutations. Hence, the results demonstrate the robustness of ShinkaEvolve and that our result does not depend on the relaxation. Furthermore, we discuss in Appendix B.1 that the relaxation can be easily made strict by subtracting a small floating point number (1e-08) from each circle radius.
>
>
> > “Could you provide an estimate of the system's sensitivity across domains? Does the performance observed in one domain generalize to a new one without requiring extensive retuning of the system’s parameters?”
>
> As discussed in our previous responses, we show that a single configuration generalizes across 9 math tasks, leading to new state-of-the-art discoveries and beating AlphaEvolve on 4 of them (given a limited API budget). We found that different domains can require some initial calibration and construction of feedback strings, but the overall default parameters work well. One core heuristic to highlight is the number of independent subpopulations (islands), which tend to work best when being scaled with the API budget.

---

> > ### Author Response · Authors · 2025-11-25
> > **Rebuttal Response Reviewer 6V9n Continued**
> >
> > > “Could you please elaborate on the limitations of using an embedding model for code novelty rejection-sampling? Also, in Table 3 (ALE-Bench config) it seems like it’s not being used.”
> >
> > That's correct. Unfortunately, ALE-Bench programs are, on average, much longer in terms of lines of code and tokens. This exceeds the context length supported by common embedding models such as OpenAI's or Gemini embeddings (e.g., 8192 tokens). Hence, we were restricted to using novelty-rejection sampling for all non-ALE-Bench tasks.
> >
> > We agree that the LLM-novelty-judge approach relies on the capability of LLMs to act as ‘code diff‘-interpreters. This may not be given for every setting, but our ablation (Figure 8, right) demonstrates that this is at least the case for mathematical optimization tasks. Furthermore, this approach can benefit from general improvements in LLM capabilities.
> >
> > Following the reviewer’s question, we now explicitly discussed this tradeoff in Section 6, to clarify the above point.
> >
> > > “Could you further clarify the core technical novelty of ShinkaEvolve and how it distinguishes itself form existing state-of-the-art evolutionary-based methods for LLM scientific discovery (e.g. AlphaEvolve [1])?”
> >
> > Yes, of course. First off, ShinkaEvolve is open-source, accessible, and validated across four different domains, programming languages, and program execution runtimes, e.g., from minutes (mathematical optimization) to hours (MoE loss discovery). The core technical novelty stems from the following 3 methodological contributions:
> >
> > A parent program sampling procedure, which trades off the performance and the number of previously generated programs. This approach aims to balance novelty and performance of mutated parent programs.
> > The LLM ensemble selection procedure for the mutation LLM, which uses a bandit-based sampling distribution for prioritizing LLMs based on their previous improvement performance.
> > The program novelty rejection sampling approach, leveraging code embedding similarity, aimed to incentivize diverse program evaluations and minimize redundant work.
> >
> > Following the reviewer’s question, we will revise Section 1 to make our contributions in terms of algorithms, applications, and accessibility more explicit.

---

> > > ### Comment · Reviewer_6V9n · 2025-11-26
> > >
> > > Thank you for your response. While your clarifications are helpful, they don't significantly change my assessment. I will keep my score which is already high.

---

### Official Review · Reviewer_fDBZ · 2025-11-03

**Soundness:** 2
**Presentation:** 3
**Contribution:** 2
**Rating:** 6
**Confidence:** 4

**Summary:**

This paper proposes ShinkaEvolve, a framework that addresses sample inefficiency in LLM-driven program evolution through three key innovations: adaptive parent sampling balancing exploration and exploitation, code novelty rejection-sampling to avoid redundant mutations, and bandit-based LLM ensemble selection. The approach is validated across four domains including circle packing, mathematical reasoning, competitive programming, and neural architecture design, achieving state-of-the-art results with orders of magnitude fewer evaluations than existing methods.

Strengths:
1. The paper successfully demonstrates improvements across four diverse domains: geometric optimization, mathematical reasoning, competitive programming, and neural architecture design. It shows consistent performance gains with proper ablation studies validating each algorithmic component's contribution, and includes generalization experiments such as the AIME scaffold working across different LLMs that strengthen the practical applicability claims.
2. The work introduces a well-integrated three-component system combining adaptive parent sampling, embedding-based novelty rejection, and bandit-based LLM selection that addresses real limitations in existing approaches. The framework produces immediately applicable results like the discovered MoE load balancing loss for training large language models and competitive programming solutions that outperform expert baselines, effectively bridging evolutionary computation and modern AI research with both principled algorithmic design and strong empirical validation.

However, the bandit-based LLM ensemble approach uses a relatively simple UCB1 variant. More sophisticated multi-armed bandit algorithms or contextual bandits that consider the current state of the search could potentially yield better results. The paper doesn't justify why this approach was chosen.
In addition, this study is an applied engineering method of LLMs, the innovation of technical theory is relatively general.

**Strengths:**

1.	The study successfully demonstrates improvements across four diverse domains: geometric optimization, mathematical reasoning, competitive programming, and neural architecture design. It shows consistent performance gains with proper ablation studies validating each algorithmic component's contribution, and includes generalization experiments such as the AIME scaffold working across different LLMs that strengthen the practical applicability claims.
2.	The work introduces a well-integrated three-component system combining adaptive parent sampling, embedding-based novelty rejection, and bandit-based LLM selection that addresses real limitations in existing approaches. The framework produces immediately applicable results like the discovered MoE load balancing loss for training large language models and competitive programming solutions that outperform expert baselines, effectively bridging evolutionary computation and modern AI research with both principled algorithmic design and strong empirical validation.

**Weaknesses:**

the bandit-based LLM ensemble approach uses a relatively simple UCB1 variant. More sophisticated multi-armed bandit algorithms or contextual bandits that consider the current state of the search could potentially yield better results. The paper doesn't justify why this approach was chosen.
In addition, this study is an applied engineering method of LLMs, the innovation of technical theory is relatively general.

**Questions:**

the bandit-based LLM ensemble approach uses a relatively simple UCB1 variant. More sophisticated multi-armed bandit algorithms or contextual bandits that consider the current state of the search could potentially yield better results. The paper doesn't justify why this approach was chosen.

---

> ### Author Response · Authors · 2025-11-25
> **Rebuttal Response Reviewer fDBZ**
>
> We thank the reviewer for their time and appreciation of our proposed system and ablation studies.
>
>
> > “The bandit-based LLM ensemble approach uses a relatively simple UCB1 variant. More sophisticated multi-armed bandit algorithms or contextual bandits that consider the current state of the search could potentially yield better results. The paper doesn't justify why this approach was chosen.”
>
> While we recognize that various bandit approaches exist, efficient ensembling of LLMs is a significant research topic in itself. In order to address your concern, we have implemented an additional ablation by implementing a Thompson sampling bandit algorithm for LLM selection. In Appendix Figure 19, we compare UCB1 (our proposed approach), Thompson sampling, and fixed (uniformly sampling models) strategies. While all approaches show similar asymptotic performance, UCB1 (followed by Thompson sampling) performs the most sample-efficient.
>
> > “In addition, this study is an applied engineering method of LLMs, the innovation of technical theory is relatively general.”
>
> ShinkaEvolve introduces an agentic evolutionary test-time scaling paradigm with several core technological innovations, including adaptive parent sampling, LLM bandit prioritization and novelty rejection sampling. Furthermore, we have demonstrated that this system is capable of making state-of-the-art discoveries across four diverse application fields. Thereby, we argue that generality and open-source accessibility are actually strengths of our work.
>
> Following the reviewer’s feedback, we made edits to the introduction to make the democratization-oriented nature of our contribution more explicit.

---

### Official Review · Reviewer_R7yb · 2025-11-11

**Soundness:** 2
**Presentation:** 2
**Contribution:** 2
**Rating:** 4
**Confidence:** 2

**Summary:**

This paper introduces ShinkaEvolve, an evolutionary framework to overcome the sample efficiency problems in prior works with three synergistic pieces: Parent sampling, code-novelty rejection sampling, and Bandit-based LLM ensemble selection. Experiments on several benchmarks and ablations prove its sample efficiency and broad utility.

**Strengths:**

1. **Clear Motivation**: The paper targets sample inefficiency in LLM-driven program evolution and proposes three synergistic components:  novelty-weighted parent selection, code-novelty rejection sampling, and bandit-based LLM ensemble selection—all described with algorithmic detail.

2. **Concrete Reproducibility Details**: The paper includes hyperparameters, task harness details, and states that anonymized code is provided with plans to open-source.

3. **Multi-domain Evidence**: Extends beyond a single toy task: evolves AIME reasoning scaffolds, improves ALE-Bench solutions, and proposes an MoE load-balancing regularizer for training—broadening the paper’s relevance.

**Weaknesses:**

**Heavy System Complexity and Tuning without Sufficient Sensitivity Analysis**: The framework has many moving parts and hyperparameters (islands, migration, thresholds, patch types, UCB1, etc.), increasing the risk that performance depends on careful engineering. Specifically, a fixed embedding-similarity cutoff (0.95) is used; aside from showing it helps, there’s little analysis of the sensitivity or cross-task robustness of that threshold.

**Modest Gains without Statistical Significance**: Average improvement for ALE-Bench is about 2.3% over a strong baseline across 10 tasks—useful but relatively small; variance or statistical significance isn’t discussed. The SOTA claim for Circle-packing comparisons is presented mainly relative to AlphaEvolve; the breadth of baselines in this domain isn’t deeply examined.

**Effect-size Ambiguity for Ensembling**: Text/figure messaging is inconsistent, see question 1.

**Questions:**

1. Could you explain why it claims the bandit ensemble “significantly outperforms” both baselines (line 455), yet also notes it only “slightly improves” (Figure 8) over a fixed uniform ensemble? Quantitative deltas are not explicit.

2. What’s the exact parent-selection formula? The ablation plot labels “Novelty Weighted,” but the description elsewhere emphasizes fitness and offspring-count weighting—does novelty enter the score, and how?

3. Why choose a 0.95 embedding-similarity threshold—and how sensitive are results to it? The threshold is fixed for several tasks without a sweep or rationale.

4. Why is novelty filtering disabled on ALE-Bench but enabled on AIME/circle-packing? What trade-offs dictated this per-domain choice?

---

> ### Author Response · Authors · 2025-11-25
> **Rebuttal Response Reviewer R7yb**
>
> We thank R7yb reviewer for their time and detailed investigation of settings and sensitivity. We are also happy with the assessment of our clear motivation and multi-domain evidence.
>
> > **System complexity**: “aside from showing it helps, there’s little analysis of the sensitivity or cross-task robustness of that threshold.” and “Why choose a 0.95 embedding-similarity threshold—and how sensitive are results to it? The threshold is fixed for several tasks without a sweep or rationale.”
>
> Following the reviewer’s feedback, we added to the latest revision two more ablations on the similarity threshold used for novelty rejection sampling and the bandit LLM selection algorithm. First, we compare thresholds of 1.0 (no rejection), 0.995, 0.9, and 0.5 (heavily rejecting similar programs) next to the base threshold of 0.95. The larger values perform better, indicating that conservatively rejecting similar programs performs well for this domain (Figure 18). Second, we compare UCB1 (our proposed approach), Thompson sampling, and fixed (uniformly sampling models) strategies. While all approaches show similar asymptotic performance, UCB1 (followed by Thompson sampling) performs the most sample-efficient (Figure 19).
>
> Finally, we show that a single configuration (threshold 0.995) can perform well on 9 mathematical optimization tasks, outperforming AlphaEvolve's discoveries on 4 out of them with a budget of only $20 per task (see details and table in the general comment).
>
> > **Statistical significance**: “Average improvement for ALE-Bench is about 2.3% over a strong baseline across 10 tasks—useful but relatively small; variance or statistical significance isn’t discussed.”
>
> In Figure 6 (right), we show that ShinkaEvolve can discover improvements on 8 out of 10 tasks. While the 2.3% improvement may appear small, we note that we are using a highly optimized initial solution provided by the ALE-Agent to seed the search process. Hence, the observed improvements within a small number of ShinkaEvolve program generations (only 50 for ALE tasks, while ALE-Agent generated approximately 1000 programs) are noteworthy. Furthermore, on one of the tasks (ahc039), the placement of the combination of agents would have led to a second-place finish (improvement from the original ALE-Agent being placed fifth). We picked this setup to demonstrate that ShinkaEvolve can work on top of other coding/optimization agents with potentially orthogonal gains. Following the reviewer’s feedback, we have edited Section 4.3 to make this more explicit within the text.
>
> > **Statistical significance**: “The SOTA claim for Circle-packing comparisons is presented mainly relative to AlphaEvolve; the breadth of baselines in this domain isn’t deeply examined.”
>
> In Figure 1, we also compare against discoveries reported by LLM4AD and OpenEvolve. Additionally, our ablations show comparisons across different ShinkaEvolve variants, including, e.g., hill climbing parent selection, no rejection sampling, and fixed model ensembling. If the reviewer would like to see any specific comparison, we are happy to do our best to provide this during the remainder of the rebuttal period.
>
> Nonetheless, following the reviewer’s feedback, we have added 3 more independent runs for the circle packing task (see appendix, Figure 15). Three out of the overall 4 runs discover solutions either on par or better than AlphaEvolve in a maximum of 200 program evaluations. This indicates the robustness of the discovery process across trials.
>
> > Question: “Could you explain why it claims the bandit ensemble “significantly outperforms” both baselines (line 455), yet also notes it only “slightly improves” (Figure 8) over a fixed uniform ensemble? Quantitative deltas are not explicit.”
>
> We thank the reviewer for spotting this inconsistency. We excuse the imprecise wording and have updated the manuscript in order to address this.

---

> > ### Author Response · Authors · 2025-11-25
> > **Rebuttal Response Reviewer R7yb Continued**
> >
> > > Question: ‘What’s the exact parent-selection formula? The ablation plot labels “Novelty Weighted,” but the description elsewhere emphasizes fitness and offspring-count weighting—does novelty enter the score, and how?’
> >
> > Again, we excuse the imprecision. Novelty-weighted corresponds to the second equation, factoring in the number of parent offspring. Hence, “weighted” and “novelty-weighted” correspond to the same parent selection outlined in section 3.1. The sampling distribution downweights programs with many offspring, while it upweights high-performing programs. We have modified the latest revision to make this clearer.
> >
> > > Question: “Why is novelty filtering disabled on ALE-Bench but enabled on AIME/circle-packing? What trade-offs dictated this per-domain choice?”
> >
> > ALE-Bench programs, compared to the other considered tasks, are often much longer in terms of lines of code and tokens. This exceeds the context length supported by common embedding models (e.g., 8192 tokens). Hence, while we would have liked to use the embedding rejection-sampling also for ALE-Bench tasks, it was not technically feasible.
> >
> > Following the reviewer’s feedback, we will explicitly mentioned this when listing our hyperparameter choices in Appendix B.

---

### Author Response · Authors · 2025-11-25
**General Rebuttal Response**

We thank all reviewers for their thoughtful feedback. The manuscript has been significantly enhanced in order to address the reviewer's feedback. We want to quickly address the point of novelty and highlight the major revision additions we made to our paper:

**Contribution and complexity of our work**: Our work introduces an open-source framework for evolutionary program optimization with improved sample efficiency. The results demonstrate that ShinkaEvolve can successfully be applied to four vastly different domains, including mathematical optimization, agent scaffold design, competitive programming, and automated loss function design. Our core methodological introductions, compared to AlphaEvolve and OpenEvolve, include the following:

1. A parent program sampling procedure, which trades off the performance and the number of previously generated programs. This approach aims to balance novelty and performance of mutated parent programs.
2. The LLM ensemble selection procedure for the mutation LLM, which uses a bandit-based sampling distribution for prioritizing LLMs based on their previous improvement performance.
3. A program novelty rejection sampling approach leveraging code embedding similarity aimed to incentivize diverse program evaluations and minimize redundant work.

While all of these come with (a limited set of) additional hyperparameters, our rebuttal results (see below) show that these only affect the sample efficiency of the method and not the asymptotic performance. Hence, the method still successfully demonstrates the automated discovery of high-performing program solutions and democratizes access to such approaches.

**Hyperparameter robustness of ShinkaEvolve**: With this rebuttal, we have conducted several additional hyperparameter sensitivity ablations. We demonstrate that the embedding threshold, when set in a reasonable range (not too conservative), does not affect the performance beyond the speed of discovery. Furthermore, UCB LLM ensembling outperforms a newly added Thompson Sampling bandit approach. Finally, we demonstrate that a single hyperparameter configuration robustly performs well across 9 mathematical optimization problems. More details are provided below and in the appendix.

**Additional revision experiments**:

1. **Reproducibility of Circle Packing results discovered by ShinkaEvolve**: We have added 3 additional independent runs of ShinkaEvolve on the circle packing problem. 2 out of 3 discover solutions that outperform the previous best-known bound found by AlphaEvolve (see appendix, Figure 15) within less than 200 program evaluations. This highlights the reproducibility of our results.

2. **Overall hyperparameter robustness**: We have added results on 8 additional mathematical optimization tasks used in AlphaEvolve. All preliminary results are collected using the same hyperparameter settings across the tasks and an API budget limit of $20 per task. Given this limited budget, we find that on 4 out of 9 tasks ShinkaEvolve either outperforms or performs approximately on par with AlphaEvolve (see table below):

| Task                                   | Pre-AlphaEvolve | AlphaEvolve | ShinkaEvolve ($20) |
|----------------------------------------|------------------|-------------|---------------------|
| Analysis: first_autocorr (↓)           | 1.5098           | 1.5053      | 1.520388829         |
| Analysis: sec_autocorr (↑)             | 0.88922          | 0.8962      | **0.90461387**      |
| Analysis: uncert_ineq (↓)              | 0.35229          | 0.3521      | 0.35209909          |
| Combinatorics: min_overlap (↓)         | 0.380927         | 0.380924    | 0.380989888         |
| Combinatorics: sumdiff_sets (↑)        | 1.14465          | 1.1584      | 1.11843304          |
| Geometry: maxmin_dist_ratio (↓)        | 12.890           | 12.88927    | **12.8892299**      |
| Geometry: heilbronn_triangles (↑)      | 0.036            | 0.0365      | **0.03652584**      |
| Geometry: kissing_11d (↑)              | 592              | 593         | 348.0               |
| Geometry: pack_circ26 (↑)              | 2.634            | 2.63586     | **2.63598325**      |

---

> ### Author Response · Authors · 2025-11-25
> **General Rebuttal Response (continued)**
>
> 3. **Alternative bandit algorithms for LLM prioritization**: We added an additional experiment comparing our proposed UCB prioritized LLM ensemble approach with Thompson sampling and fixed uniform sampling. All three approaches discover circle packing solutions rivaling AlphaEvolve, but they require different numbers of program proposals. UCB prioritization is the most sample-efficient (see appendix, Figure 19).
>
> 4. **Embedding Threshold Sensitivity**: Next, to the base threshold of 0.95, we compared four different embedding thresholds (0.5, 0.9, 0.995, and no threshold) for rejecting program proposals based on their embedding similarity. The most efficient discovery (the lowest number of evaluated program proposals) can be achieved with a lenient threshold; using no threshold at all also works for high budgets (see appendix, Figure 18).
>
> 5. **Asynchronous scaling of ShinkaEvolve**: The results displayed in the paper used a single sequential proposal generation worker and multiple evaluation workers. Since then, we have put significant effort into scaling the number of proposal workers and demonstrated clear runtime scaling with additional worker resources. The overall runtime can approximately be linearly sped up using more proposal workers for short runtime evaluation settings (see appendix, Figure 17).
>
> 6. **API Cost Breakdown**: We also added more detail on the API cost breakdown across program proposals, code embedding, rejection sampling, and meta-summarization in the appendix. More than 95% of the cost is spent on program proposals (see appendix, Figure 16).
>
> **Claims of full LLM generation of the paper**: We reject the claim that our paper was generated by an LLM in full. As evidence, we forward the analysis conducted by Pangram: https://www.pangram.com/history/2cf38e1f-6337-4a1f-b366-33c9a22330e6. It states that at least 61.7 percent of the paper was entirely generated by human authors. Furthermore, there is only a single passage deemed as a “high confidence” AI detection. This is in line with what we have disclosed and the ICLR guidelines. LLMs were solely used to improve the grammar, clarity, and overall presentation of this manuscript. All ideation, core methodology development, and result collection were done by the authors.
>
> We have improved the notation throughout the manuscript. All main text changes have been highlighted in red. We further provide detailed responses to all individual reviewers.
>
> Again, we thank the reviewers for significantly improving the work through their constructive feedback. We hope that these revisions managed to appropriately address your suggestions, and that the reviewers will not hesitate to get back to us for any outstanding questions or concerns.

---

### Meta-Review · Area_Chair_nHws · 2025-12-30

**Summary:**

* Major weaknesses found by the reviewers
* Sensitivity to numerous hyperparameters (R7yb, 6V9n)
* Lack of better bandit baselines (fDBZ)
* Lack of multiple runs with clock time and other costs on x-axes (nTzs, 6V9n)

**Reviewer Concerns:**

* Sensitivity to numerous hyperparameters (R7yb, 6V9n): The authors presented a few more ablations with hyperparameters and provided a list of suggested hyperparameters. However, there are still outstanding concerns, for example, in Figure 18, the optimal threshold turns out to be different from the one originally used (0.995 vs 0.95). Similarly, as suggested by the rebuttal, many experiments have novelty filtering disabled. The rebuttal does, still makes me think that the system is sensitive to hyperparameters and better hyperparameters can be explored.

* Lack of better bandit baselines (fDBZ): The authors addressed this sufficiently and added more bandit-based baselines in their work.

* Lack of multiple runs with clock time and other costs on x-axes (nTzs, 6V9n): The next experiments address this sufficiently.

* The authors agree to NTZ's claims that their work is not truly open-ended discovery. Therefore, I’m recommending that the authors not make this claim as strongly as it is present in the paper.

* Discounting Smk8 reviews: The previous AC, and I do not believe the paper looks AI-generated; therefore, I'm discounting the 0 score. Nonetheless, I would recommend that the authors take Smk8's remaining advice and improve the writing. For example, even if the terms might be common in evolutionary algorithm literature, not every ICLR reader would be familiar with them. Therefore, it is important to define terms from evolutionary algorithms or, at the very least, refer to the prior work that the reader should look at.

Moreover, their last 3 concerns still need to be addressed:

Line 129: F, P_i are not defined before use. p_i is defined twice, differently.
r_i is both used for the rank (line 124) and for fitness value (line 210)
The name ShinkaEvolve is never explained.

**Reviewer Scores:**

* R7y b- 4: The sensitivity analysis is not sufficient, and I believe they would have kept the same rating.
* fDBZ - 8: Other bandit baselines would have resulted in improved score.
* 6V9n - 6: The sensitivity analysis is not sufficient, and I believe they would have kept the same rating.

---

### Decision · Program_Chairs · 2026-01-26

Accept (Poster)